# MYG1 drives glycolysis and colorectal cancer development through nuclear-mitochondrial collaboration

Jianxiong Chen[1,2,4], Shiyu Duan[1,2,4], Yulu Wang[1,2], Yuping Ling[1,2], Xiaotao Hou[1,2], Sijing Zhang[1,2], Xunhua Liu[1,2], Xiaoli Long[1,2], Jiawen Lan[1,2], Miao Zhou[2], Huimeng Xu[1,2], Haoxuan Zheng[3] ✉ & Jun Zhou [1,2] ✉

Metabolic remodeling is a strategy for tumor survival under stress. However, the molecular mechanisms during the metabolic remodeling of colorectal cancer (CRC) remain unclear. Melanocyte proliferating gene 1 (MYG1) is a 3′−5′ RNA exonuclease and plays a key role in mitochondrial functions. Here, we uncover that MYG1 expression is upregulated in CRC progression and highly expressed MYG1 promotes glycolysis and CRC progression independent of its exonuclease activity. Mechanistically, nuclear MYG1 recruits HSP90/GSK3β complex to promote PKM2 phosphorylation, increasing its stability. PKM2 transcriptionally activates MYC and promotes MYC-mediacted glycolysis. Conversely, c-Myc also transcriptionally upregulates MYG1, driving the progression of CRC. Meanwhile, mitochondrial MYG1 on the one hand inhibits oxidative phosphorylation (OXPHOS), and on the other hand blocks the release of Cyt c from mitochondria and inhibits cell apoptosis. Clinically, patients with KRAS mutation show high expression of MYG1, indicating a high level of glycolysis and a poor prognosis. Targeting MYG1 may disturb metabolic balance of CRC and serve as a potential target for the diagnosis and treatment of CRC.

Colorectal cancer (CRC) is the third most commonly diagnosed malignancy and the second leading cause of cancer-related death worldwide. It is often diagnosed at advanced clinical stages, and approximately 900,000 individuals die from this malignancy per year in the world[1]. The heterogeneity of tumors increases the difficulty and complexity of treatment. Metabolic remodeling, a hallmark of cancer, plays a crucial role in enabling CRC cells to survive in complex stress conditions[2]. Understanding the mechanism of metabolic remodeling may provide new insights and methods for preventing CRC and improving patient outcomes.

MYG1 (Melanocyte proliferating gene 1) is the only member of the uncharacterized protein family UPF0160. Proteins of this family contain a large number of metal-binding residues. This pattern suggests a phosphoesterase function[3]. The conserved DHH (Asp-His-His) motif is crucial for its 3′−5′ exonuclease activity[4,5]. Previous studies have demonstrated the existence of nuclear and mitochondrial targeting signals in the N-terminal region of human and mouse MYG1 proteins, indicating its nuclear and mitochondrial location[3,5]. In *S. cerevisiae*, MYG1 processes RNA and couples the translational program of the nucleus and mitochondria, acting as a coordinator of the nucleo-mitochondrial crosstalk. Its dual localization is essential for mitochondrial activity and cellular respiration, which is dependent on its exonuclease activity[5]. Notably, MYG1-deficient mice display alterations in stress-induced response[6], and emerging evidence suggests a

[1]Department of Pathology, Nanfang Hospital, Southern Medical University, Guangzhou 510515, China. [2]Department of Pathology, School of Basic Medical Sciences, Southern Medical University, Guangzhou 510515, China. [3]Guangdong Provincial Key Laboratory of Gastroenterology, Department of Gastroenterology, Nanfang Hospital, Southern Medical University, Guangzhou 510515, China. [4]These authors contributed equally: Jianxiong Chen, Shiyu Duan. ✉e-mail: ryan801218@163.com; jhzhou@smu.edu.cn; jzhou16@163.com

potential association between MYG1 and vitiligo development as well as antiviral responses[7–12]. A recent study has revealed its oncogenic role in cancer. MYG1 activates the AMPK/mTOR complex 1 signaling pathway and inhibits autophagy in lung adenocarcinoma cells[13]. Despite these findings, the specific function of MYG1 in tumors, particularly in CRC, remains poorly understood.

Cells derive energy through oxidative phosphorylation (OXPHOS) for diverse activities under physiological conditions. However, tumors enhance glycolysis and reshape their metabolism to provide the material foundation for rapid proliferation, division, and reshape the favorable metabolic microenvironment[14,15]. Pyruvate kinase (PK) is a key limiting factor of aerobic glycolysis. PK catalyzes the last and physiologically irreversible step in glycolysis, the conversion of phosphoenolpyruvate (PEP) to pyruvate by transferring a phosphate group to ADP. PKM2, wildly expressed in various tissues and cell types, plays an important role in the maintenance of the metabolic programs in various cancer cells[16]. PKM2 dimers and tetramers, composed of the same monomer, showed significantly different biological effects. PKM2 tetramer mainly functions as pyruvate kinase and the dimer PKM2 can regulate gene expression by acting as a protein transactivator along with HIF1α and β-Catenin, and acting as a protein kinase[16]. The synchronization of nuclear and mitochondrial functions is essential for cellular bioenergetics and tumor metabolic remodeling[17]. Some famous oncogenes and anti-oncogenes play a central role in cancer cell metabolic reprogramming[16–20].

We aim to identify additional proteins localized to both the nucleus and mitochondria and may play a central role in CRC metabolic reprogramming. In the present study, we uncovered an oncogene MYG1, which triggers a metabolic shift in CRC. Specifically, MYG1 promotes glycolysis through its functional coordination of nucleus and mitochondria, independent of its exonuclease activity. Understanding the function and mechanism of MYG1 may deepen our insight into tumor metabolic vulnerabilities and provide possibilities for targeted therapy.

## Results

### MYG1 is an oncogenic gene associated with CRC progression and clinical outcomes

To identify the key genes driving CRC progression and metabolic reprogramming, we first selected differentially expressed genes (DEGs) in CRC tissues compared with normal mucosa in public datasets (GSE24514, GSE9348, GSE20842, and GSE74602) with |log$_2$ fold change (log$_2$ FC)| >1.5 and $p < 0.01$. Among the 577 DEGs, 64 genes aberrantly expressed in both adenoma tissues compared to normal mucosa (GSE20916) and metastatic cells compared to primary tumor cells from the same patient (GSE1323) with |log$_2$ FC | > 1.5 and $p < 0.05$ were further selected as candidates that might drive CRC progression. Among them, we identified TXNIP and MYG1, which have both nuclear and mitochondrial localization (Supplementary Fig. 1a). TXNIP is an oxidative stress-responsive signal transducer redoxisome with a wide range of biological functions. It is a regulator of glucose and lipid metabolism, and its function has been widely reported in various cancers including CRC[18–21]. However, the role of MYG1 (also named C12orf10) in metabolism and CRC development remains unclear. Subsequently, we performed the Gene Set Enrichment Analysis (GSEA) using the RNA-seq datasets from the Cancer Genome Atlas (TCGA) colon and rectal cancer (COADREAD) to understand the potential metabolic function of MYG1 in CRC. As shown in Fig. 1a, MYG1 may participate in multiple metabolic pathways (Supplementary Data 1). We next explored the role of MYG1 in CRC progression and metabolic reprogramming.

To validate the aberrant expression of MYG1 in CRC, we measured the expression levels of MYG1 in CRC and paired normal mucosa samples. The results showed that MYG1 expression was upregulated in CRC compared to adjacent normal tissues (Supplementary Fig. 1b and

Fig. 1b) and higher in patients with distant metastasis (mCRC) compared to patients with non-metastasis (nmCRC) (Supplementary Fig. 1b). We also confirmed the result in 149 paired formalin-fixed paraffin-embedding (FFPE) samples from NF-CRC1 cohort (Fig. 1c). Clinical correlation analysis revealed that MYG1 expression was significantly associated with the lymph node metastasis, distant metastasis, T stage and clinical stage (Supplementary Table 1). Furthermore, survival analysis of the TCGA COADREAD and NF-CRC1 cohorts indicated that patients with high MYG1 expression showed poorer overall survival and progression-free survival (Supplementary Fig. 1c and Fig. 1d).

Additionally, the expression of MYG1 in CRC development was evaluated in NF-CRC2 cohort containing the normal mucosa, adenoma, early and advanced carcinoma. The IHC results showed that MYG1 was upregulated gradually in the evolution of CRC (Supplementary Fig. 1d, e). To clarify the reason for the upregulation of MYG1, GISTIC2[22] was used to analyse the copy number variation (CNV) of MYG1 in TCGA COADREAD cohort. The results revealed that MYG1 was rarely amplified in CRC patients (Fig. 1e). This suggested that CNV does not account for the upregulation of MYG1. The loss of the tumor suppressor APC and KRAS mutation are the early events in CRC tumor progression. We next investigated whether APC loss and KRAS mutation were associated with MYG1 expression. By comparing the mRNA levels of *Myg1* in *Apc$^{fl/fl}$*, *Kras$^{G12D/+}$*, and *Apc$^{fl/fl}$Kras$^{G12D/+}$* mice (GSE160478), we found that *Myg1* was significantly upregulated in *Apc$^{fl/fl}$Kras$^{G12D/+}$* mice (Supplementary Fig. 1f). Interestingly, we also found that MYG1 was upregulated in cell lines with KRAS mutation compared to KRAS wild-type cells (Fig. 1f and Supplementary Fig. 1g). To validate the above finding, we further analyzed the correlation of MYG1 expression and KRAS mutation. MYG1 was upregulated in specimen with KRAS mutation instead of APC mutation in TCGA COADREAD cohort (Fig. 1g and Supplementary Fig. 1h). In addition, MYG1 expression was higher in tissues with KRAS mutation compared to KRAS wild-type patients in NF-KRAS cohort (Fig. 1h). In summary, these results suggest that upregulation of MYG1 is closely related to the progression of CRC, especially those with KRAS mutation, and suggests poorer prognosis in patients.

### MYG1 accelerates CRC proliferation and metastasis in vitro and in vivo

To investigate the function of MYG1 in CRC cellular behaviors, we first knocked down MYG1 using two shRNAs in RKO and LoVo cells (Supplementary Fig. 2a). MYG1 knockdown strikingly inhibited proliferation, colony formation, invasion and migration ability of CRC cells (Fig. 2a–d and Supplementary Fig. 2b–d). To investigate whether the effect of MYG1 is dependent on its enzymatic activity, we generated an exonuclease inactive MYG1 mutant by replacing the DHH residue with ALL (MYG1$^{ALL}$) as previously described[5]. Upon flag-tagged wild-type MYG1 and MYG1$^{ALL}$ expression at comparable levels in HCT116 and SW480 cells (Supplementary Fig. 2e), MYG1$^{ALL}$ also promoted cell proliferation, colony formation, invasion and migration, although to a lesser extent than wild-type MYG1 (Fig. 2e–h and Supplementary Fig. 2f–h). These results implied that MYG1 has exonuclease-independent cellular functions in CRC. To exclude off-target effect of shRNAs, we also used a sgRNA to knockout MYG1 in LoVo cells (Supplementary Fig. 2i). MYG1 knockout blocked the proliferation, colony formation, invasion and migration ability of LoVo cells (Fig. 2i–l and Supplementary Fig. 2j–l).

We next extended our study in vivo by employing a subcutaneous xenograft mouse model. Silencing MYG1 inhibited tumor growth and upregulating MYG1 showed the opposite effect (Fig. 2m). MYG1 accelerated CRC cell division as detected by Ki-67 IHC staining in mice tumors (Supplementary Fig. 2m, n). Additionally, metastasis ability was evaluated in a liver metastasis mouse model of CRC by injecting cells into the velamen of the spleen. The results showed that MYG1

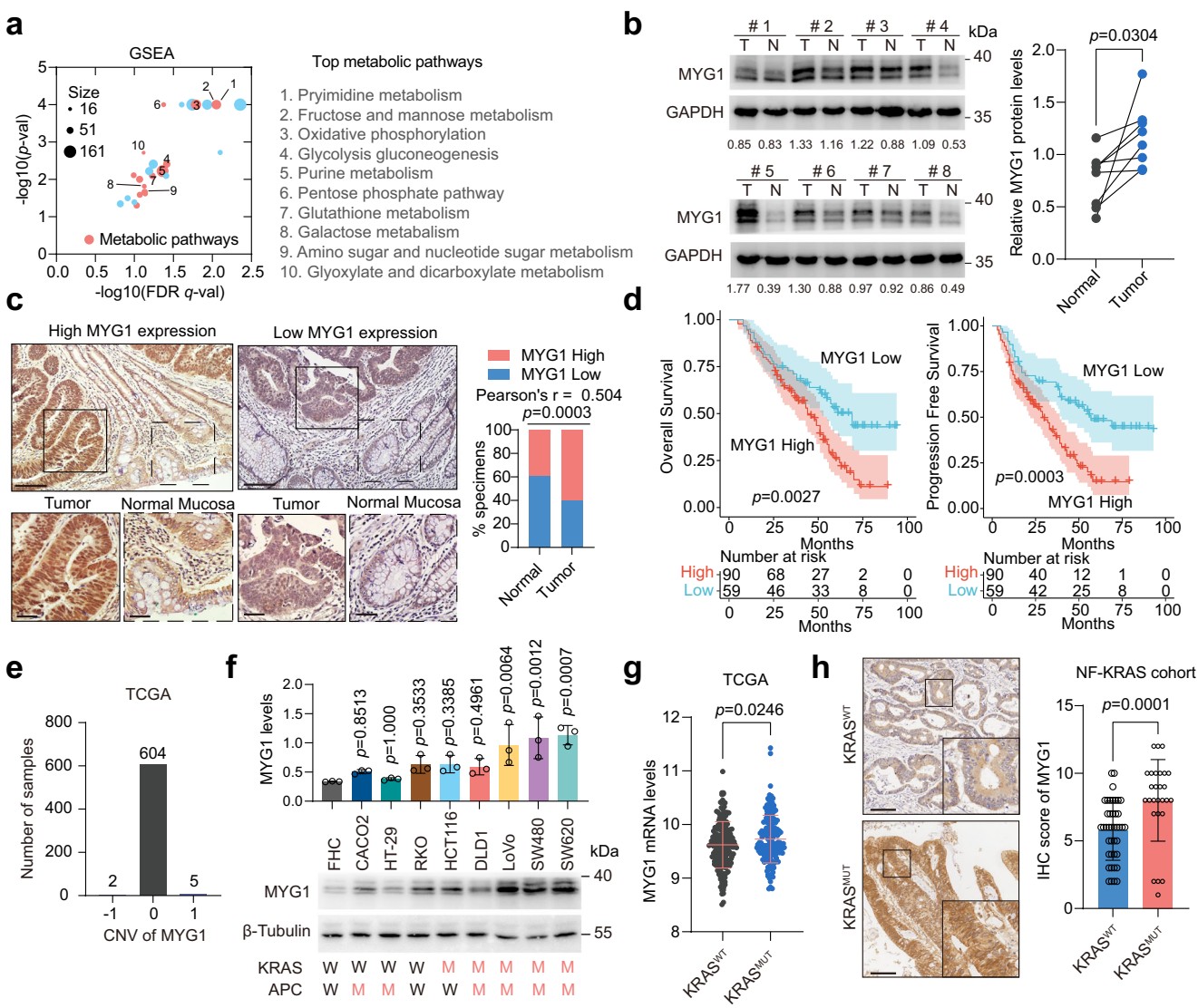

**Fig. 1 | MYG1 is an oncogenic gene associated with CRC progression and poor clinical outcomes. a** Enriched KEGG pathways of GSEA in TCGA COADREAD cohort (n = 433). Top 10 metabolic pathways were labeled. Red: metabolic pathways. FDR: False Discovery Rate. **b** MYG1 expression in normal mucosa and paired tumor samples were tested by western blot (n = 8) followed by quantification. Quantification is for the presented representative blot from n = 3 independent experiments. **c** Representative image and quantification of MYG1 IHC analysis of CRC tumor and adjacent normal mucosal in NF-CRC1 cohort (n = 149 pairs). Scale bar, 200 µm (main macrophages) and 50 µm (insets). **d** Kaplan–Meier analysis of overall survival and progression free survival of patients in NF-CRC1 cohort (n = 149 pairs) based on MYG1 expression evaluated in **c. e** The number of MYG1 CNV events in TCGA COADREAD cohort (n = 611). **f** The level of MYG1 protein was detected in normal colon mucosa cell line FHC and CRC cell lines by western blot (bottom) and followed by quantification (top). KRAS and APC status of cell lines were labeled below.

W, wild type; M, mutation. The samples derive from the same experiment but different gels for MYG1, and another for β-Tubulin were processed in parallel. n = 3 independent experiments. **g** Transcription levels (log$_2$ (x + 1) transformed RSEM normalized count) of MYG1 in TCGA COADREAD cohort with KRAS wild type (WT, n = 194) and mutation (MUT, n = 163). **h** Representative images and quantification of MYG1 IHC analysis in NF-KRAS cohort with KRAS wild type (WT, n = 41) and mutation (MUT, n = 27). Scale bar, 100 µm. Two-sided permutation test for GSEA and two-sided hypergeometric test for KEGG pathway enrichment analysis (**a**). Paired two-sided Student's t-test (**b**). Two-sided Pearson Chi-square test (**c**). Log-rank test (**d**). One-way ANOVA, Dunnett's multiple comparisons test, comparison with FHC (**f**). Unpaired two-sided Student's t-test (**g**). Two-sided Mann–Whitney test (**h**). p and r values were provided in the figure. Error bars, mean ± SD. See also Supplementary Fig. 1. Source data are provided as a Source Data file.

increased metastasis formation in the liver (Fig. 2n–o). These results reveal that MYG1 accelerates the proliferation and metastasis of CRC in vitro and in vivo.

Besides, we detected the effect of MYG1 on cell cycle. The results showed that MYG1 accelerates the G1-S cell cycle transition (Supplementary Fig. 2o). We next detected the expression of key proteins regulating epithelial-mesenchymal transition (EMT), a pivotal process for the invasion of tumor cells, and observed that MYG1 promoted the expression of β-Catenin, Vimentin and Fibronectin, but inhibited the expression of ZO-1 and E-Cadherin, suggesting that MYG1 enhanced

EMT in CRC cells (Fig. 2p–q and Supplementary Fig. 3). Together, these data clearly demonstrate the oncogenic role of MYG1 in CRC.

## MYG1 promotes aerobic glycolysis of CRC cells independent of its exonuclease activity

Given that MYG1 might participate in metabolic pathways, we screened the transcriptomic alterations in LoVo cells with MYG1 knocked out. By examining the functional enrichment of differentially expressed genes, we found that MYG1 may regulate metabolic processes, including glycolysis (Fig. 3a), consistent with GSEA analysis of TCGA COADREAD

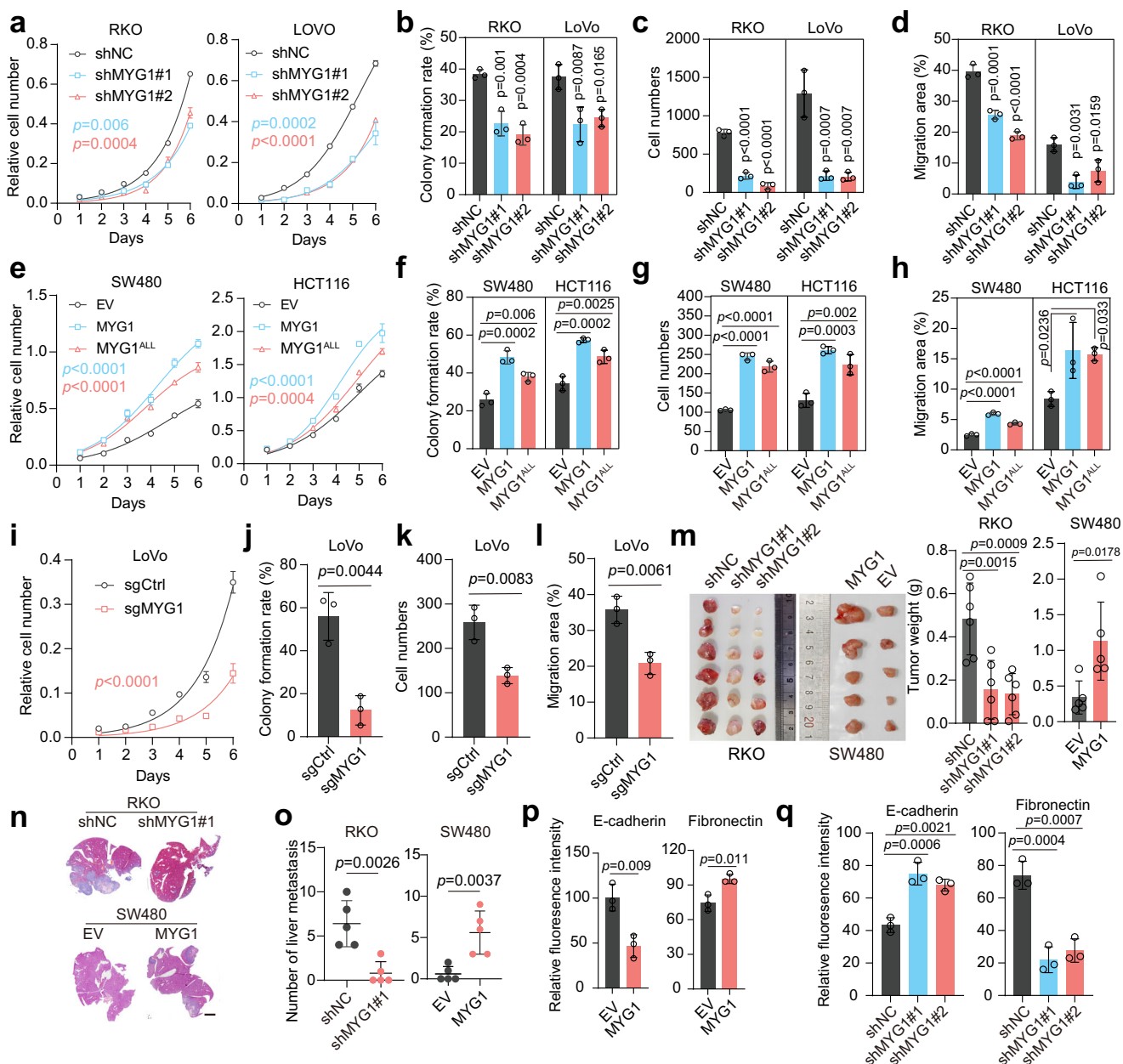

**Fig. 2 | MYG1 accelerates CRC proliferation and metastasis in vitro and in vivo.**
**a–l** RKO and LoVo cells were infected with lentivirus expressing control shRNA (shNC) and two targeting shRNAs (shMYG1#1, shMYG1#2). SW480 and HCT116 cells were infected with lentivirus expressing empty vector (EV), MYG1, and MYG1^ALL. LoVo cells were transfected with control sgRNA (sgCtrl) and sgMYG1. Tumor biological behaviors of CRC cells as indicated were examined by proliferation assay (**a, e, i**) (*n* = 3 technical replicates, representative data from *n* = 3 independent experiments), colony formation assay (**b, f, j**), transwell invasion assay (**c, g, k**) and wound healing assay (**d, h, l**), respectively. *p* value represents comparison with shNC group (**b, c,** and **d**). *n* = 3 independent experiments (**b–d, f–h,** and **j–l**). **m–o** RKO and SW480 cells treated as indicated were utilized to establish subcutaneously xenograft tumor model and liver metastasis model of CRC in BALB/c

nude mice (xenograft tumor model: *n* = 6 in RKO and *n* = 5 in SW480; liver metastasis model: *n* = 5 in each group). The photograph (left) and weight (right) of subcutaneously xenograft tumors (**m**). Representative images of H&E staining in mouse liver from a CRC liver metastasis model (**n**). Scale bar, 3 mm. The number of liver metastatic lesions was counted in CRC liver metastasis models (**o**). **p–q** Fluorescence intensity quantification of E-cadherin and Fibronectin in SW480 (**p**) and RKO (**q**) cells (*n* = 3 independent experiments). Two-way ANOVA, Tukey's multiple comparisons test (**a, e,** and **i**). Unpaired two-sided Student's *t*-test (**j–m** and **o–p**). One-way ANOVA, Dunnett's multiple comparisons test (**b–d, f–h, m,** and **q**). *p* value was provided in the figure. Error bars, mean ± SD. See also Supplementary Figs. 2 and 3. Source data are provided as a Source Data file.

cohort (Supplementary Fig. 4a). To further validate this, we analyzed the correlation between MYG1 and genes involved in glycolysis in TCGA COADREAD cohort. The results showed that MYG1 positively correlated with several genes, most of which were transcriptionally regulated by c-Myc and HIF1α (Supplementary Fig. 4b). We next evaluated the effect of MYG1 on the glycolysis-related genes that are highly correlated with MYG1 in HCT116 cell. The results showed that MYG1

promoted the expression of LDHA, GLUT1, and PKM2 (Fig. 3b). The positive expression correlation of MYG1 with LDHA, GLUT1, and PKM2 was also validated in another CRC cohort (GSE39582) (Supplementary Fig. 4c). Increased expression of GLUT1 and LDHA can facilitate glucose uptake and lactate production. We then measured mitochondrial function and aerobic glycolysis in CRC cells to evaluate the effect of MYG1 on glucose metabolic process. The results showed that the

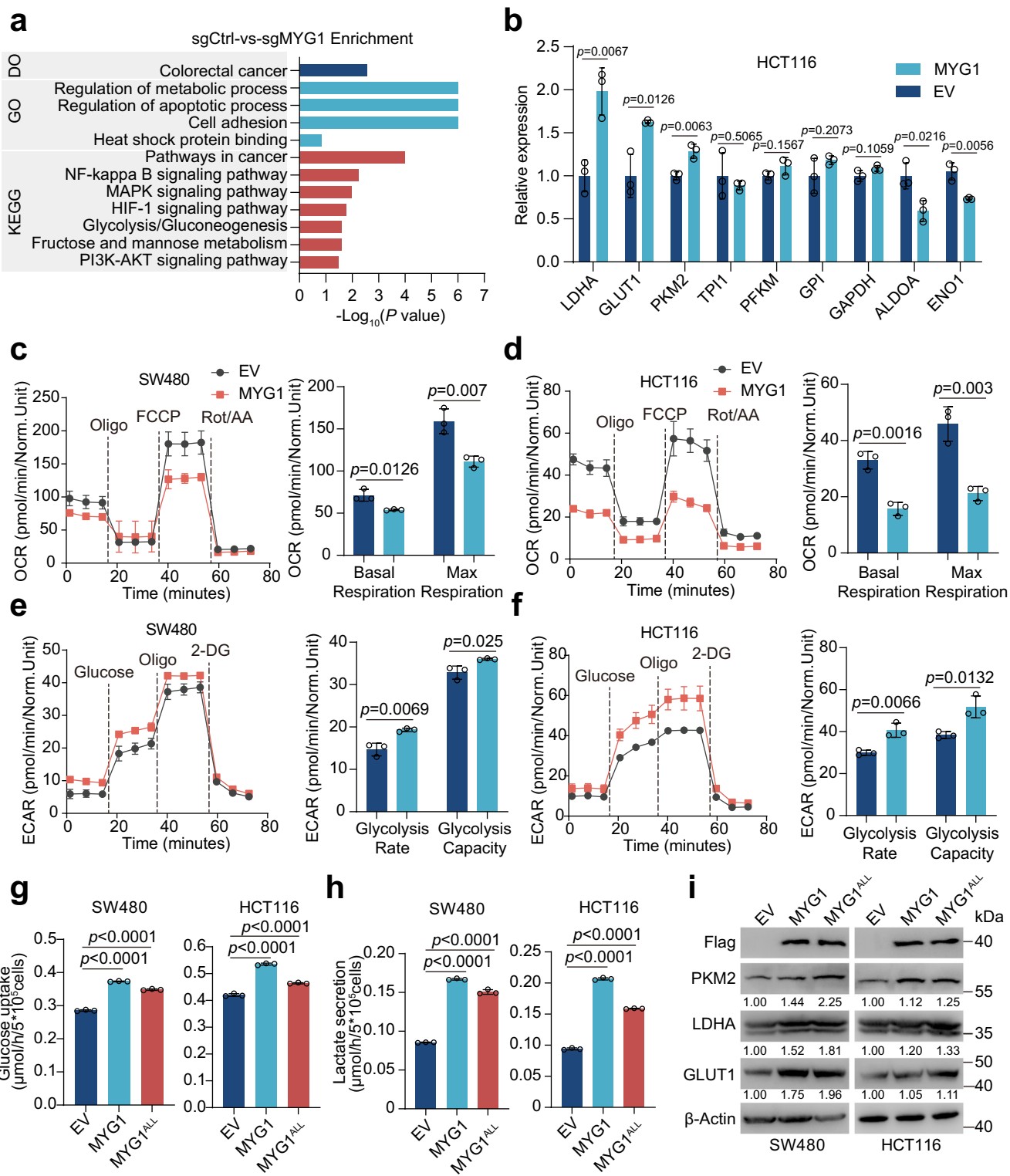

oxygen consumption rates (OCR) were significantly decreased in MYG1 overexpressed cells and increased in cells with MYG1 silenced (Fig. 3c–d and Supplementary Fig. 4d–e). MYG1 also increased the extracellular acidification rate (ECAR), while silencing MYG1 decreased the ECAR (Fig. 3e–f and Supplementary Fig. 4f–g). We also evaluated whether the function of MYG1 promoting glycolysis depends on its exonuclease activity. The results showed that MYG1[ALL] promotes glucose uptake and lactate secretion although to a lesser extent compared to wild-type (Fig. 3g–h). The results of western blot also indicated that MYG1[ALL] promotes the expression of GLUT1, LDHA, and

PKM2 as well as wild-type (Fig. 3i). Taken together, these data indicate that MYG1 promotes glycolysis in CRC cells independent of its exonuclease activity.

As previous research suggested that MYG1 may promote OXPHOS in lung adenocarcinoma (LUAD)[13], we next assessed the potential reasons behind this difference. Firstly, we examined the impact of MYG1 on glucose metabolism in human LUAD cell line PC9. We observed that both MYG1[ALL] and wild-type MYG1 significantly inhibited glucose uptake and lactate secretion of PC9 cells (Supplementary Fig. 4h–i). This suggested that although MYG1 can promote tumor

**Fig. 3 | MYG1 promotes aerobic glycolysis of CRC in vitro. a** Bar plot of Disease Ontology (GO), Gene Ontology (GO), and KEGG analysis results based on RNA-seq data of control and MYG1 knockout LoVo cells. $n = 3$ samples generated after independent generation of cells and processed on different days. **b** Relative mRNA expression of glycolysis-related genes in control and stable MYG1 overexpressed HCT116 cells detected by RT-qPCR ($n = 3$ technical replicates, representative data from $n = 3$ independent experiments). OCR was determined in control and stably MYG1 overexpressed SW480 (**c**) and HCT116 (**d**) cells (left). Basal respiration and max respiration were analyzed (right). ECAR was determined in control and MYG1 overexpressed SW480 (**e**) and HCT116 (**f**) cells (left). Glycolysis rate and glycolysis capacity were analyzed (right). Norm.Unit represents the normalized OCR and ECAR. Glucose uptake (**g**) and lactate secretion (**h**) of control, stable MYG1 and MYG1[ALL] overexpressed SW480 and HCT116 cells. **i** Protein expressions of PKM2, LDHA, and GLUT1 in control, stable MYG1 and MYG1[ALL] overexpressed SW480 and HCT116 cells were tested by western blot. The samples derive from the same experiment but different gels for Flag, another for PKM2, another for LDHA and GLUT1, and another for β-Actin were processed in parallel. The quantification provided under the blots is for the representative blot from $n = 3$ independent experiments. $n = 3$ independent experiments (**c**–**h**). Two-sided Hypergeometric test (**a**). Unpaired two-sided Student's $t$-test (**b**–**f**). One-way ANOVA, Dunnett's multiple comparisons test (**g**–**h**). $p$ value was provided in the figure. Error bars, mean ± SD. See also Supplementary Fig. 4. Source data are provided as a Source Data file.

progression in both CRC and LUAD, it regulates metabolism in contrasting roles independent of its enzymatic activity. As KRAS mutation is also very prevalent in non-small cell lung cancer, especially in LUAD, we further evaluated the relationship between MYG1 expression and KRAS mutation in LUAD. Interestingly, MYG1 expression was not associated with KRAS mutations in TCGA LUAD cohort (Supplementary Fig. 4j). We reasoned that the contrary functions of MYG1 in LUAD and CRC metabolism may be attributed to the heterogeneity among cancer types, owing to distinct upstream regulators as well as downstream function networks.

## Unveiling the dual localization of MYG1: emphasizing nucleus-driven glycolysis and tumor progression

As previous studies have reported that MYG1 has a dual location of nucleus and mitochondria in several cells[3,5], we first validated its subcellular location in CRC cells through cell component separation. The results showed that MYG1 was located in the nucleus and mitochondria (Fig. 4a). We further performed the proteinase K shaving assay to validate the mitochondrial location of endogenous and exogenous MYG1 in 293 T and LoVo cells. The results showed that MYG1 was located in the mitochondria instead of the mitochondrial outer membrane (Fig. 4b and Supplementary Fig. 5a, b). Immunoelectron microscopy was also employed to observe the location of MYG1 and the results showed that MYG1 was located in mitochondria (Fig. 4c). These results concluded that MYG1 has dual localization of nucleus and mitochondria.

Philips MA et al. have found that there was no shuttle of MYG1 between nucleus and mitochondria[3]. Based on this, we wondered whether nuclear or mitochondrial MYG1 plays a role in promoting glycolysis and CRC progression. According to previous study[3], we overexpressed MYG1 with both signal peptide reserved (wild-type), both signal peptide deleted (MYG1[ΔL]), mitochondrial localization peptide deleted (MYG1[N]), and nuclear localization peptide deleted (MYG1[M]) in CRC cells and validated their expression (Fig. 4d and Supplementary Fig. 5c). To validate the location of MYG1 variants, we performed IF in HCT116 cells expressing different MYG1 variants and the exogenous MYG1 was correctly located in the corresponding cell compartment (Fig. 4e). To clarify which part of MYG1 participated in promoting glycolysis, we measured glucose uptake and lactate secretion levels in cells expressing different variants. As shown in Fig. 4f, MYG1 significantly increased glucose uptake and lactate secretion in HCT116 and SW480 cells, whereas MYG1[N] showed a dominant effect on glucose uptake and lactate secretion. These results suggested that nuclear MYG1 plays a pivotal role in glycolysis in CRC cells.

To validate the result in vivo, we constructed a CRC orthotopic mouse model using HCT116 cells expressing different variants. Tumor formation was confirmed using the in-vivo imaging systems 30 days after transplantation (Fig. 4g). Luciferin signal suggested that the MYG1[N] tumor had a more rapid proliferation in comparison to MYG1[M] counterparts (Fig. 4h). [18]F-FDG uptake of different tumors was then detected by PET/CT scanning to evaluate the glucose uptake of tumors (Fig. 4i). The [18]F-FDG micro-PET/CT data showed that MYG1 induced

glucose uptake in the tumor, and MYG1[N] showed the dominant effect (Fig. 4j). In addition, we also evaluated the effect of different MYG1 variants on tumor weight (Fig. 4k) and progression. H&E staining of tumors showed that MYG1[N] tumors were more aggressive as well as wild-type (Supplementary Fig. 5d–e). We next examined the expression of glucose transporter GLUT1 in different tumors by IHC. The results showed that MYG1[N] promoted the expression of GLUT1, while MYG1[M] showed negligible effect (Supplementary Fig. 5d, f). These results imply that MYG1[N] plays a dominant role in glycolysis and CRC progression.

To test the effect of MYG1 variants on tumor progression, we also performed in vitro experiments. Proliferation and transwell invasion assays showed that MYG1[N] promoted proliferation and invasion as well as wild-type, while MYG1[M] showed a minor effect and MYG1[ΔL] did not show any effect (Supplementary Fig. 5g–h). These results showed that both MYG1[N] and MYG1[M] had tumor-promoting effects and the function of MYG1[N] was decisive. However, there was no statistical difference between MYG1[M] and the control group in the animal model, indicating that the effect of MYG1[M] was too weak to display differences in complex in-vivo environments.

To investigate whether the oncogenic role of MYG1[N] was dependent on glycolysis, we treated the cells with 2-DG, an inhibitor of glycolysis, and detected their proliferation and invasion abilities. The results showed that oncogenesis induced by MYG1[N] was efficiently abrogated by 2-DG, as detected by transwell invasion and colony formation assays (Fig. 4l–m and Supplementary Fig. 5i–j). Collectively, these results indicate that MYG1[N] promotes CRC progression via glycolysis.

We also evaluated the glucose uptake and lactate secretion levels in PC9 cells expressing different variants, the results showed that MYG1[M] inhibited glucose uptake and lactate secretion as well as wild-type, and MYG1[N] showed a weaker effect (Supplementary Fig. 5k). These results suggested that MYG1 has different functional patterns in CRC and LUAD cells.

## MYG1 recruits HSP90 to phosphorylate PKM2 and increases the stability of PKM2 in the nucleus

As the above results showed that both nuclear and mitochondrial MYG1 can promote the progression of CRC, we speculate that MYG1 functions independently in the nucleus and mitochondria. Considering this, we separated the nucleus and mitochondria in SW480 cells overexpressing MYG1 and performed co-immunoprecipitation (Co-IP). The proteins that potentially interacted with MYG1 were identified using immunoprecipitation-mass spectrometry (IP-MS) (Supplementary Fig. 6a). There are different MYG1 interacting proteins in the nucleus and mitochondria, which also imply that MYG1 has different functions in the nucleus and mitochondria. Among 180 candidate proteins in the nucleus (Supplementary Data 2), we focused on PKM2, a key enzyme that catalyzes the conversion of phosphoenolpyruvate to pyruvate and plays a pivotal role in glycolysis of CRC[23]. PKM2 is mainly located in the cytoplasm. EGF can activate EGFR/Ras/Raf signaling, increase PKM2 phosphorylation and nuclear accumulation,

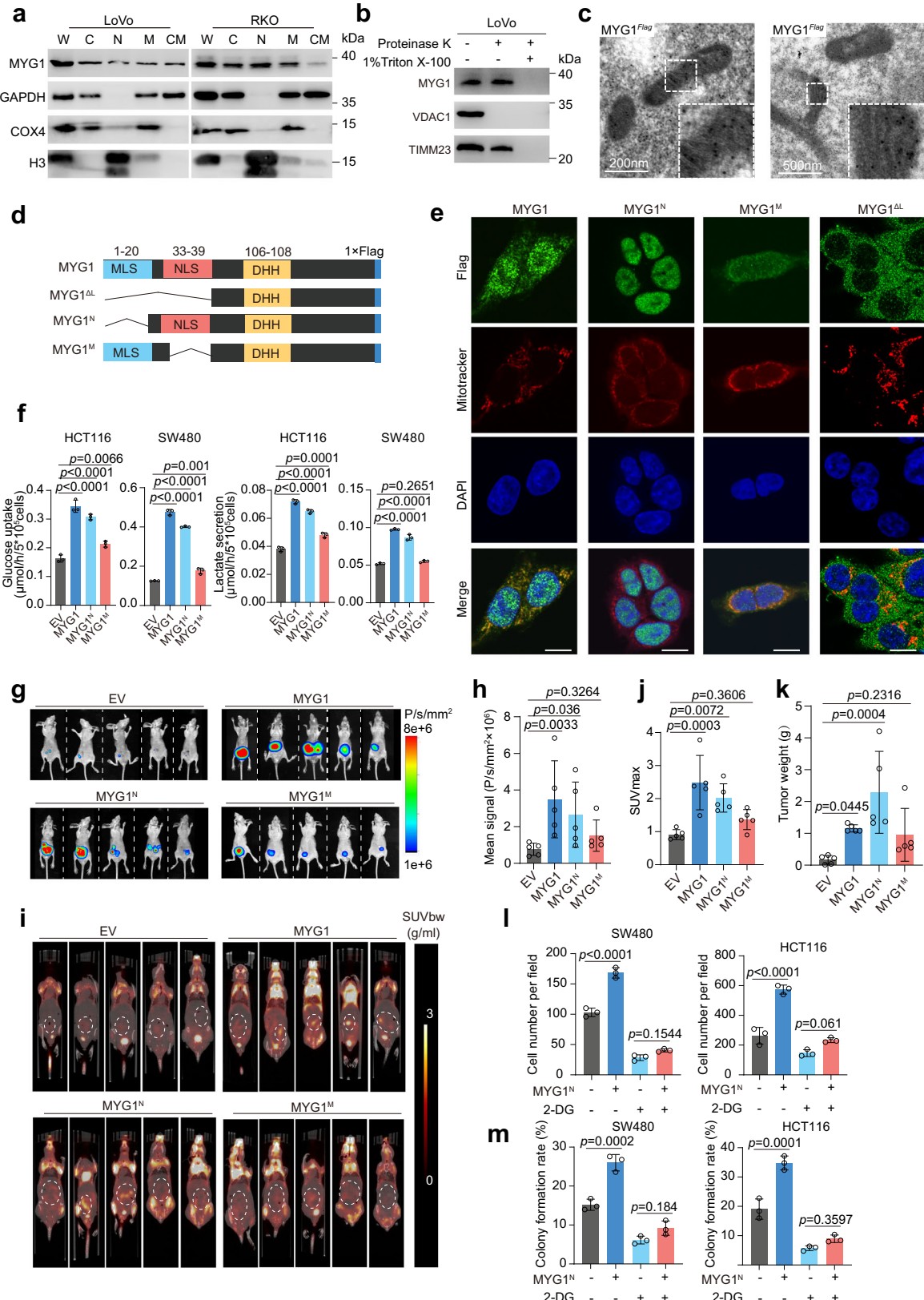

promoting tumorigenesis and progression[24,25]. To validate the interaction of MYG1 and PKM2 in the nucleus, we treated HCT116 and LoVo cells with EGF. EGF increases the nuclear translocation of PKM2 in 293 T cells, but its role in LoVo cells with KRAS continuously activated is limited (Supplementary Fig. 6b). Firstly, we observed the partial co-localization of MYG1 and PKM2 in the nucleus of 293 T and HCT116

cells with EGF treated, as well as SW480 cells without EGF treatment (Fig. 5a). We next performed Co-IP and found that MYG1 interacted with PKM2 and EGF treatment increased the interaction in 293 T and HCT116 cells instead of LoVo cells (Fig. 5b–e). Nucleocytoplasmic separation followed by Co-IP further confirmed that MYG1 interacted with PKM2 in the nucleus (Fig. 5f). We also observed partial co-

**Fig. 4 | Nuclear MYG1 plays a more critical role in promoting CRC progression through glycolysis. a** MYG1 expression in the subcellular compartments of cells were determined by western blot. W, whole cell lysis. C, cytoplasm. N, nucleus. M, mitochondria. CM, cytoplasm with mitochondria removed. The samples derive from the same experiment but different gels for MYG1 and COX4, and another for GAPDH and H3 were processed in parallel. **b** Mitochondrial fractions were analyzed by protease K shaving assay. VDAC1 was sensitive to protease K treatment, whereas TIMM23 was resistant. Proteins were digested by protease K in the presence of 1% Triton X-100. The samples derive from the same experiment but different gels for MYG1 and VDAC1, and another for TIMM23 were processed in parallel. **c** Images of Flag-tagged MYG1 in mitochondria of 293 T cells as detected by immunoelectron microscopy. Scale bar, 200 nm (left) and 500 nm (right). Representative of 25 images from *n* = 2 independent experiments. **d** Schematic of MYG1 domain and variant constructs. **e** Cellular location of Flag-tagged MYG1 variants was confirmed by IF in HCT116 cells. Scale bar, 10 μm. Representative images from *n* = 2

independent experiments. **f** The levels of glucose uptake and lactate secretion were detected. **g–k** Luciferase-labeled HCT116 were utilized to establish orthotopic CRC mouse models (*n* = 5 in each group). Bioluminescence imaging of mice was detected on the 30th day (**g**) and the signals were quantified (**h**). PET/CT images with the signals indicated SUV normalized to body weight (SUVbw) of mice. The maximum [18]F-FDG uptake value (SUVmax) was obtained by browsing different layers of PET imaging (**i**, dashed cycle marked) and quantified (**j**). Tumors were separated and weighted (**k**) after sacrificing the mice. The invasion (**l**) and colony formation ability (**m**) were detected. *n* = 3 independent experiments (**a–b**, **f**, and **l–m**). Kruskal–Wallis test, Dunn's multiple comparisons test (**h** and **k**). One-way ANOVA, Tukey's multiple comparisons test (**f** and **j**). Two-way ANOVA, Tukey's multiple comparisons test (**l–m**). *p* value was provided in the figure. Error bars, mean ± SD. See also Supplementary Fig. 5. Source data are provided as a Source Data file.

localization of MYG1 and PKM2 in KRAS mutated CRC tissues in the nucleus (Supplementary Fig. 6c). Subsequently, we constructed truncated fragments of MYG1 (Fig. 5g) and explored the binding sequence of MYG1 to PKM2 by Co-IP. The results showed that MYG1 directly interacts with PKM2 through the 149-199 fragment (Fig. 5g).

Previously, we found that MYG1 upregulated PKM2 expression in CRC cells (Fig. 3b). We then determined the effect of MYG1 variants on the expression of PKM2. Western blot showed that MYG1[N] promoted PKM2 expression as well as wild-type, while MYG1[M] did not show any effect and MYG1 knockout inhibited PKM2 expression (Fig. 5h and Supplementary Fig. 6d). Moreover, we determined whether MYG1 could affect the pyruvate kinase activity of PKM2. Notably, MYG1[N] and MYG1[M], as well as wild-type did not increase the enzyme activity of PKM2 (Supplementary Fig. 6e). Previous studies reported that PKM2 dimers are mainly distributed in the nucleus and have kinase activity, whereas PKM2 tetramers are mainly distributed in the cytoplasm and have pyruvate kinase activity. Therefore, we doubted whether MYG1[N] led to the accumulation of PKM2 in the nucleus. We next performed western blot and IF to determine the distribution of PKM2. The results showed that MYG1[N] can induce the obvious accumulation of PKM2 in the nucleus (Supplementary Fig. 6f–g). We also crosslinked the polymer form of PKM2 in SW480 cells, and western blot showed that MYG1[N] increased the dimers of PKM2 (Supplementary Fig. 6h). Taken together, these results demonstrate that MYG1[N] interacts with PKM2 and induces PKM2 accumulation in the nucleus.

To clarify how MYG1 induced the accumulation of PKM2 in the nucleus, we first detected the effect of MYG1 on the stability of PKM2. Knockout MYG1 in LoVo cells significantly promoted the degradation of PKM2 (Fig. 5i). This suggested that MYG1 can enhance the stability of PKM2. RNA-seq data suggested that MYG1 may be related to heat shock protein binding ability (Fig. 3a) and two independent studies have reported the possible interaction of MYG1 with HSP90[26,27]. Considering the role of HSP90 in regulating PKM2 abundance[28], we validated the interaction between MYG1 and HSP90, as well as PKM2 and HSP90. The results showed that HSP90 interacted with MYG1 and PKM2, and EGF increased the interaction between MYG1 and HSP90 in 293 T cells instead of LoVo cells with KRAS mutation. (Fig. 5j–k). As a molecular chaperone, HSP90 recruits client protein GSK3β and forms a complex to phosphorylate PKM2 at Thr-328, increasing its stability in hepatocellular carcinoma[28]. Thus, we doubted whether MYG1 recruits HSP90 and promotes PKM2 phosphorylation in CRC cells. We also validated the interaction of GSK3β with MYG1 and PKM2, and EGF also increased the interaction between MYG1 and GSK3β in 293 T cells (Fig. 5l–m). In addition, MYG1 promoted the serine/threonine (Ser/Thr) phosphorylation of PKM2, and MYG1[Δ], which lacks the binding sequence with PKM2, does not show the effect (Fig. 5n). GSK3 inhibitor IX, a GSK3β inhibitor, inhibited the MYG1-induced phosphorylation of PKM2 (Fig. 5n). Besides, we also silenced HSP90 using two siRNAs and knockdown of HSP90 inhibited the Ser/Thr phosphorylation of PKM2

(Fig. 5o). Together, these results indicates that MYG1 recruits HSP90 to phosphorylate PKM2 and induces the accumulation of PKM2 in the nucleus.

## MYG1 accelerates glycolysis through PKM2/c-Myc signaling pathway

To validate the role of PKM2 in MYG1-induced glycolysis, we silenced PKM2 and detected the glucose uptake and lactate secretion as well as OCR and ECAR in CRC cells. The results showed that silencing PKM2 blocked the increased glucose uptake, lactate secretion, and ECAR induced by MYG1 and increased the OCR that was blocked by MYG1 (Fig. 6a–b and Supplementary Fig. 7a–b). We next analyzed the glucose uptake and lactate secretion in cells expressing MYG1[Δ], and the results showed that deleting the PKM2 binding sequence of MYG1 significantly inhibited the glycolysis levels compared with wild-type (Fig. 6c). Moreover, the proliferation and invasion ability of CRC cells and the tumor growth in vivo were also greatly inhibited compared to its wild-type counterpart (Fig. 6d–f and Supplementary Fig. 7c). Besides, IHC analysis of mice tumors also indicated that MYG1[Δ] inhibited the expression of GLUT1 and PKM2, implying low glycolysis levels of MYG1[Δ] tumors (Supplementary Fig. 7d–e). These data suggested the pivotal role of PKM2 in MYG1-induced glycolysis and oncogenic functions.

In addition, we treated CRC cells with C3K, which can selectively inhibit pyruvate kinase activity of PKM2 at an appropriate concentration[29]. We treated cells with C3K in a concentration gradient and the pyruvate kinase activity of cells was decreased (Supplementary Fig. 8a). IF and western blot showed that C3K did not affect the expression of PKM2 and MYG1, as well as its distribution in cytoplasm and nucleus (Supplementary Fig. 8b–d). Cells expressing MYG1[N] were treated with C3K and the invasion and colony formation ability were detected. Although the oncogenic abilities of cells were both downregulated, C3K did not influence the oncogenic effect of MYG1[N] in HCT116 cells compared to the control group (Supplementary Fig. 8e), indicating that the function of MYG1[N] was independent of the pyruvate kinase activity of PKM2.

Previous research has shown that PKM2 can transcriptionally regulate MYC expression in the nucleus, independent of its pyruvate kinase activity[24]. MYC can promote the transcription of genes involved in glycolysis, such as GLUT1, LDHA, and PKM2. The GSEA results in TCGA COADREAD cohort also suggested that the expression of MYG1 was related to MYC targets (Fig. 6g). Based on these, we speculated that MYG1 might promote the expression of MYC and thus promote the transcription of glycolysis-related target genes. We first detected the regulation of MYG1 on c-Myc by western blot, and the results showed that MYG1[N] promoted the expression of c-Myc, whereas MYG1[M] had no obvious effect (Fig. 6h), and silencing MYG1 inhibited the expression of c-Myc (Supplementary Fig. 8f). Subsequently, we analyzed the expression of PKM2 and c-Myc in

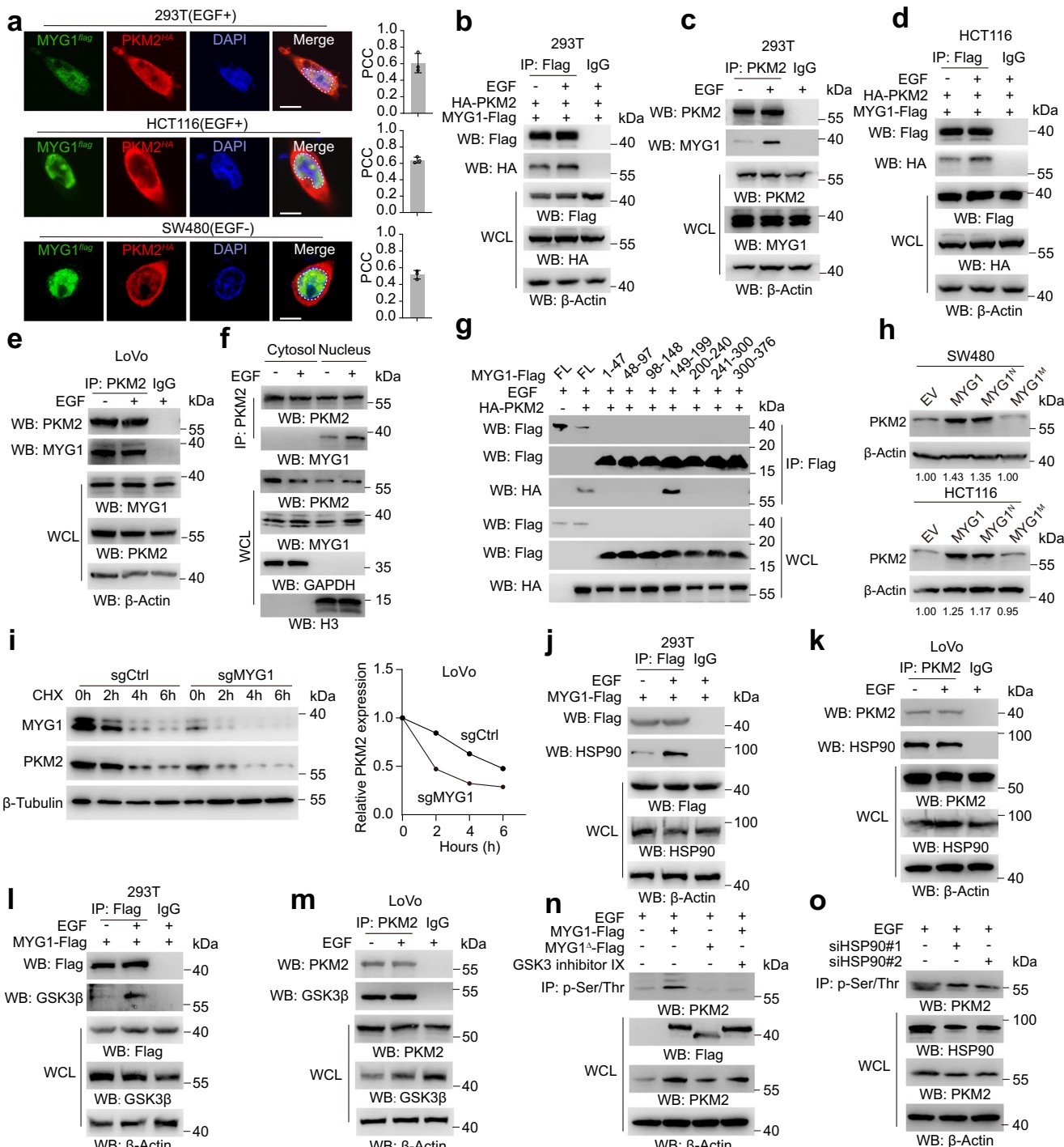

**Fig. 5 | Nuclear MYG1 recruits HSP90 to phosphorylate PKM2 and increase its stability. a** Representative images of the co-localization of MYG1 and PKM2 in 293 T, HCT116, and SW480 cells (left, scale bar, 10 μm), and Pearson's correlation coefficient (PCC) was analyzed in the ROI marked with dashed lines. 293 T and HCT116 cells were treated with EGF (100 ng/mL) for 10 h before fixing. Representative of 24 images from *n* = 3 independent experiments (left), and each point represents the average PCC of each experiment (right). 293 T (**b**) and HCT116 (**d**) cells were treated with EGF (100 ng/mL) for 10 h and lysed for immunoprecipitation. 293 T (**c**) and LoVo (**e**) cells were treated with EGF (100 ng/mL) for 10 h and lysed for immunoprecipitation. **f** 293 T cells treated with EGF (100 ng/mL) for 10 h were fractioned into cytosol and nucleus and then subjected to immunoprecipitation. **g** 293 T cells transfected with different MYG1 constructs were treated

with EGF (100 ng/mL) for 10 h and lysed for immunoprecipitation. **h** SW480 and HCT116 cells expressing MYG1 variants were detected for PKM2 protein level. **i** MYG1 KO LoVo cells treated with CHX (50 μg/mL) were harvested at 0, 2, 4, 6 h followed by detecting MYG1 and PKM2 protein levels by western blot (left) and quantified (right). **j–m** 293 T and LoVo cells treated as indicated were collected to detect the interaction of MYG1 with HSP90 (**j**), PKM2 with HSP90 (**k**), MYG1 with GSK3β (**l**), and PKM2 with GSK3β (**m**). **n** HCT116 cells transfected with MYG1 or MYG1$^Δ$ treated with EGF (100 ng/mL) or GSK3 inhibitor IX (10 μM) for 10 h were subjected to immunoprecipitation. **o** LoVo cells transfected with siRNAs of HSP90 (#1, #2) were treated and detected as in **n**. *n* = 3 independent experiments (**b–o**). See also Supplementary Fig. 6. Source data are provided as a Source Data file.

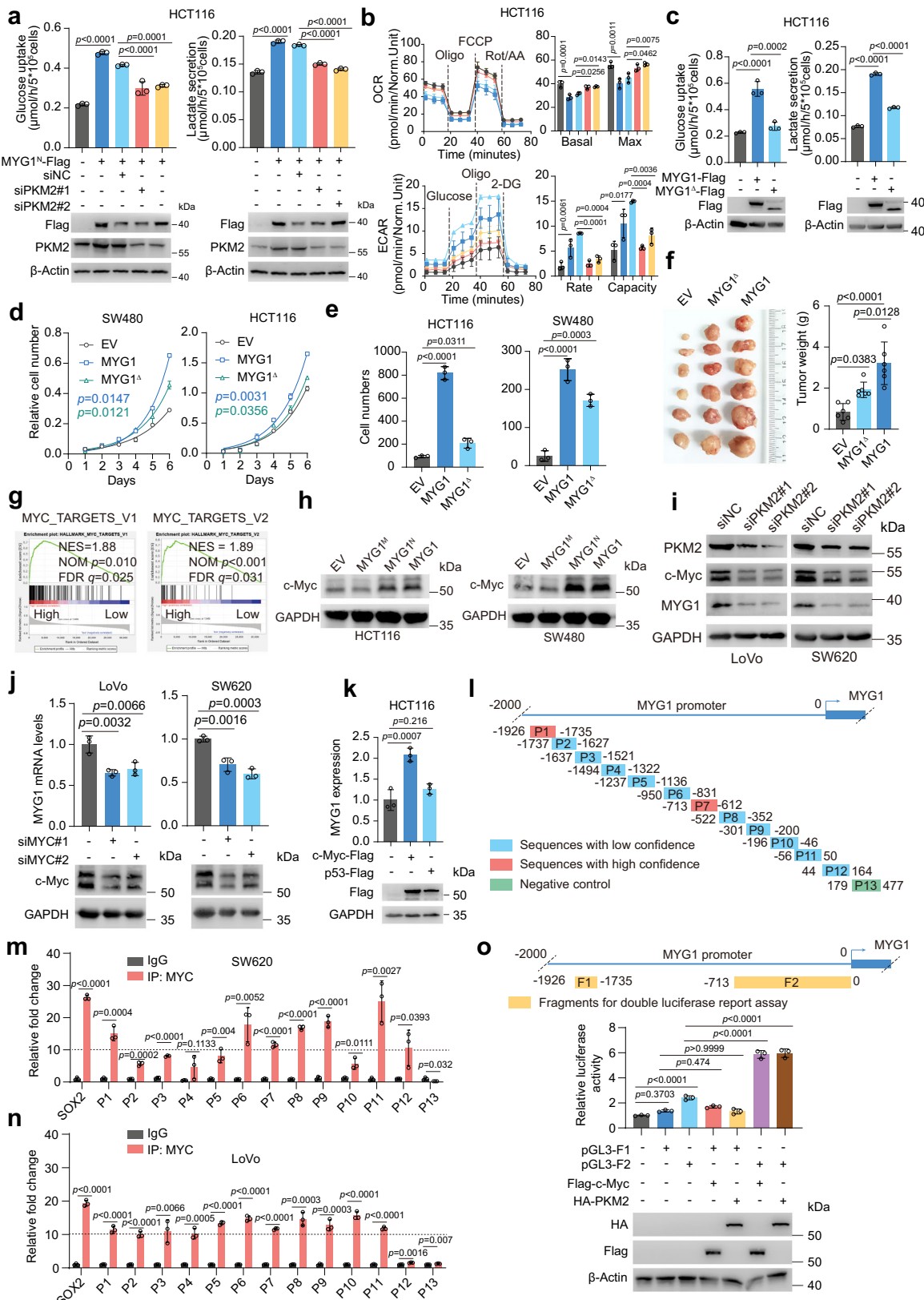

FHC and CRC cell lines. The results showed that PKM2 and c-Myc were highly expressed in CRC cell lines compared to FHC, and the expression of PKM2 and c-Myc was highly correlated with that of MYG1 (Supplementary Fig. 8g). We next knocked down PKM2 in SW620 and LoVo cells and detected c-Myc expression. The results showed that c-Myc was downregulated when PKM2 was knocked

down (Fig. 6i). Surprisingly, MYG1 was also downregulated when knocking down PKM2. This makes us consider whether c-Myc can regulate the expression of MYG1. Subsequently, we knocked down c-Myc in LoVo and SW620 cells, and the mRNA of MYG1 was downregulated (Fig. 6j). Overexpression of c-Myc instead of p53 in HCT116 cells increased the mRNA expression of MYG1 (Fig. 6k).

**Fig. 6 | Nuclear MYG1 accelerates glycolysis through the PKM2/c-Myc signaling pathway.** The PKM2 expression (**a**, bottom), glucose uptake and lactate secretion (**a**, top), OCR and ECAR (**b**) were detected. Norm.Unit represents the normalized OCR and ECAR (**b**). **c** Glucose uptake and lactate secretion was detected. **d**, **e** Proliferation and invasion ability were examined by proliferation assay (**d**, $n = 3$ technical replicates, representative data from $n = 3$ independent experiments) and transwell invasion assay (**e**). **f** Tumor growth was examined using subcutaneous xenograft tumor model ($n = 6$ in each group). The photograph (left) and weight (right) of tumors. **g** MYC TARGETS pathways were enriched in MYG1 highly expressed CRC patients (TCGA COADREAD cohort). **h** c-Myc expression was detected by western blot. The samples derive from the same experiment but different gels for c-Myc, and another for GAPDH were processed in parallel. **i** The expression of PKM2, c-Myc, MYG1, and GAPDH were detected by western blot. The samples derive from the same experiment but different gels for PKM2 and GAPDH, and another for c-Myc and MYG1 were processed in parallel. MYG1 mRNA was detected in cells with MYC koncked down (**j**) and MYC or TP53 overexpressed (**k**, the samples derive from the same experiment but different gels for Flag, and another for GAPDH were processed in parallel). **l**–**n** SW620 (**m**) and LoVo (**n**) cell lysate were subjected to ChIP-qPCR. Schematic of primer pairs for qPCR (**l**). P13 was designed as a negative control and SOX2 was set as the positive control. **o** Schematic of fragments used for double luciferase report assay (top). Transcriptional activity of luciferase was detected and normalized by Renilla luciferase activity in 293 T cells (bottom). $n = 3$ independent experiments (**a**–**c**, **e**, **h**–**k**, **m**–**o**). Unpaired two-sided Student's $t$-test (**m**–**n**). One-way ANOVA, Tukey's multiple comparisons test (**a**–**c**, **f**, and **o**), Dunnett's multiple comparisons test (**e**, and **j**–**k**). Two-way ANOVA, Tukey's multiple comparisons test (**d**). Kolmogorov–Smirnov test (**g**). $p$ value was provided in the figure. Error bars, mean ± SD. See also Supplementary Figs. 7 and 8. Source data are provided as a Source Data file.

These data indicated that c-Myc can regulate the expression of MYG1 conversely.

We next analyzed whether c-Myc could bind to MYG1 promoter using the online ChIP-seq datasets. The results showed an enrichment of a peak before the transcriptional start site (TTS) of MYG1 in several CRC cell lines (Supplementary Fig. 8h). To examine the possible binding site of c-Myc, we analyzed the 2000 bases before TTS of MYG1 on JASPR website. The results showed multiple loci exhibited the potential of binding to c-Myc (Fig. 6l). We then performed chromatin immunoprecipitation and qPCR (ChIP-qPCR) using SW620 and LoVo cells. As the result shows, c-Myc can bind to the MYG1 promoter at multiple sites, mainly at P1 and P6-P11 sequence (Fig. 6m–n and Supplementary Fig. 8i). However, whether c-Myc transcriptionally regulates MYG1 expression was unclear. We next performed the double luciferase reporter gene experiment to examine the regulatory sequences. Fragment 1 (F1, from -1926 to -1735 bases) and fragment 2 (F2, from -713 to 0 bases) from MYG1 promoter were used to validate the transcriptional regulation activity and F2 was regulated by c-Myc and PKM2 significantly (Fig. 6o). Collectively, these results indicated that MYG1 promotes glycolysis through PKM2/c-Myc signaling and c-Myc transcriptionally regulates MYG1 expression.

**Mitochondrial MYG1 inhibits CRC cell apoptosis and OXPHOS**

In addition to MYG1[N], MYG1[M] also promoted CRC progression in vitro, although to a lesser extent (Supplementary Fig. 5). We next explored the mechanism underlying this function. Among 184 candidate proteins that may interact with MYG1 in the mitochondria (Supplementary Fig. 6a and Supplementary Data 3), we focused on Cyt c, a key protein in mitochondrial oxidative respiratory chain, which can also regulate apoptosis and ROS balance in cells[30]. As MYG1 may be related to mitochondrial metabolism and apoptosis in CRC (Fig. 3a and Supplementary Data 1), we wondered whether MYG1 could interact with Cyt c and regulate apoptosis and OXPHOS. Firstly, we observed the co-localization of MYG1 and Cyt c in the mitochondria (Fig. 7a). The interaction of MYG1 and Cyt c in SW480 cells overexpressing MYG1 was confirmed by Co-IP (Fig. 7b). In addition, the mitochondria of SW480 cells overexpressing MYG1 was extracted to further validate the interaction of MYG1 and Cyt c in the mitochondria (Fig. 7c). Moreover, we also detected the co-localization of MYG1 and Cyt c in tumor tissues from CRC patients (Fig. 7d). All these results indicate the interaction of MYG1 and Cyt c in CRC cells. To investigate whether MYG1 affects Cyt c expression, we overexpressed MYG1 mutant as well as wild-type and found that Cyt c was downregulated in cells expressing MYG1[M] and wild-type (Fig. 7e). To examine whether OXPHOS was affected in the mitochondria, we detected the OCR in SW480 and HCT116 cells expressing MYG1[M]. The results showed that MYG1[M] inhibited the OCR of CRC cells (Fig. 7f), suggesting that the OXPHOS of cells was inhibited. The release of Cyt c from mitochondria is the key step of apoptosis. Subsequently, we also determined whether MYG1[M] could affect the release of Cyt c and apoptosis. Western blot showed that MYG1[M] overexpression inhibited the release of Cyt c from the mitochondria into the cytoplasm in CRC cells (Fig. 7g) and reduced the cleaved caspase-3 and caspase-9 (Fig. 7h). Flow cytometry assay also demonstrated that MYG1[M] inhibited the apoptosis of CRC cells (Fig. 7i). Together, these results shows that mitochondrial MYG1 interferes the OXPHOS and blocks the release of Cyt c from the mitochondria, inhibiting the apoptosis, in CRC cells.

**MYG1 indicates high glycolysis levels−clinical samples and in vivo correlation studies**

As the oncogenic protein MYG1 promoted glycolysis in CRC cells and the mouse model, whether it can indicate high glucose uptake in CRC patients remains unclear. We next detected the expression of MYG1, PKM2, and c-Myc in a NF-PET cohort including 43 CRC patients who underwent [18]F-FDG PET/CT scanning and did not receive other treatment before scanning and surgery. Our results showed that the maximum [18]F-FDG uptake value ($SUV_{max}$) of patients was positively correlated with the expression of MYG1 (Fig. 8a–b). Besides, the expression of MYG1 was also correlated with that of PKM2 and c-Myc (Fig. 8b). In these patients, MYG1, PKM2, and c-Myc were all highly expressed in tumor tissues compared to normal mucosa (Fig. 8c). These results indicate that MYG1 is a reliable indicator of glycolysis in CRC patients. Similarly, we also analyzed the expression of PKM2, c-Myc and Ki-67 in mouse model tumors. As expected, MYG1 variants were overexpressed in the corresponding cellular compartment (Fig. 8d). Besides, MYG1[N] tumors showed higher expression of PKM2 and c-Myc, as well as a higher proliferation index (Fig. 8d–e). TdT-mediated dUTP Nick-End Labeling (TUNEL) technique was used to study apoptosis in situ of mice tumors and MYG1[N] tumors showed a lower percentage of apoptotic cells compared with wild-type and nuclear-located counterparts (Fig. 8d–e). Taken together, our study identifies the oncogenic protein MYG1 and reveals that both nuclear and mitochondrial MYG1 cooperatively promote metabolic remodeling and tumor progression of CRC (Fig. 8f).

## Discussion

Here, we report a novel oncogenic factor MYG1 that drives CRC progression and promotes glycolysis by coordinating nuclear and mitochondrial function. High MYG1 expression in CRC was observed in several cohorts and was associated with advanced stage and poor outcomes of patients. MYG1 promotes CRC proliferation and metastasis in vitro and in vivo. Nuclear MYG1 promotes glycolysis through the PKM2/c-Myc signaling, while mitochondrial MYG1 inhibits apoptosis and OXPHOS. Patients with high [18]F-FDG uptake tend to exhibit high MYG1 levels. These functions of MYG1 might be very important not only for understanding the vulnerability of CRC metabolic reprogramming but also for intervention in metabolic remodeling and providing a novel therapeutic target as well.

In addition to the proliferation and metastatic ability of MYG1, we also discussed the driver function of MYG1 in CRC. Activating

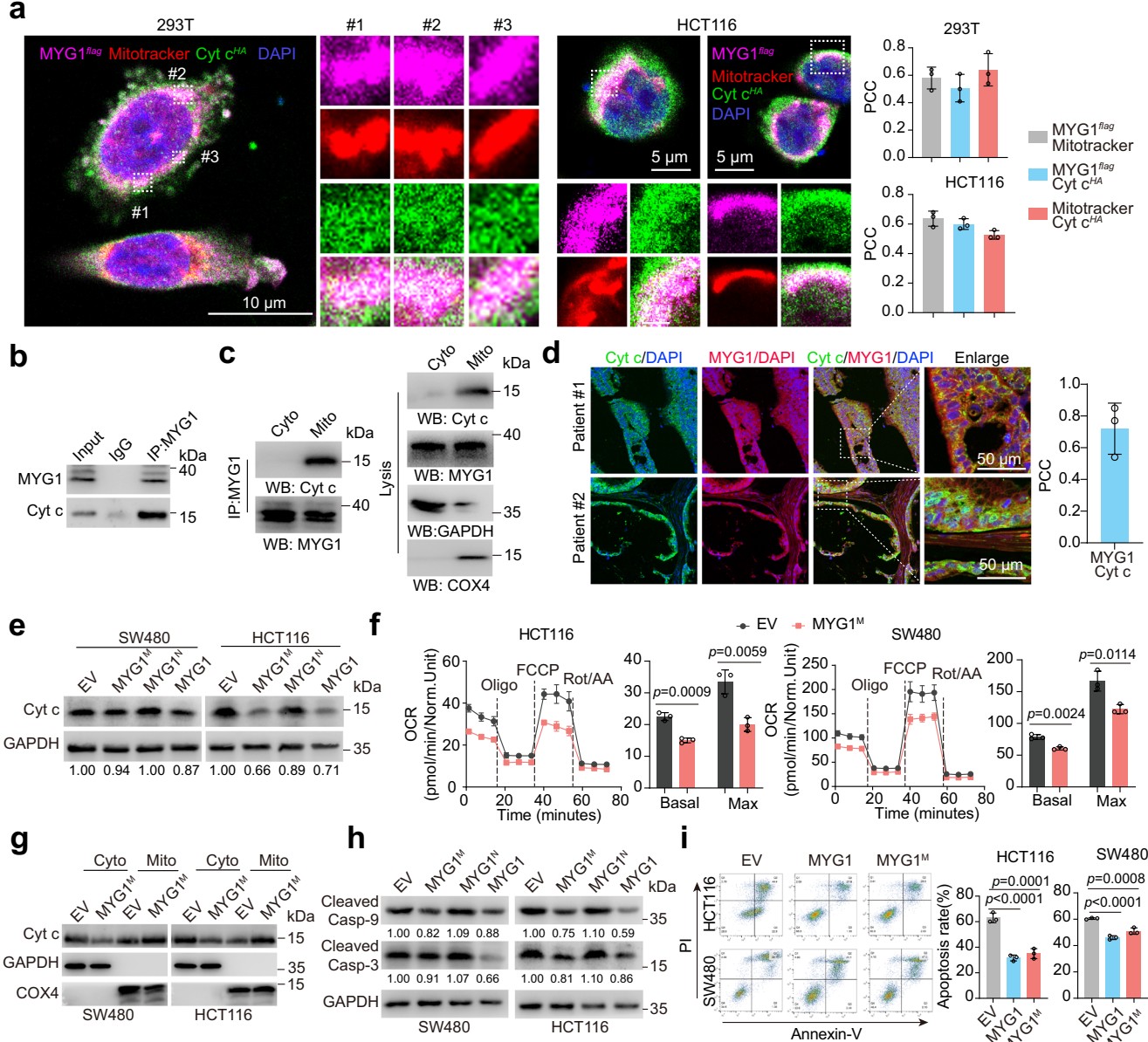

**Fig. 7 | Mitochondrial MYG1 inhibits OXPHOS and apoptosis in CRC cells. a** Co-localization of MYG1 and Cyt c in 293 T and HCT116 cells was detected by IF. Mitotracker was used as an indicator of mitochondria. Scale bar, 10 μm (293 T) and 5 μm (HCT116). Representative of 13 images for 293 T and 9 images for HCT116 from $n = 3$ independent experiments. PCC of at least two ROIs from each image were analyzed and averaged. Dashed box displayed the selected ROI in the images (left). Graphs represent PCC of pairwise co-localization analysis of $n = 3$ independent experiments (right). **b, c** SW480 cells stably overexpressing MYG1 were used for immunoprecipitation in total cell lysates (**b**), cytoplasm (Cyto) and mitochondrial (Mito) (**c**, the lysis samples derive from the same experiment but different gels for Cyt c and GAPDH, and another for MYG1 and COX4 were processed in parallel). **d** Representative images of tumor tissues from $n = 3$ CRC patients and graphs represent PCC of MYG1 and Cyt c co-localization analysis from $n = 3$ patients (right).

Scale bar, 50 μm. **e** The expression of Cyt c was detected by western blot. **f** OCR were detected in control and MYG1$^M$ expressed SW480 and HCT116 cells. **g** Distribution of Cyt c was detected in cytoplasm (Cyto) and mitochondrial (Mito) lysate by western blot. The samples derive from the same experiment but different gels for Cyt c, and another for COX4 and GAPDH were processed in parallel. **h** Cleaved Caspase 3 and Cleaved Caspase 9 expression were detected by western blot after treatment of 5-Fu (50 μM for 72 h). The samples derive from the same experiment but different gels for Cleaved Caspase 9 and GAPDH, and another for Cleaved Caspase 3 were processed in parallel. **i** Apoptosis was analyzed by flow cytometry after treatment of 5-Fu (50 μM for 72 h). $n = 3$ independent experiments (**b–c** and **e–i**). Unpaired two-sided Student's $t$-test (**f**). One-way ANOVA, Dunnett's multiple comparisons test (**i**). $p$ value was provided in the figure. Error bars, mean ± SD. Source data are provided as a Source Data file.

mutations in KRAS often arise after mutations in APC and are found in nearly 40% of CRC tumors[31]. The increase of MYG1 is associated with KRAS mutation. This is the early event in CRC tumorigenesis. Since KRAS mutation can induce increased MYC expression, we addressed a possible role for MYC in driving MYG1 expression. This hypothesis was confirmed by our study. Nuclear distribution of PKM2 was increased in patients with KRAS, increasing the interaction of

MYG1 and PKM2 in the nucleus and activating the c-Myc signaling and glycolysis. In summary, our results indicate that the elevated expression of MYG1 in CRC can be induced by c-Myc activation especially in patients with KRAS mutation, and more convincing evidence still needs further confirmation by genetically engineered mouse models. In addition, some functional studies of MYG1 were conducted using a single shRNA or sgRNA, which suggests

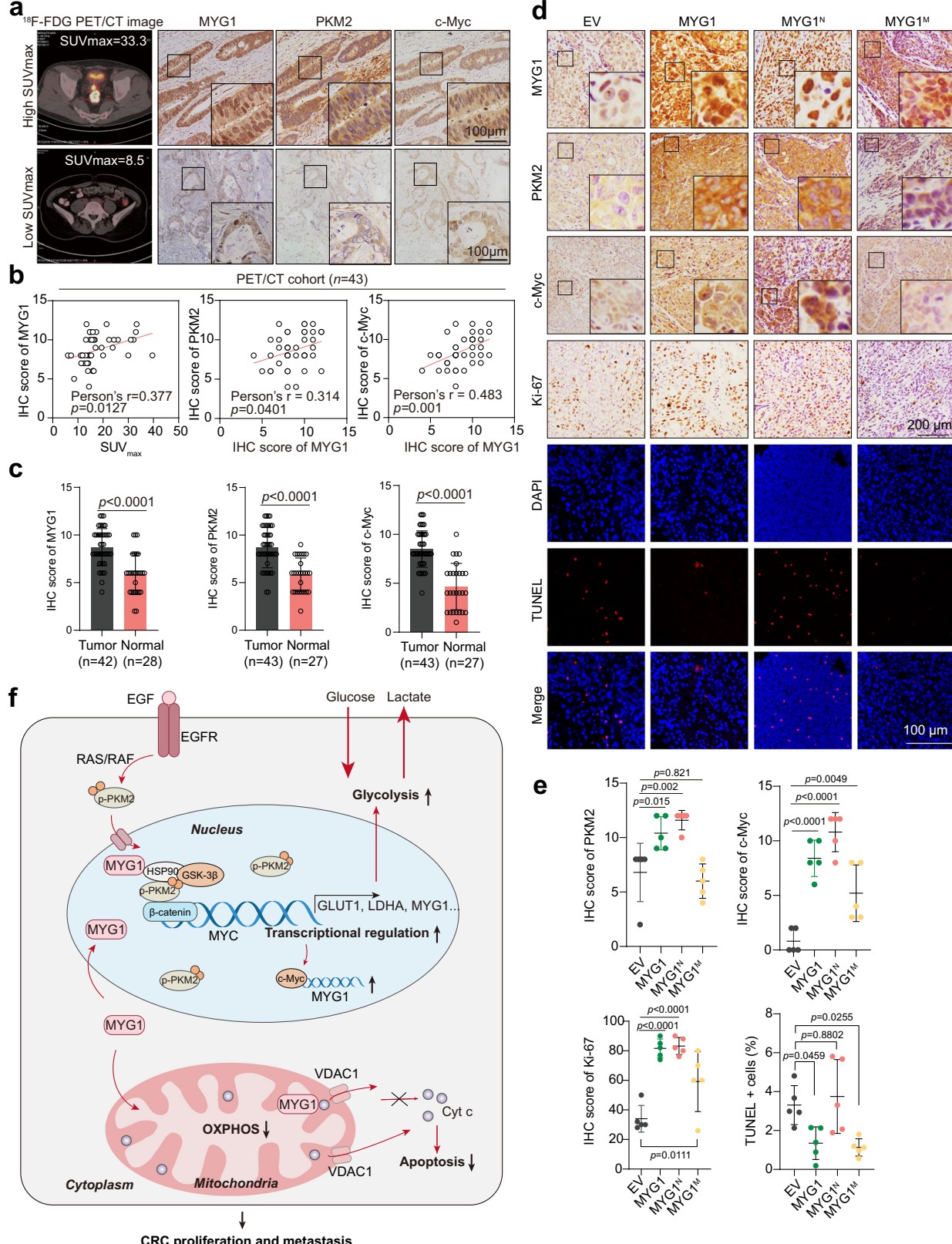

potential off-target effects. Future studies will evaluate and mitigate these risks.

It has been reported that MYG1 is highly conserved, and knockout of MYG1 in yeast showed defects in respiratory growth. Knockout any one of the localization peptides or DHH domain limits oxygen consumption[5]. Our findings reveal novel functions of MYG1 in CRC

independent of its exonuclease activity. Highly expressed MYG1 induces OXPHOS damage and strengthens glycolysis in CRC, and differently localized MYG1 showed a cooperated function. Our study reveals a mechanism of MYG1, especially in KRAS-mutant CRC patients where MYG1 is upregulated. MYG1, in a manner independent of exo-nuclease activity, promotes tumor glycolysis and progression through

**Fig. 8 | MYG1 correlates with an active glycolysis pathway in CRC patients and tumor model. a–c** MYG1, PKM2 and c-Myc protein levels were evaluated by IHC staining in specimens of CRC patients from NF-PET cohort ($n = 43$) and followed by quantification. Representative images of PET/CT and IHC staining (**a**). The areas marked by squares were magnified. Scale bar, 100 μm (insets). The correlations between the level of MYG1 and $SUV_{max}$, the level of PKM2 and c-Myc were analyzed (**b**). The protein levels of MYG1, PKM2, c-Myc in tumor and normal mucosa (**c**). **d**, **e** Expression of MYG1, PKM2, c-Myc, Ki-67 in tumor tissues from orthotopic CRC models were evaluated by IHC and apoptosis evaluated by TUNEL staining (**d**), and quantified (**e**) ($n = 5$ per group). The areas marked by squares were magnified. Scale bar, 200 μm (in IHC), and 100 μm (in IF). **f** Work model for MYG1 driving glycolysis and CRC development. Two-sided Pearson correlation (**b**). Unpaired two-sided Student's $t$-test (**c**). One-way ANOVA, Dunnett's multiple comparisons test (**e**). $p$ value was provided in the figure. Representative results were shown (**a** and **d**). Error bars, mean ± SD. Source data are provided as a Source Data file.

nuclear and mitochondrial functions. This remodeling of CRC metabolism enhances proliferation and invasive capabilities. Detection of glycolytic phenotype and tumor biology functions reveals the dominant role of nuclear MYG1, while mitochondrial MYG1 also demonstrates functionality despite its weaker effect. However, since our focus was on the mechanisms of nuclear MYG1, we only investigated the function of mitochondrial MYG1 on OXPHOS and apoptosis. Mitochondrial MYG1 interacts with Cyt c and regulates its expression. It remains unclear whether the downregulation of Cyt c is dependent on the binding of MYG1 and how this affects the degradation of Cyt c.

The functional investigation of MYG1 is not well established, and recent studies have gradually revealed its role in mitochondrial function. Particularly, research on the role of MYG1 in tumors is still in its early stages, and its functions necessitate extensive investigations. Our study found that MYG1 binds to PKM2 and maintains its stability, and we attempted to verify the importance of MYG1-PKM2 binding in glycolysis and CRC tumor progression by deleting the PKM2 binding site of MYG1. While our findings demonstrate that MYG1$^\Delta$, which lacks the PKM2 binding site, partially reverses glycolysis and tumor progression compared to wild-type MYG1, it is important to acknowledge that there may be other functional abnormalities associated with MYG1$^\Delta$, such as exonuclease activity or mitochondrial function. Hence, the observed effects cannot definitively exclude the potential impact of these alternative functions.

The research of MYG1 in LUAD suggests that it plays a promoting role in tumor progression, consistent with its role in CRC. However, there are some discrepancies between the function in CRC and LUAD in terms of metabolic regulation. Through preliminary experiments, we have discovered that MYG1 is associated with KRAS mutations in CRC but not in LUAD. KRAS mutation drives metabolism reprogramming and enhances glycolysis of CRC cells, while recent studies have shown the opposite in lung cancer. Lung cancer cells tend to rely on OXPHOS to supply energy for tumor progression[32,33]. Therefore, we speculate that identifying the key factors regulating MYG1 in different cancers may help unravel its role in different tumors. Considering MYG1's oncogenic role in tumors, it may serve as a potential target for cancer treatment. However, due to its varying roles in different types of tumors, it is necessary to study the mechanisms underlying its function across different cancers. This would enable the identification of more suitable patient populations for targeted interventions and provide valuable insights for precision therapy. In summary, our study revealed a key driver of CRC, MYG1, and highlighted the potential of therapeutics targeting MYG1 in CRC, which deserves further verification as a prospective therapeutic strategy.

## Methods

### Ethics statement
All human CRC patient samples used in this study were collected from Nanfang Hospital of Southern Medical University (Guangzhou, China). The protocols were approved by the Ethics Committee of Nanfang Hospital of Southern Medical University. The patients were informed of the study, and signed informed consent forms were obtained. The acquisition and publication of patient clinical information have been authorized. All animal experiments were approved and performed in accordance with the guidelines of the Institutional Animal Care and Use Committee (IACUC) at the Southern Medical University in Guangzhou.

### Human tissue samples
Thirty-one paired fresh CRC tissues and adjacent normal mucosa were collected and stored in liquid nitrogen to analyze the mRNA levels of MYG1 via qPCR. Eight paired fresh CRC tissues and adjacent normal mucosa were collected and stored in liquid nitrogen for analyzing protein expression of MYG1 by western blot. A total of 120 paired FFPE tissues with the prognosis and clinical information of patients were collected (NF-CRC1 cohort) to analyze the relationship between the expression of MYG1 and the clinical characteristics of patients. FFPE tissues from NF-CRC2 cohort including normal mucosa, adenoma, intramucosal carcinoma, and distant metastasis were collected to analyze the expression of MYG1 in the progression of CRC. FFPE tissues from NF-KRAS cohort including CRC tissues with or without KRAS mutation were collected to analyze the relationship between the expression of MYG1 and KRAS status. Forty-four FFPE tissues from CRC patients who underwent PET/CT scanning before surgery were collected to analyze the uptake of $^{18}$F-FDG and IHC staining. All samples were collected from patients who did not receive adjuvant therapy before surgery. The details of patients' information were provided in the Source Data.

### Cell lines and reagents
The human normal colonic epithelial cell FHC (CRL-1831) and CRC cell lines HCT116 (ATCC CCL-247), HT-29 (ATCC HTB-38), SW480 (ATCC CCL-228), SW620 (ATCC CCL-227), RKO (ATCC CRL-2577), LoVo (ATCC CCL-229), DLD1 (ATCC, CCL-221), and CACO2 (ATCC HTB-37), as well as 293 T (ATCC CRL-3216), were purchased from the American Type Culture Collection (ATCC; http://www.atcc.org/). PC-9 cells were from Cell Bank/Stem Cell Bank, Chinese Academy of Sciences (http://www.cellbank.org.cn/). All cells were grown in Dulbecco's Modified Eagle Medium (DMEM) supplemented with 10% fetal bovine serum (Gibco, USA) except for CACO2 and FHC, which was cultured in DMEM with 20% fetal bovine serum. Cells were maintained at 37 °C in a humidified atmosphere containing 5% $CO_2$. All cell lines were authenticated by STR profiling and tested for mycoplasma contamination. Cells were cultured in DMEM medium (Gibco) supplemented with recombinant human EGF (100 ng/mL, MCE) in specific assays. PKM2 inhibitor (compound 3k, C3K) (Selleck, S8616) and GSK 3 Inhibitor IX (MCE, HY-10580) were used in specific assays.

### Plasmid, siRNA, lentivirus, and the transfection
C-terminally Flag-tagged human MYG1 (full length), MYG1$^N$ (del 1-20), MYG1$^M$ (del 33-39), MYG1$^{\Delta L}$ (40-376), MYG1$^{ALL}$, MYG1$^\Delta$ and seven truncated MYG1 fragments were cloned into double-digested pcDNA3.1(+) with KpnI and EcoRI. N-terminally HA-tagged human PKM and CYCS were cloned into double-digested pcDNA3.1(+) with BamHI and EcoRI. C-terminally Flag-tagged human MYC and TP53 were cloned into double-digested pcDNA3.1(+) with NheI and XhoI. Double-stranded oligonucleotides encoding the shRNA sequences of MYG1 were cloned into double-digested pLKO.1-puro luciferase shRNA vector with AgeI and EcoRI. SiRNAs used in this study were generated by RiboBio, China. The target sequences of shRNAs and siRNAs are listed

in Supplementary Table 2. Double-stranded oligonucleotides encoding the sgRNA sequences of MYG1 were cloned into BmsBI-digested plasmid LentiCRISPRv2 (deposited by F. Zhang of MIT to Addgene, Cambridge, MA). SgRNA sequences targeting human MYG1 were: TGGGGGGCGAGTACGACCCTCGG. Plasmids and siRNAs were transfected using Lipofectamine 3000 (Invitrogen) according to the manufacturer's protocol. Lentivirus vectors expressing MYG1 and MYG1 variants (pCDH-CMV-MCS-EF1-copGFP-T2A-Puro as plasmid backbone) and shRNA- or sgRNA-encoding lentivirus vectors were co-transfected with the packaging vectors psPAX2 (Addgene) and pMD2.G (Addgene) into 293 T cells for lentivirus production. Cells were infected with the above lentiviruses for up to 48 h. Starting from 72 h after infection, cells were screened with 2 μg/mL puromycin for at least 5 days. Lentivirus vectors expressing luciferase were transfected into HCT116 cells for in-vivo study.

### Examination of cell malignant phenotypes in vitro

Proliferation assay was carried out to evaluate the cell viability using a Cell Counting Kit-8 (GLPBIO, USA) followed with the manufacturer's instructions. Transwell invasion assay was carried out to evaluate the cell invasion ability. Cells were seeded at a density of $1 \times 10^5$ cells in the cell culture insert (BD Biosciences) with a pore size of 8 μm and cultured in a 24-well plate for in vitro migration assays. Matrigel matrix (Corning) was added to the cell culture insert before seeding cells. Cells were cultured in the insert with serum-free medium and 10% FBS was added to the 24-well plate as an inducer. Cells passing through the insert pores were counted after fixed and stained. Wound healing assay was carried out to evaluate the migration ability of cells. Cells were seeded in a 6-well plate and the confluent monolayers were wounded in a line across the plates with sterile 20 μL plastic pipette tips. To avoid the influence of cell proliferation, cells were cultured with serum-free medium. The area of migration was measured by ImageJ (NIH, USA, version 1.54 f). Colony formation assay was carried out to evaluate the colony formation ability. Cells were seeded at a density of 250 cells per well in 12-well plates. The colony was counted two weeks later after being fixed and stained.

### In vivo oncogenesis assays

The mice were purchased from the Experimental Animal Center of Southern Medical University. Four- to five-week-old BALB/c (nu/nu) nude mice were purchased and housed in a specific-pathogen-free condition with a dark/light cycle of 12-h of light/12-h of darkness, ambient temperature of 20–26 °C and humidity of 40–70%. Mice in each group were randomly assigned for the experiment. Only female mice were used in the experiments to ensure the reproducibility of tumor kinetics and growth (without gender bias). A subcutaneous xenograft model was established to evaluate tumor growth. Cells ($1 \times 10^6$) were subcutaneously injected into the dorsal flanks of the mice. Tumors were measured once a week, and tumor volume was evaluated using the following formula: V = (shortest diameter)$^2$ × (longest diameter) × 0.5. After 3 weeks, the mice were euthanized and the tumors were separated for IHC and H&E staining. A liver metastasis model was established to evaluate liver metastasis. Four- to five-week-old BALB/c (nu/nu) nude mice underwent surgery under anesthesia. Cells ($1 \times 10^6$) were injected into the splenic capsules of mice. After 3 weeks, the mice were euthanized and the liver and spleen were separated for IHC and H&E staining. According to the approved animal protocol, the maximum diameter of all tumors is less than 15 mm, and all tumors in the experiment did not exceed the limit. An orthotopic CRC mouse model was established to evaluate the uptake of $^{18}$F-FDG in vivo and the progression of CRC. The model was established by transplanting the subcutaneous tumor onto the cecum according to a published protocol[34]. Four weeks after surgery, tumor growth was monitored using an in-vivo imaging system (Bruker, USA) by intraperitoneal injection of D-fluorescein (Promega,

USA). The uptake of $^{18}$F-FDG was measured 30 days after surgery. Mice were fasted for 8 h and injected with approximately $4.5 \pm 0.5$ MBq of $^{18}$F-FDG via lateral tail vein (the exact dose was calculated by measuring the syringe before and after injection). After injection, the mice were maintained in cages at RT for 40 min and then anesthetized with isoflurane. Next, the mice were placed on the pad in the prone position, followed by micro-PET and micro-CT imaging (Inviscan, France). $^{18}$F-FDG-uptake rate was determined in the light of the following formula: (activity in tumor in Bq)/(injected activity in Bq)/(mouse weight in cm$^3$) in order to adjust the injected and metabolic activity changes between inspections and to obtain tumor-specific uptake. The SUV$_{max}$ was quantified by drawing the region of the tumor using IRIS PET/CT software. All mice were sacrificed 6 weeks after surgery. If there was a significant decrease in mouse weight or other terminative indicator reported in the experimental protocol during the experiment, the mice were euthanized timely. During the experiments, the investigator was blinded to the group allocation when assessing the outcome.

### Western blot

Western blot was performed to detect the protein expression. Cell or tissue samples were lysed with RIPA buffer containing freshly added complete protease inhibitor cocktail (Roche) and PMSF (LEAGENE). Protein concentrations were measured using a BCA protein assay kit (EpiZyme) according to the manufacturer's instructions. Proteins were separated by sodium dodecyl sulfate polyacrylamide gel electrophoresis (SDS-PAGE) and transferred onto a 0.22 μm polyvinylidene difluoride (PVDF) membrane (Merck Millipore). The membranes were blocked with 10% skim milk dissolved in PBST for 1 h at RT. The membranes were incubated with primary antibodies overnight at 4 °C. After washing thrice with PBST, the membranes were incubated with secondary antibodies for 1 h at RT. Finally, the membranes were incubated with FDbio-Femto ECL substrate and scanned with the Tanon 5200 Multi System. Antibodies: MYG1 (C12orf10) (Abcam, 1:1000, ab204420), GAPDH (Proteintech, 1:5000, 60004-1-Ig), β-Actin (Proteintech, 1:5000, 66009-1-Ig), β-Tubulin (Proteintech, 1:5000, 10068-1-AP), COX4 (Proteintech, 1:1000, 11242-1-AP), HA tag (Proteintech, 1:1000, 51064-2-AP), Flag tag (Cell Signaling, 1:1000, 8146), PKM2 (Proteintech, 1:1000, 60268-1-Ig), c-Myc (Proteintech, 1:1000, 10828-1-AP), ZO-1 (Cell Signaling, 1:1000, 13663), E-Cadherin (Cell Signaling, 1:1000, 14472), β-Catenin (Cell Signaling, 1:1000, 8480), Vimentin (Cell Signaling, 1:1000, 46173), LDHA (HUABIO, 1:1000, ET1608-57), GLUT1 (ABclonal, 1:500, A6982), H3 (ABclonal, 1:500, A2348), VDAC1 (Proteintech, 1:1000, 55259-1-AP), TIMM23 (HUABIO, 1:1000, HA500361), HSP90 (Proteintech, 1:2000, 13171-1-AP), GSK3β (Abmart, 1:1000, T40069), Cyt c (Proteintech, 1:1000, 10993-1-AP), Caspase-3 (Cell Signaling, 1:1000, 9662), Caspase-9 (Cell Signaling, 1:1000, 9502), MRPS27 (HUABIO, 1:1000, ER64052). All results are derived from at least three independent biological replicates, and representative results are shown. Protein levels were quantified by densitometry using ImageJ software. The original data of western blot was supplied in the Source Data file.

### Immunohistochemistry (IHC)

IHC was performed to evaluate the expression of proteins in tissues. FFPE tissues underwent dewaxing, antigen repair and blocking before incubating with primary antibodies overnight at 4 °C. After being washed with PBST three times, slides were incubated with HRP-conjugated secondary antibodies for 1 h at RT. The slides were incubated with DAB chromogenic Kit (ZSGB-Bio). The slides were stained with hematoxylin, dehydrated and sealed for observation and scanning. Antibodies: MYG1 (C12orf10) (Abcam, 1:100, ab204420), Ki-67 (ZSGB-BIO, working solution, ZM-0167), GLUT1 (ABclonal, 1:100, A6982), PKM2 (Abcam, 1:100, ab85555), c-Myc (Proteintech, 1:100,

10828-1-AP). The scoring was conducted according to the standard of 12-point scoring by three pathologists independently.

## Immunofluorescence (IF)

Immunofluorescence was carried out to assay the location and expression of proteins in cells and tissues. For cells, we cultured cells in a confocal dish and fixed cells with 4% paraformaldehyde. Cells were then washed with cold PBS buffer and treated with 0.5% Triton X-100 for 20 min. After blocked with 10% goat serum for 30 min at RT, cells were incubated with antibody at 4 °C overnight. The next day, cells were washed with PBST and incubated with secondary antibodies with fluorescence conjugated for 1 h at 37 °C. After washing with PBST, nucleus of cells was stained by DAPI for 10 min at RT. Last, cells were mounted with glycerin and observed under a confocal microscope (FV3000, Olympus). For mitochondria labeling, Mitotracker (Invitrogen, working concentrations of 25–500 nM) was incubated with cells at 37 °C for 30 min before IF. For tissue samples, the procedure before secondary antibody incubation was similar to that of IHC. The subsequent procedure was the same as that in cells. Antibodies: E-Cadherin (ABclonal, 1:500, A22850), Fibronectin (ABclonal, 1:100, A12977), Flag tag (Cell Signaling, 1:800, 8146), HA tag (Proteintech, 1:100, 51064-2-AP), PKM2 (Proteintech, 1:200, 60268-1-Ig), MYG1 (C12orf10) (Abcam, 1:100, ab204420), Cyt c (Proteintech, 1:100, 10993-1-AP). The co-localization and fluorescence intensity analysis were quantified using ImageJ. For immunofluorescence quantitative analysis, the Laser power and voltage remain unchanged during capturing the pictures. For co-localization analysis, at least two regions of interest (ROIs) are selected for each image, and ImageJ is used for analysis. Pearson correlation coefficient (PCC) of two channels was calculated and averaged across all images in each independent experiment[35].

## Immunoprecipitation (IP)

Cells were lysed with lysis buffer (25 mM Tris-HCl, pH = 7.4; 150 mM NaCl; 1 mM EDTA; 1% NP-40; 5% glycerol) containing freshly added complete protease inhibitor cocktail (Roche) and PMSF (LEAGENE). After lysis for 10 min on ice, the cells were centrifuged at 4 °C and 14,000 rpm for 30 min. The supernatant was collected in a new tube and 1% supernatant was used as the input. Antibodies were incubated with Protein A/G magnetic beads (Selleck) for 15 min at RT. Beads were washed by lysis buffer and then incubated with cell lysate for 8 to 12 h at 4 °C. The beads were washed three times with lysis buffer and denatured by adding loading buffer. Samples were separated by SDS-PAGE and analyzed by LC/MS or western blot. Antibodies: MYG1 (C12orf10) (Abcam, ab204420), Flag tag (Cell Signaling, 8146), HA tag (Proteintech, 51064-2-AP), PKM2 (Proteintech, 60268-1-Ig), p-Ser/Thr (Abcolonal, AP0893).

## Chromatin immunoprecipitation and qPCR (ChIP-qPCR)

ChIP assays were performed using a ChIP Kit (Abcam, ab500), following the manufacturer's instructions. Briefly, cells were cultured and fixed with 1% formaldehyde to cross-link histone and non-histone proteins to DNA at RT for 10 min and quenched with glycine. Chromatin was digested and sonicated into 150-900 bp DNA/protein fragments. Antibody specific to c-Myc (10828-1-AP, Proteintech) was used for immunoprecipitation and the co-precipitates complex was captured using Protein A/G beads. Finally, the cross-links were reversed, and target DNA fragments were purified by DNA purifying slurry. One-tenth of the input chromatin was also treated in the same way and purified. The binding of the MYG1 promoter to c-Myc, H3 or IgG was quantified using quantitative PCR with primers and PCR products were subjected to agarose gel electrophoresis. ChIP primer sequences are listed in Supplementary Table 2. The percentage of enriched DNA fragments in the input indicated the degree of enrichment. Enrichment

of the IP more than ten times that of the IgG was considered a positive signal.

## Glucose metabolism analysis

The OCR was measured using a Seahorse XFe 96 Extracellular Flux Analyzer with an Agilent Seahorse XF Cell Mito Stress Test Kit (Agilent, 103015-100). In brief, CRC cells were seeded into the XF96-well culture plates and incubated at 37 °C overnight for detection. Mitochondrial stress was assessed using oligomycin, FCCP, and rotenone & antimycin A. The experiments were performed according to the introduction of a manual. The ECAR was monitored based on the XF Glycolysis Stress Test kit (Agilent, 103020-100) protocol using glucose, oligomycin, and 2-DG. The Seahorse Wave software was used to analyze the data. The cells in each well were digested and counted to normalize the results after detection. The non-mitochondrial OCR and non-glycolytic acidification were subtracted when performing the quantification.

For glucose uptake and lactate secretion assays, cells ($5 \times 10^5$) were seeded in a 96-well plate and cultured overnight in serum-free medium. The following day, the medium was discarded and cells were incubated with PBS for 40 min to induce starvation. Then, 100 μL of DMEM medium containing 10% FBS was added and incubated at 37 °C for 1 h. Subsequently, the supernatant from the cell culture was collected, and Glucose Assay Kit (Abbkin) and Lactate Assay Kit (Abbkin) were used to measure the glucose and lactate levels in the supernatant, respectively. Detection was followed the operating instructions. The initial glucose and lactate levels in the medium of both blank controls and samples were simultaneously measured for comparison in order to calculate the glucose uptake and lactate secretion. All experiments were performed at least three times.

## Nucleocytoplasmic separation and mitochondrial separation

The nucleocytoplasmic separation experiment was carried out according to the introduction of the Mammalian Nuclear and Cytoplasmic Protein Extraction Kit (TRAN, DE201-01). For crude mitochondria isolation, cells ($1 \times 10^7$) were collected and pelleted at 4 °C and 1000 g for 15 min. Cells were then resuspended in 500 μL ice-cold CHM buffer (150 mM MgCl$_2$; 10 mM KCl; 25 mM Tris HCl, pH = 6.7; 1 mM EDTA). After leaving on ice for 2 min, cells were homogenized with syringe until more than 90% cells were broken. Add 200 μL ice-cold CHM containing 1 M sucrose and mix gently by repeated inversion. Nuclei were pelleted at 4 °C and 1000 g for 10 min. Supernatant was collected and centrifuged at 4 °C and 10,000 g for 10 min. The pellet was resuspended and washed by ice-cold mitochondrial suspension medium (0.25 M sucrose; 25 mM Tris base; adjust pH to 7.0 with acetic acid) for further analyses. For purified mitochondria isolation, sucrose gradient sedimentation was performed according to the protocol[36].

## Protease K shaving assay

Mitochondria from LoVo and 293 T cells were obtained according to the procedure above. Mitochondria were incubated with 280 μg/mL protease K (Beyotime) for 30 min on ice with or without 1% Triton X-100. The reactions were terminated by adding 1 mM PMSF. Mitochondrial lysate was then subjected to western blot.

## Flow cytometry (FACS) analysis

Flow cytometry was performed to evaluate the apoptosis rate and cell cycle of cells. The cells were treated with 5-Fu (50 μM) for 72 h to induce apoptosis. The cells ($1 \times 10^5$) were collected and stained with the Annexin V, FITC Apoptosis Detection Kit (Dojindo, AD10). The cell cycle was also detected by flow cytometry using the Cell Cycle Staining Kit (Multi Sciences). The cells were analyzed in LSRFortessa X-20 Cell Analyzer (BD Biosciences). Data were analyzed using FlowJo software (version 10.6.2).

## Pyruvate kinase activity detection

Pyruvate kinase activity in CRC cells was measured using the Pyruvate Kinase Assay Kit (Abbkine, KTB1120) according to the manufacturer's instructions. Cells ($5 \times 10^5$) were collected and lysed. Supernatant was used for detection. The results were normalized by protein concentration of supernatant.

## PKM2 cross-linking assay

Cross-linking experiments of PKM2 were performed following the previous report[37]. Samples were separated by SDS/PAGE and analyzed by western blot and the expression of β-Actin was set as the internal control.

## Total RNA extract, reverse transcription (RT) and real-time fluorescence quantitative polymerase chain reaction (qPCR)

Total RNA of cells or tissues was extracted using TRIZOL reagent (AG21101). Reverse transcription was performed following the description of the RT kit (AG11706). QPCR was performed using the SYBR Green chimeric fluorescence method following the description of the kit (AG11718). Indicated genes were detected by specific primer pairs on the generated cDNA using the 7500 Fast Real-Time PCR System (ThermoFisher). GAPDH or ACTB was used as the internal reference. The results are expressed as mean ± SD. $2^{-\Delta\Delta Ct}$ method was applied to analyze the relative expression of genes. Primer sequences used in our study are listed in Supplementary Table 2.

## Immuno-electron microscopy

Immuno-electron microscopy was performed by fixing, embedding, and immunolabeling. In brief, cells transiently expressing MYG1-Flag were fixed by 0.5% glutaraldehyde (GA) and 4% paraformaldehyde (PFA) for 4 h. After dehydration, cells were embedded in resin and followed by UV polymerization for 4 days at −20 °C. Ultrathin sections were cut and immunolabeled with anti-Flag (Cell Signaling, 1:100, 8146) and colloidal gold secondary antibody. After uranium lead staining, samples were viewed on a transmission electron microscope (JEM-1400, JEOL) operating at suitable acceleration voltages (80 kV).

## Dual-Luciferase reporter assay

Fragment 1 (F1) and fragment 2 (F2) of MYG1 promoter were cloned into double-digested pGL3-Basic with KpnI and EcoRI. pGL3 vectors were subsequently co-transfected with the active reporter renilla luciferase vector phRL-TK (Promega) into 293 T cells. Two days after infection, the bioluminescence of both luciferases was measured using Dual Luciferase Reporter Gene Assay Kit (Yeasen). The firefly-derived luciferase signals were standardized by renilla-derived luciferase signals according to the instructions.

## One-step TdT-mediated dUTP Nick-End Labeling (TUNEL) apoptosis assay

Apoptosis of tissues was detected using One-step TUNEL Apoptosis Assay Kit (Abbkine). In brief, FFPE tissues of tumor were undergoing dewaxing and incubated with proteinase K (20 μg/mL). After being washed with PBS for three times, slides were incubated with TdT labeled reaction buffer at 37 °C for 2 h. Lastly, tissues were stained with DAPI and slides were sealed with glycerol for observation under a fluorescence microscope.

## Bioinformatics analysis and RNA-seq

Public CRC dataset was downloaded from GEO using "GEOquery" package. RNA-seq, clinical information, and CNV data of TCGA COADREAD and LUAD cohorts were downloaded from TCGA (https://portal.gdc.cancer.gov/), and the data was processed with R (version 3.6.1) and R Studio (version 3.4.2). Differently expressed genes (DEGs) in tumor and normal tissues were analyzed using "limma" package in several datasets (GSE24514, GSE9348, GSE20842, and GSE74602). The genes with |log₂ FC| > 1.5 and $p < 0.01$ were selected. To further identify the genes that may play a role in CRC initiation and metastasis, we further selected these DEGs with |log₂ FC| > 1.5 and $p < 0.05$ in adenoma (GSE20916) and metastasis (GSE1323). Ultimately, 64 genes that may associated with CRC progression were identified. We also screened the list of nuclear and mitochondrial proteins from The Human Protein Atlas and selected genes with both nuclear and mitochondrial locations for further study. RNA from control and MYG1 KO LoVo cells were extracted and the cDNA libraries were sequenced on the Illumina sequencing platform by Genedenovo Biotechnology Co., Ltd (Guangzhou, China).

## GSEA analysis

RNA-Seq (level-3) data of COADREAD ($n = 433$) were downloaded from TCGA. Initially, we ranked the tumor samples according to the expression levels of MYG1 and divided them into two groups: high expression (top 50% samples) and low expression (bottom 50% samples). Using the GSEA (version 4.2.1) default preranked method, Signal2Noise, we ranked all genes and conducted enrichments using Hallmarkers or KEGG pathways as reference sets[38].

## Statistical analysis

All assays were performed in at least three independent experiments. Statistical analyses were performed using the SPSS (version 26.0) software and GraphPad Prism (version 9.2.0) software. Before conducting statistical analysis, normality and homogeneity of variance tests were conducted first. Unpaired two-tailed Student's $t$-tests were used to analyze two unpaired samples. One-way ANOVA was used to analyze multiple unpaired samples. Paired two-tailed Student's $t$-test was performed to analyze the statistical significance of matched tissue samples. Chi-square tests were performed to analyze the correlation between gene expression and clinical characteristics. Pearson's coefficient tests were performed to assess the statistical significance of the correlations between the expression of two genes or gene signatures. Kaplan–Meier analysis was used for survival analysis and compared by the Log-rank test. Statistically significance was set at $p < 0.05$. Error bars represent mean ± SD. Some studies choose a representative experimental result from independent experiments to present, where independent experiments refer to experiments conducted on different days. No statistical method was used to predetermine sample size, and no data were excluded from the analyses.

## Reporting summary

Further information on research design is available in the Nature Portfolio Reporting Summary linked to this article.

## Data availability

Data from TCGA COADREAD cohort including copy number, RNA-Seq, somatic mutation and phenotypes information used in this study were downloaded from Xena Browser [https://xenabrowser.net/datapages/?cohort=TCGA%20Colon%20and%20Rectal%20Cancer%20(COADREAD)&removeHub]. The information on proteins' location was available from https://www.proteinatlas.org/. The datasets including GSE24514, GSE9348, GSE20842, GSE74602, GSE20916, GSE1323, GSE160478, and GSE39582 were downloaded from GEO database. The raw data of RNA-seq generated in this paper was deposited in GEO database (GSE241878). All data in this study are provided in the Supplementary Information and Data. Source data are provided in this paper. Source data are provided with this paper.

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

## Acknowledgements

This work was supported by the National Natural Science Foundation of China (grant 81272763, 81672466, 81972334, 82173297, and 82373066 awarded to J.Z. and grant 82273351 awarded to H.Z.) and the Natural Science Foundation of Guangdong Province (grant 2017A030313550, 2019A1515011205, and 2023A1515010329 awarded to J.Z.). We thank Dr. Jianming Zeng (University of Macau) for generously sharing their experience and codes, and Yin Wang for providing assistance during the revision.

## Author contributions

Jun Zhou and Haoxuan Zheng conceptualized and supervised this study; Jianxiong Chen and Shiyu Duan designed research and performed most experiments. Yulu Wang, Yuping Ling, Xiaotao Hou, Sijing Zhang, Jiawen Lan, Miao Zhou, and Huimeng Xu assisted in sample collection, clinical data collection, and association analyses. Xiaotao Hou provided technique support. Xiaoli Long and Xunhua Liu were engaged in statistical analyses. Jun Zhou, Jianxiong Chen, and Haoxuan Zheng prepared the manuscript. All authors reviewed and approved the manuscript.

## Competing interests

The authors declare no competing interests.
