## [Peer Review File · Nature Communications]

MYG1 drives glycolysis and colorectal cancer development through nuclear-mitochondrial collaborationREVIEWER COMMENTS

Reviewer #1 (Remarks to the Author): expertise in mitochondrial metabolism

Chen et al. investigate the role of MYG1 as a pro-tumor gene in colorectal cancer (CRC). Analysis of patient-samples reveals a correlation between MYG1 and tumor progression and low survival. To establish the relevant mechanisms, the authors make use of CRC cell lines in vitro and in xenografts to suggest that MYG1 primarily works via promoting increased glycolysis. Their major mechanistic insight is that MYG1 binds to PKM2 and induces nuclear localization of this glycolytic gene. Overall, these insights are novel, and would contribute to our understanding of the function of MYG1 in tumorigenesis. However, several technical and statistical issues (described in detail below) reduce confidence in their findings and sometimes go against the author's main conclusions. There would need to be a significant revision to address these issues. In addition, there are three key conceptual issues that preclude publication at this point:

1) The authors implicate MYG1 binding to PKM2 as a key regulator of its activity. To test their model and the importance of this binding, they would need to identify mutated versions of MYG1 (or PKM2) which specifically eliminate binding, and test how these impact glycolysis, tumorigenesis, etc.

2) The authors suggest a complicated transcriptional feedback model (Figure 8), based on a number of experiments shown in Figure 6. It is not clear that this is the simplest and only model that fits the observations. The rationale for forming this model is not clearly described, and evidence for feedback is not definitively shown in any particular experiment. Thus, I find their model to be speculative at this stage, without definite experiments to test if feedback exists.

3) The authors indicate that MYG1 has mitochondrial localization, based on their own experimental findings as well as previous publications (references 3,5). However, these data do not reliably indicate mitochondrial localization of MYG1. The gold standard for mitochondrial localization is a proteinase K shaving assay, which should be performed for both endogenous MYG1, as well as evaluation of the exogenously expressed constructs. As the authors do not particularly focus on the mitochondrial role of MYG1, or do not believe it is important, it might be prudent to eliminate evaluation of mitochondrial MYG1 from the manuscript, and instead mention it in the discussion.

Technical issues:

1) Fig. 1A – This panel is referred to as “KEGG pathway analysis of MYG1 in TCGA CRC cohort” which does not have a precise meaning and prevents the reader from understanding this panel. Please clarify the precise calculations done to create this plot.

2) A major technical issue throughout the paper is that all MYG1 knockdown experiments are done with a single shRNA. As a result, it is not known if the findings are caused by knockdown of MYG1 or an off-target effect of the shRNA. These cell lines should be rescued with a shRNA-resistant expression of MYG1 to establish specificity. Alternatively, the authors could test multiple independent shRNAs targeted to MYG1.

3) Figure 2h, what is the y-axis? No explanation is given in the text or figure legend.

4) The units for glucose consumption and lactate secretion (e.g., Figure 3b,c, Figure 4e,f, etc.) should be provided in absolute units (nmol/hr/# of cells). It is not clear if the data in these figures are appropriately normalized. There is no description for how glucose and lactate measurements are performed in the methods.

5) For all figure legends, no description is given on the statistical tests used in each panel. This precludes the reviewer from evaluating if statistics were appropriately applied to these data.

6) With reference to lines 213-218, Figure 8, etc... FDG-PET reports glucose uptake, not glycolysis. Please revise.

- 7) With respect to data in Figure 4, the FDG-PET results presented correlate with tumor volume, and thus cannot be used to make conclusions regarding the relative amounts of glucose uptake. These experiments should be performed with similar size tumors to evaluate changes in glucose uptake.
- 8) For all MYG1 overexpression experiments (Figure 2b,e, etc.) , it is not clear what "controls" refer to. Are these overexpressing a control protein? This should be clearly stated in the text and figure legends.
- 9) For extracellular flux measurements (Figures 3d,e, etc.), the flux values must be normalized to the number of cells, particularly considering the changes in proliferation. No methods were provided regarding how cell numbers were calculated and used for normalization. No statistics were performed on these flux data, and thus it is not actually statistically justified or clear that glycolysis is changing, which is a central conclusion of this manuscript.
- 10) In figure 4a, the nuclear and cytoplasmic fractions are contaminated, precluding any conclusions being made from these blots. Proteinase K shaving assays should be performed to definitively show mitochondrial localization. I also observe that different size bands are presented in the various fractions, which complicates the interpretation of this blot.
- 11) Figure 4b is confusing as presented. If different signal sequences are being deleted, what are they being replaced with?
- 12) With reference to Figure 4c, and sup figure 5a, this experimental design precludes distinguishing between endogenous and exogenous MYG1 protein. These experiments should be repeated using a tagged version of Myg1 to assess localization of the exogenously expressed protein. Proteinase K shaving assays need to be performed to validate mitochondrial localization.
- 13) With respect to Figure 4j,k: The authors results indicate that MYG1-deltaM promotes tumorigenesis in vitro even in the presence of 2-DG, which goes against the authors' conclusions that glycolysis is required for MYG1's effects. 2-DG certainly impairs tumorigenesis in both control and MYG1-deltaM lines, but this result does not imply that glycolysis is specifically required for MYG1-deltaM's effects, only that glycolysis is important for tumorigenesis.
- 14) With respect of Figure 5: As presented, figure 5a implies that there is limited to no-colocalization between MYG1 and PKM2 ... they appear to be in separate compartments. It is not clear how they can conclude that there is co-localization. Similarly, the resolution/magnification presented in Figure 5c, precludes assessment of subcellular colocalization.
- 15) With respect to Figure 6K,L – several primer pairs are used to validate c-myc binding in SW620 cell lines. No negative control primer set is used. The authors should discuss why they observe such broad binding over such a large region (~2kb) ... are there 100's of c-myc binding sequences in the MYG1 promoter region? Other cell lines should be similar assessed.
- 16) With respect to Figure 7A: resolution is not sufficient to observe mitochondrial co-localization ... the protein appears to be mainly nuclear. Mitochondrial localization should be validated with proteinase K shaving assays.
- 17) With respect to Figure 7C: the magnification presented is not appropriate to observe cytochrome C co-localization with MYG1. Similarly, the "obvious" nuclear localization in Figure 8D is not appropriately presented at the provided magnification.
- 18) With respect to Figure 7F,G,H – the authors show that MYG1-deltaN promotes cytochrome C release from mitochondria in panel F, which is counter to the inhibition of apoptosis shown in panels G,H, and the discussion in the text (lines 354-6).

Reviewer #2 (Remarks to the Author): expertise in MYG1 biology

In this work the authors address the molecular mechanistic basis of colorectal cancer (CRC) progression. Based on poor prognosis of CRC patients with elevated Myg1 levels, this nucleo-mitochondrial orchestrator emerges as a key orchestrator of CRC progression.

The authors demonstrate that this effect is mediated via the MYC transcription factor and involves alterations in OXPHOS and inhibition of apoptosis. The work is meticulously designed and executed well with appropriate controls for comparison.

Overall I am happy with the work. However some of the points need to be clarified and adequately addressed to warrant publication.

1. Since Myg1 is a mitochondrial gene it is likely that silencing affects mitochondrial functions and thereby reduces migration, invasion and cell proliferation. In this light the over expression data is important and has been demonstrated. However, if Myg1 accelerates cell cycle, how are the authors sure that observed effects on migration and invasion are not a consequence of altered proliferation?

2. In Fig 2d silencing of MYG1 in LoVo cells is not evident and the same must be quantified.

3. In Fig 3 the authors compare the MYG1 high and Myg1 low CRCs in TCGA data. While some of them could be attributed to Myg1 others can be confounding changes not attributable to Myg1 directly. Have the authors performed transcriptomic analysis of the Myg1 OE or silenced CRC cells to ascertain the causality? Also a previous study in Myg1 identified several transcripts altered by Myg1 in melanoma cells (Grover R et al NAR 2019). Have the authors compared these datasets?

4. Since silencing of MYC as well as PKM2 both downregulate MYG1, suggestion of the positive feedback loop needs substantiation as it is likely that MYC and PKM2 may be upstream of MYG1. Could the augmented levels of PKM2 upon over expression MYG1 a mere stabilization effect? Or there are alternate regulatory mechanisms present? This needs clarity.

5. While the nuclear form of MYG1 seems to be important and mechanistic support of its involvement is clear, the involvement of the mitochondrial form is not very clear and needs to be elaborated.

6. From the study the role of MYG1 as an exonuclease is not explored. Do the authors think that DHH mutant would be unable to recapitulate the effects of the nuclear MYG1? Also previous work seems to suggest that the nuclear role of Myg1 is two-fold, it regulates NE mito genes and alters ribosomal RNA processing and eventually controls ribosomal assembly and the overall translation in the cell.

How do the authors interpret their results in this context?

Reviewer #3 (Remarks to the Author): expertise in colorectal cancer metabolism

In the current manuscript, the authors report that melanocyte proliferating gene 1 (MYG1) can regulate glycolysis and promote CRC progression by coordinating its nuclear-mitochondrial functions. They proposed that nuclear MYG1 forms a complex with PKM2 dimer and induces the accumulation of PKM2 in the nucleus. Nuclear PKM2 transcriptionally activates MYC and promotes MYC-mediated glycolysis. c-Myc also transcriptionally upregulates MYG1, forming a positive feedback loop. They also proposed that mitochondrial MYG1 inhibits apoptosis and interferes with OXPHOS.

There are a few critical comments:

1. In the introduction, the authors talked about "approximately 900 000 individuals die from this malignancy". But it is not clear how long and where this number is related to.

2. Statistical analysis is missing for Figure 1C.

3. The statement that "MYG1 is upregulated in the genetic background of APC loss and KRAS mutation in CRC" is not convincing. Figure 1G showed that SW480 cells have mutations in APC and

- KRAS, but MYG1 expression was very low. Moreover, RKO cells do not have mutations in APC and KRAS, but MYG1 expression was very high.
4. It is very confusing that "high expression of MYG1 harms the progression and survival of CRC patients".
 5. It is not clear what is "MYG1 accelerated G1/S cells conversion in CRC cells".
 6. The quantification results for figure 2D and Figure S3 are lacking. The quality for Figure S3 needs to be improved as well.
 7. "Next, we confirmed that MYG1 promoted the expression of GLUT1, LDHA, and PKM2 using qPCR (Supplementary Fig. S4B)." Why only these three genes were selected? Why not GAPDH?
 8. Quantification and statistical analysis are preferred for Fig. 3D-E.
 9. Figure 4A, "Interestingly, MYG1 was also detected in the cytoplasm with the mitochondria removed (C-M) with a smaller molecular weight." Which is the right band for MYG1? It seems that there are no gel shifts in RKO cells.
 10. Where is the construct with both signal peptide deleted (MYG1 Δ L) in figure 4C?
 11. It is also not convincing that the construct for nuclear localization peptide deleted (MYG1 Δ N) is working based on the results of Figure S5A.
 12. Again, the results in figure 4d is not convincing.
 13. "Although the MYG1 Δ N tumor showed no statistical difference in tumor size, the average volume and luciferin signal were also higher than those in the control group." This sentence is confusing.
 14. "In addition, MYG1 Δ M mice also had advanced tumor cell invasion and intraperitoneal dissemination of tumors compared with the MYG1 Δ N group (Supplementary Fig.S5C-D)." It is hard for the reviewer to appreciate these results.
 15. "The transwell invasion assay showed that cells in MYG1 Δ M group were more aggressive (Supplementary Fig. S5G)." It seems that the WT MYG1 had the greatest effect. It may be worth to include data from MYG1 Δ L.
 16. "Among 180 candidate proteins in the nucleus (Supplementary table S3), we focused on PKM2, a key enzyme that catalyze the conversion of phosphoenolpyruvate to pyruvate." Only PKM1 was found in the table. Any explanation?
 17. Figure 5, the majority of PKM2 locates in the cytosol. The reason for the specificity of PKM2 and MYG1 interaction in the nucleus needs to be discussed.
 18. Grammar and spelling errors need to be carefully corrected. Eg. Line 258, 266, 295, 349, 382, 573, 730.
 19. How many times the authors have repeated for figure 5D? Statistical analysis is needed.
 20. The labeling for constructs for figure 5e should be consistent with others in the manuscript.
 21. "The results showed that PKM2 and c-Myc were highly expressed in CRC cell lines compared with FHC, and the expression of PKM2, c-Myc, and MYG1 was highly correlated (Supplementary Fig. S7A)." This statement needs statistical analysis support.
 22. Figure 6C, which is the right band for c-Myc?
 23. Figure 6D, why suddenly switched to Caco-2 cells? No quantification?
 24. Figure S7E, were the changes statistically significant?
 25. Figure 6K-I, where are the exact binding sites for c-Myc in MYG1 promoter? Any Myc response element? Figure 6L is not shown.
 26. This statement "The result showed that cyt c was obvious downregulated in cells expressing MYG1 Δ WT and MYG1 Δ N (Fig. 7D)." lacks statistical analysis support.
 27. This statement "MYG1 Δ N can downregulate cyt c and inhibit the OXPHOS of cells, while it can also inhibit the release of cyt c from the mitochondria and inhibit the apoptosis of CRC cells" conflicts with the results in Figure 7F at least.
 28. Figure 8d, apoptosis should be examined to verify the function of mitochondrial MYG1.
 29. In figure 6E, why C3K inhibits PKM2 but increased the expression of c-Myc and MYG1? This conflicts with Figure 6B. The discussion should move to the results section.
 30. Line 479-480, Cells (1x 10⁶) cells were subcutaneously injected into the splenic capsules of mice?
 31. Supplementary information is available at Cell death & Differentiation's website?

Reviewer #4 (Remarks to the Author): expertise in colorectal cancer cell biology

In this manuscript Chen and colleagues describe how MYG1 regulates tumor aggressiveness in

CRC. They show that MYG1 is higher expressed in CRC, associated with prognosis and stage and that loss of MYG1 dampens and gain of MYG1 increases migration proliferation, invasion and in vivo metastasis. They continue to show that MYG1 requires its nuclear and not its mitochondrial localization for this effect and show that MYG1 is regulated by PKM2 and MYc. Importantly, MYG1 dampens respiration and enhances glycolysis.

Although the data are of interest they require a lot more validation in relation to earlier data with MYG1. The findings of the authors maybe true and tissue specific but to choose to ignore the 180 degree different observations in LUAD, skin and yeast cells is not the way forward.

1. Others have provided compelling evidence that loss of MYG1 decreases respiration, while gain of expression as seen in LUAD is associated with enhanced oxidative phosphorylation. The authors will need to explain these differences and study their effects in relation to the observed exonuclease activity of MYG1. Maybe the difference is in nuclear versus mitochondrial but they need to address these differences experimentally. If they want to claim this role in CRC they need to show that under the exact same expression levels the role is different in LUAD. If proven then it would also make sense to look for the reason as Myc is equally important in LUAD.

2. The authors start with the overexpression data of MYG1, but the data shown do not show a highly convincing impact of transformation. A lot of tumors by RNA or western do not show overexpression at all. What is the underlying reason? The authors suggest that this would be APC Kras but this is also not very solid data. The mice data are marginal higher expressed at best and the human data appear overinterpreted (RKO is a very strong expressor while SW480 has no overexpression). The authors should analyse the databases which often have mutation data associated to it (TCGA) or use subtype stratification.

3. The authors should also increase the survival analysis as the association with poor prognosis is not detected in publicly available sets.

4. The authors show that the glucose uptake is dep on nucl localization of MYG1 but surprisingly the lactate production seems less or not affected if nuclear localization is blocked. This needs further substantiation. Same for the in vivo Pet analysis which to this reviewer shows that the PET activity measured is identical in all 4 conditions. If the authors would have normalized the signal to the tumor volume there is no difference left. This implies that all tumor cells are as active in glucose uptake which is not consistent with the in vitro data. They clearly grow differently but do not take up glucose at a different rate.

RESPONSE TO REVIEWERS' COMMENTS

Note to reviewers: We thank reviewers for taking efforts to review our manuscript and for providing insightful comments to further improve our paper. Below we provide the detailed point-by-point response to address all the comments raised by reviewers. To facilitate the review of our manuscript and rebuttal letter by reviewers, we present the key data as rebuttal letter figures in this letter, with referrals to corresponding figures and text in our revised manuscript. We have also marked all the changes in our revised manuscript by red highlighting.

REVIEWER COMMENTS

Reviewer #1 (Remarks to the Author): expertise in mitochondrial metabolism

Chen et al. investigate the role of MYG1 as a pro-tumor gene in colorectal cancer (CRC). Analysis of patient-samples reveals a correlation between MYG1 and tumor progression and low survival. To establish the relevant mechanisms, the authors make use of CRC cell lines in vitro and in xenografts to suggest that MYG1 primarily works via promoting increased glycolysis. Their major mechanistic insight is that MYG1 binds to PKM2 and induces nuclear localization of this glycolytic gene. Overall, these insights are novel, and would contribute to our understanding of the function of MYG1 in tumorigenesis. However, several technical and statistical issues (described in detail below) reduce confidence in their findings and sometimes go against the author's main conclusions. There would need to be a significant revision to address these issues. In addition, there are three key conceptual issues that preclude publication at this point:

We thank the reviewer for reviewing our manuscript and providing valuable comments. We have taken these concerns seriously and revised our manuscript. The newly added contents are highlighted in the red text and we hope our response sufficiently addresses your concerns. Thank you for your time and consideration. Please note that the figure citations in our response blow refer to the rebuttal figures and the corresponding figures in the revised manuscript.

1) The authors implicate MYG1 binding to PKM2 as a key regulator of its activity. To test their model and the importance of this binding, they would need to identify mutated versions of MYG1 (or PKM2) which specifically eliminate binding, and test how these impact glycolysis, tumorigenesis, etc.

Thanks for your significant suggestions. Based on your suggestions and those of other reviewers, we have further investigated the regulatory mechanism of

MYG1 on PKM2 and provided additional experimental evidence to confirm the significance of the interaction between MYG1 and PKM2 in glycolysis and tumor progression.

According to your advice, we constructed seven truncated fragments of MYG1 and added a Flag tag at the C-terminus. We explored the binding sequence of MYG1 to PKM2 by Co-IP. The results showed that MYG1 directly interacts with PKM2 through the 149-199 fragment (**Rebuttal Fig 1a**; Fig. 5g in the manuscript).

To further investigate the importance of MYG1 binding to PKM2 in MYG1-induced glycolysis and tumorigenesis, we generated a mutant MYG1 by deleting the 149-199 amino acid sequence (MYG1 Δ), which lost the ability of binding to PKM2 and stabilizing PKM2 in the nucleus (**Rebuttal Fig. 1b**; Fig. 5n in the manuscript). We next detected the effect of MYG1 Δ on the glycolysis and the proliferation and invasion ability of CRC cells. Compared to the wild type, the MYG1 Δ variant exhibited a significant reduction in its ability to promote lactate secretion and glucose uptake in HCT116 cells (**Rebuttal Fig. 1c**; Fig. 6c in the manuscript). Furthermore, MYG1 Δ also displayed marked decreases in cell proliferation and invasion capabilities compared to wild-type (**Rebuttal Fig. 1d-e**; Fig. 6d-e and Supplementary Fig. 7c in the manuscript). These results suggested that the interaction of MYG1 and PKM2 is important for glycolysis and tumorigenesis induced by MYG1 in CRC.

Rebuttal Fig. 1. The interaction of MYG1 and PKM2 is important for glycolysis and tumorigenesis induced by MYG1 in CRC.

2) The authors suggest a complicated transcriptional feedback model (Figure 8), based on a number of experiments shown in Figure 6. It is not clear that this is the simplest and only model that fits the observations. The rationale for forming this model is not clearly described, and evidence for feedback is not definitively shown in any particular experiment. Thus, I find their model to be speculative at this stage, without definite experiments to test if feedback exists.

Thank you for providing constructive feedback. We have carried out additional experiments to comprehensively validate our model.

1) To begin with, we studied the regulation mechanism of PKM2 by MYG1. MYG1 can regulate the mRNA and protein levels of PKM2, but it remains unclear whether the regulation of PKM2 by MYG1 depends on this interaction. In response to your 1st question, we have identified the binding sequences between MYG1 and PKM2 and proved the significance of their interaction in glycolysis and tumorigenesis. We also confirmed that the upregulation of PKM2 depends on its interaction with MYG1 (**Rebuttal Fig. 1**). Subsequently, we explored the specific mechanism. By knocking out MYG1 with protein translation inhibited by CHX in LoVo cells, we demonstrated that MYG1 can maintain the protein stability of PKM2 (**Rebuttal Fig. 2a**; Fig. 5i in the manuscript).

It is reported that post-translational modifications such as phosphorylation, acetylation, or ubiquitination of PKM2 may affect its stability or degradation ¹. Hence, we propose that MYG1 might influence the post-translational modifications of PKM2. Results from our RNA-seq indicated a potential association between MYG1 and heat shock proteins (**Rebuttal Fig. 2b**; Fig. 3a in the manuscript). Heat shock proteins play crucial roles in cellular process, interacting with newly synthesized proteins and participating in post-translational modifications such as phosphorylation, acetylation, methylation, etc. Through these modifications, heat shock proteins can influence the stability, activity, and interaction of proteins, thereby regulating cellular physiological processes or tumorigenesis ²⁻³. Previous two independent studies have reported the physical interaction between MYG1 and Hsp90 ⁴⁻⁵. These clues led us to speculate that MYG1 might mediate the post-translational modifications of PKM2 by interacting with HSP90, thereby enhancing its stability. Subsequently, we confirmed the interaction between MYG1 and HSP90 through Co-IP, and HSP90 was found to interact with PKM2 as well (**Rebuttal Fig. 2c-d**; Fig. 5j-k in the manuscript). Xu Q. et al. has reported that HSP90 can recruit GSK3 β to promote PKM2 Thr-328 phosphorylation, increasing its stability and facilitating glycolysis and proliferation in liver cancer cells ⁶. Therefore, we hypothesize that a similar mechanism may exist in CRC, where MYG1 serves as a client protein of HSP90, recruiting it to modify PKM2 and

enhance its stability. To investigate this, we validated that GSK3 β can interact with MYG1 and PKM2 (**Rebuttal Fig. 2e-f**; Fig. 5l-m in the manuscript). Finally, we also confirmed that overexpression of MYG1 in HCT116 increased the phosphorylation of PKM2 (**Rebuttal Fig. 2g**; Fig. 5n in the manuscript). Knocking down the level of HSP90 in LoVo cells led to a decrease in PKM2 phosphorylation (**Rebuttal Fig. 2h**; Fig. 5o in the manuscript). Treatment with GSK3 inhibitor IX, a GSK3 β inhibitor, also reduced the phosphorylation of PKM2 (**Rebuttal Fig. 2g**; Fig. 5n in the manuscript). These findings suggest that MYG1 facilitates the phosphorylation of PKM2 through recruits HSP90/GSK3 β complex, enhancing its stability and thereby regulating its protein level.

Rebuttal Fig. 2. MYG1 recruits HSP90/GSK3 β complex and enhances the stability of PKM2.

2) Nuclear PKM2 transcriptionally regulate MYC expression. Nuclear PKM2 can regulate glycolysis by acting as a kinase to regulate downstream key transcription factors, or directly interacting with transcription factors to regulate transcription in cancer¹. GSEA analysis of TCGA CRC datasets revealed the association of MYG1 with the MYC targets pathway (**Rebuttal Fig. 3a**; Fig. 6g in the manuscript). Studies have reported that PKM2 can interact with β -catenin to promote MYC transcription and thereby regulate key enzymes in the glycolysis process⁷. Therefore, we speculate that MYG1 may activate MYC through PKM2 in CRC. Firstly, we established the correlation between MYG1,

PKM2, and MYC expression in cell lines (**Rebuttal Fig. 3b**; Supplementary Fig. 6i in the manuscript). Additionally, we found that nuclear MYG1 can enhance MYC protein expression (**Rebuttal Fig. 3c**; Fig. 6h in the manuscript).

3) c-Myc binds to MYG1 promoter and promotes MYG1 transcription.

Interestingly, when we knocked down PKM2, we observed a downregulation not only in c-Myc expression but also in MYG1 protein levels (**Rebuttal Fig. 3d**; Fig. 6i in the manuscript). This led us to consider the existence of a feedback regulatory mechanism, where c-Myc may regulate MYG1 transcription. Initially, we knocked down and overexpressed c-Myc and found that c-Myc can regulate MYG1 mRNA levels (**Rebuttal Fig. 3e-f**; Fig. 6j-k in the manuscript). We next analyzed whether c-Myc could bind to MYG1 promoter using online ChIP-seq dataset. The results showed an enrichment of a peak before the transcriptional start site (TTS) of MYG1 in several CRC cell lines (**Rebuttal Fig. 3g**; Supplementary Fig. 8h in the manuscript). To examine the possible binding site of c-Myc, we analyzed the 2000 bases before TSS of MYG1 on JASPER website. The results showed multiple loci exhibited the potential of binding to c-Myc (**Rebuttal Fig. 3h**; Fig. 6l in the manuscript). We then performed ChIP-qPCR using SW620 and LoVo cells. As the result shown, c-Myc can bind to the MYG1 promoter at multiple sites, mainly at P1 and P6-P11 sequence (**Rebuttal Fig. 3i-j**; Fig. 6m-n and Supplementary Fig. 8i in the manuscript). However, whether c-Myc transcriptionally regulate MYG1 expression was unclear. We next performed double luciferase reporter gene experiment to examine the regulatory sequences. Fragment 1 (F1, from -1926 to -1735 bases) and fragment 2 (F2, from -713 to 0 bases) from MYG1 promoter were used to validate the transcriptional regulation activity and F2 was regulated by c-Myc and PKM2 significantly (**Rebuttal Fig. 3k**; Fig. 6o in the manuscript).

In summary, these experiments successfully confirmed the presence of a positive feedback regulatory loop, where MYG1 regulates PKM2 and MYC expression, while c-Myc, conversely, promotes MYG1 transcription.

References:

- 1, Zhang Z et al. PKM2, function and expression and regulation. Cell Biosci. 2019 Jun 26;9:52.
- 2, Wu J et al. Heat Shock Proteins and Cancer. Trends Pharmacol Sci. 2017 Mar;38(3):226-256.
- 3, Hu C et al. Heat shock proteins: Biological functions, pathological roles, and therapeutic opportunities. MedComm (2020). 2022 Aug 2;3(3):e161.
- 4, Falsone S.F. et al. A proteomic snapshot of the human heat shock protein 90 interactome. FEBS Lett. 2005 Nov 21;579(28):6350-4.
- 5, Millson SH, et al. A two-hybrid screen of the yeast proteome for Hsp90 interactors uncovers a novel Hsp90 chaperone requirement in the activity of a

stress-activated mitogen-activated protein kinase, Slf2p (Mpk1p). *Eukaryot Cell*. 2005 May;4(5):849-60.

6, Xu Q, et al. HSP90 promotes cell glycolysis, proliferation and inhibits apoptosis by regulating PKM2 abundance via Thr-328 phosphorylation in hepatocellular carcinoma. *Mol Cancer*. 2017 Dec 20;16(1):178.

7, Yang W et al. Nuclear PKM2 regulates β -catenin transactivation upon EGFR activation. *Nature*. 2011 Dec 1;480(7375):118-22.

Rebuttal Fig. 3. c-Myc binds to MYG1 promoter and promotes MYG1 transcription.

3) The authors indicate that MYG1 has mitochondrial localization, based on their own experimental findings as well as previous publications (references 3,5). However, these data do not reliably indicate mitochondrial localization of MYG1. The gold standard for mitochondrial localization is a proteinase K

shaving assay, which should be performed for both endogenous MYG1, as well as evaluation of the exogenously expressed constructs. As the authors do not particularly focus on the mitochondrial role of MYG1, or do not believe it is important, it might be prudent to eliminate evaluation of mitochondrial MYG1 from the manuscript, and instead mention it in the discussion.

We thank the reviewer for these constructive suggestions. We appreciate your suggestion to perform a proteinase K shaving assay as the gold standard to reliably determine mitochondrial localization. This assay can determine the specific location of MYG1 in the outer mitochondrial membrane or in the mitochondrial matrix. This is important because the localization of MYG1 in the mitochondria determines how it functions.

By performed the proteinase K shaving assay, we found that both exogenously introduced and endogenous MYG1 were located in the mitochondrial matrix (Fig. 4b and Supplementary Fig. 5a in the manuscript). Besides, to visualize the mitochondrial localization of MYG1 from a morphological perspective, we also performed immunoelectron microscopy observation using exogenously expressing flag-labeled MYG1 in 293T cells. As the results shown, MYG1 was located in the mitochondrial matrix (Fig. 4c in the manuscript).

Furthermore, you also suggest that we could exclude the functional study of MYG1 in the mitochondria. However, detection of glycolytic phenotype and tumor biology functions revealed that while nuclear MYG1 plays a dominant role, mitochondrial MYG1 also demonstrates functionality. Since our main focus was on the mechanisms of nuclear MYG1, we only investigated the function of mitochondrial MYG1 on oxidative phosphorylation and apoptosis. We understand your suggestion of mentioning it in the discussion section, and we have discussed the limitations of mitochondrial functional mechanisms in the discussion section (Page 25, Line 539-543 in the manuscript). However, we believe it would be more comprehensive and scientifically rigorous to include the evaluation of mitochondrial MYG1 in the results section, as it is pertinent to the overall understanding of MYG1 localization and function.

Technical issues:

1) Fig. 1A – This panel is referred to as “KEGG pathway analysis of MYG1 in TCGA CRC cohort” which does not have a precise meaning and prevents the reader from understanding this panel. Please clarify the precise calculations done to create this plot.

Thank you for your comment. We have revised the title as “GSEA” to provide a more specific description of the calculations performed to create the plot. Specifically, we performed the Gene Set Enrichment Analysis (GSEA) of MYG1

in the TCGA COADREAD cohort. The enrichment results with $p < 0.05$ and FDR q value < 0.25 were listed in the Supplementary Data 1 and the methodology used to conduct the analysis was detailed in the Methods. To visualize whether MYG1 was associated with metabolic remodeling progress in CRC, we visualize the KEGG pathway of the GSEA analysis in Fig. 1a, highlighting the metabolic pathways with red points. Besides, top 10 metabolic pathways were listed beside the figure.

2) A major technical issue throughout the paper is that all MYG1 knockdown experiments are done with a single shRNA. As a result, it is not known if the findings are caused by knockdown of MYG1 or an off-target effect of the shRNA. These cell lines should be rescued with a shRNA-resistant expression of MYG1 to establish specificity. Alternatively, the authors could test multiple independent shRNAs targeted to MYG1.

Thank you for your insightful comment. We appreciate your concerns regarding the specificity of the observed findings and the potential for off-target effects of the single shRNA used. To address this issue, we employed two shRNAs targeted to MYG1 and performed the in vitro assays to evaluate the function of MYG1. The results showed that both shRNA inhibited the proliferation, colony formation, migration and invasion ability (Fig. 2a-d and Supplementary Fig. 2a-d in the manuscript).

Besides, we also knocked out MYG1 by incorporating CRISPR-Cas9 mediated gene silencing technology. Compared to RNAi, CRISPR-Cas9 has a lower off target effect. We knocked out MYG1 in LoVo cells and concluded similar results (Fig. 2i-l and Supplementary 2i-l in the manuscript). These results provided more robust evidence for the role of MYG1 in the observed phenotypes.

3) Figure 2h, what is the y-axis? No explanation is given in the text or figure legend.

Thank you for pointing out our omission. We have revised the figure legends in the final manuscript to provide a clear and concise description. The y-axis of Figure 2h (old version) represents the number of liver metastases. We have provided a more detailed description to ensure that readers have a comprehensive understanding of the data presented (Fig. 2o in the manuscript).

4) The units for glucose consumption and lactate secretion (e.g., Figure 3b,c, Figure 4e,f, etc.) should be provided in absolute units (nmol/hr/# of cells). It is not clear if the data in these figures are appropriately normalized. There is no description for how glucose and lactate measurements are performed in the methods.

Thank you for your professional suggestions. We have revised the glucose uptake and lactate secretion data in absolute units in the revision (Fig. 3 and 6, and Supplementary Fig. S4 in the manuscript).

Additionally, we have also provided a thorough experiment procedure with details relating to glucose uptake and lactate secretion in the Methods section (Page 36, Line 776-786 in the manuscript).

5) For all figure legends, no description is given on the statistical tests used in each panel. This precludes the reviewer from evaluating if statistics were appropriately applied to these data.

Thank you for your valuable feedback. We have updated the figure legends and included a description of the statistical tests employed for each panel. We have specified the type of test (e.g., t-test, ANOVA, etc.) and any relevant parameters employed for each panel.

6) With reference to lines 213-218, Figure 8, etc... FDG-PET reports glucose uptake, not glycolysis. Please revise.

Thank you for correction. We have made the revisions in the final manuscript.

7) With respect to data in Figure 4, the FDG-PET results presented correlate with tumor volume, and thus cannot be used to make conclusions regarding the relative amounts of glucose uptake. These experiments should be performed with similar size tumors to evaluate changes in glucose uptake.

We thank the review for the great comment. We agree that it is necessary to standardize tumor volume and individual differences when assessing the glucose uptake in vivo.

1) In fact, we have standardized the PET results using SUV in evaluating glucose uptake. During PET-CT detection of tumor glucose uptake, standard uptake rate (SUL) or standard uptake value (SUV) are commonly used as metrics. Due to the convenience and clinical relevance, SUV is a more commonly used and accepted method for assessing tumor glucose uptake in clinical practice. This metric allows for quantitative analysis of the uptake level between different patients and different lesions. It provides information about tumor biology, metabolic status, and the entire lesion or the overall tumor burden. By calculating the maximum SUV (SUV_{max}), information about the voxel with the highest uptake in the lesion can be obtained, thus reflecting the maximum uptake level of the lesion. This index makes it relatively easy to compare the metabolic activity between different lesions. According to the FDG PET/CT procedure guidelines for tumor imaging ¹, we referenced the

operational procedures and result analysis methods, and calculated SUV using the following formula: $SUV = \text{activity in tumor (kBq/mL)} / \text{injected activity (MBq)} / \text{mouse weight (kg)}$. According to the guideline, SUVmax is recommended as a tumor uptake metrics. Therefore, we employed a similar approach to previous researchers in evaluating tumor glucose uptake²⁻⁵. In the revised version, we have detailed experimental operations and analysis methods (Page 30-31).

2) However, when studying the glucose uptake function of gene in vivo using PET-CT, it is difficult to exclude the confounding effects of gene on glucose uptake. Our aim in the experiment is to demonstrate the regulatory role of MYG1 in the glycolytic process of colorectal cancer in vivo. Through PET-CT imaging, we can assess the uptake of glucose in tumors with different group of tumors. In order to simulate CRC in mice as closely as possible, we constructed an orthotopic model. However, due to the functional complexity of MYG1 in colorectal cancer, it may exert regulatory effects on various biological characteristics of tumors, such as tumor volume, the number of blood vessels in the tumor, tumor microenvironment, and so on. These factors may interfere with the evaluation of SUVmax. However, it is challenging to completely eliminate confounding factors, especially tumor volume, in in-vivo studies. Interestingly, in a recent study investigating the relationship between cold exposure and tumors⁶, researchers conducted an intriguing experiment by controlling the tumor growth to the same volume in different groups of mice before performing PET-CT scans. This approach can eliminate the confounding effect of tumor volume differences, allowing for a more accurate assessment of the impact of treatment factors on glucose uptake and avoiding bias caused by varying tumor sizes. However, there are some limitations to consider. Although standardizing tumor volume to the same size can mitigate the impact of volume differences, other extrinsic factors may still confound the results. Variations in tumor biology, cellular composition, and other factors may still exert influence on glucose uptake. Besides, prolonging the growth time of tumors may alter their pathological characteristics and malignancy level.

3) Therefore, we attempted sensitivity analysis to assess the robustness of the results to these confounding factors. By adjusting the impact of confounding factors, such as excluding volumes within a specific range, we can verify whether the results are sensitive to these factors and evaluate the consistency of the results. Due to the fact that the bioluminescent signal can better reflect the cell population within the tumor than volume, we excluded tumors with signals lower than 5×10^5 P/s/mm² and higher than 30×10^5 P/s/mm², and re-evaluated the levels of glucose uptake regulated by MYG1 in different groups. The results indicated that MYG1 regulates glucose uptake consistently, and its variation is not sensitive to tumor volume (**Rebuttal Fig. 4a**).

4) Importantly, we also examined the GLUT1 expression of different tumors to evaluate the glucose uptake. GLUT1 is a key transporter responsible for transporting glucose into epithelial cells and playing a crucial role in cellular glycolysis. Immunohistochemical staining results indicated that MYG1 upregulated the expression levels of GLUT1 in tumor cells, particularly in MYG1^N group, which is consistent with the in vitro results of western blot (**Rebuttal Fig. 4b**; Supplementary Fig. 5d and f in the manuscript). This offers additional evidence to support our conclusions regarding tumor glucose uptake.

We sincerely appreciate your insightful comments. Necessary adjustments and additional experiments have been added in the revised version to prove our conclusion.

Rebuttal Fig. 4. MYG1^N promotes glucose uptake and GLUT1 expression in vivo.

Reference:

- 1, Boellaard R et al. FDG PET/CT: EANM procedure guidelines for tumour imaging: version 2.0. *Eur J Nucl Med Mol Imaging.* 2015 Feb;42(2):328-54.
- 2, Li Q et al. Circular RNA MAT2B Promotes Glycolysis and Malignancy of Hepatocellular Carcinoma Through the miR-338-3p/PKM2 Axis Under Hypoxic Stress. *Hepatology.* 2019 Oct;70(4):1298-1316.
- 3, Weng ML et al. Fasting inhibits aerobic glycolysis and proliferation in colorectal cancer via the Fdft1-mediated AKT/mTOR/HIF1 α pathway suppression. *Nat Commun.* 2020 Apr 20;11(1):1869.
- 4, Demircioglu F et al. Cancer associated fibroblast FAK regulates malignant cell metabolism. *Nat Commun.* 2020 Mar 10;11(1):1290.
- 5, Zheng F et al. The HIF-1 α antisense long non-coding RNA drives a positive feedback loop of HIF-1 α mediated transactivation and glycolysis. *Nat Commun.* 2021 Feb 26;12(1):1341.
- 6, Seki T. Brown-fat-mediated tumour suppression by cold-altered global metabolism. *Nature.* 2022 Aug;608(7922):421-428.

8) For all MYG1 overexpression experiments (Figure 2b,e, etc.) , it is not clear

what “controls” refer to. Are these overexpressing a control protein? This should be clearly stated in the text and figure legends.

Thank you for your comment. The controls we employed were mock transfections or transfections with an empty vector, depending on the experimental design. We have revised the description and replaced it with “EV” which means empty vector.

9) For extracellular flux measurements (Figures 3d,e, etc.), the flux values must be normalized to the number of cells, particularly considering the changes in proliferation. No methods were provided regarding how cell numbers were calculated and used for normalization. No statistics were performed on these flux data, and thus it is not actually statistically justified or clear that glycolysis is changing, which is a central conclusion of this manuscript.

Thank you for your valuable feedback. The flux values were normalized to cell number in our results. To address these concerns, we have detailed the procedure of our experiments and the normalization methods. These details are explained in our methodology (Page 35, Line 765 in the manuscript).

Besides, we appreciate your suggestion regarding the statistical analysis. We have incorporated the statistical analysis in the revised version, and the specific methods are detailed in the figure legends.

10) In figure 4a, the nuclear and cytoplasmic fractions are contaminated, precluding any conclusions being made from these blots. Proteinase K shaving assays should be performed to definitively show mitochondrial localization. I also observe that different size bands are presented in the various fractions, which complicates the interpretation of this blot.

Thank you for your feedback. We have retested the expression of MYG1 in different cellular fractions and have replaced the blurry bands. Our previous bands with different size were detected by western blot using anti-MYG1 antibodies from MilliporeSigma (HPA038627). In the revision, we used anti-MYG1 antibody from Abcam (ab204420) and no gel shift was observed. We have replaced the bands of MYG1 with the new one (Fig. 4a in the manuscript).

Additionally, we have performed proteinase K shaving assay to confirm the localization of endogenous and exogenous MYG1 in the mitochondrial matrix (Fig. 4b and Supplementary Fig. 5a in the manuscript).

11) Figure 4b is confusing as presented. If different signal sequences are being deleted, what are they being replaced with?

Thanks for your question. We apologize for not providing a clear description of the vector construction strategy. In fact, we directly deleted the signal peptide in the MYG1 CDS region when constructing the vector, without substituting it with irrelevant amino acid sequences. We have redrawn the schematic diagram in Fig. 4d.

12) With reference to Figure 4c, and sup figure 5a, this experimental design precludes distinguishing between endogenous and exogenous MYG1 protein. These experiments should be repeated using a tagged version of Myg1 to assess localization of the exogenously expressed protein. Proteinase K shaving assays need to be performed to validate mitochondrial localization.

Thanks for your comments. We have designed all vectors with a C-terminal Flag tag and validated their expression and localization using western blot (WB) and immunofluorescence (IF) experiments (**Rebuttal Fig. 5a-c**; Fig. 4d-e and Supplementary Fig. 5c in the manuscript). All variants were successfully expressed and localized as intended. Additionally, we have performed proteinase K shaving assay to confirm the localization of endogenous and exogenous MYG1 in the mitochondrial matrix (Fig. 4b and Supplementary Fig. 5a in the manuscript).

Rebuttal Fig. 5. Schematic of MYG1 domain and variant constructs and variants were validated by western blot and IF.

13) With respect to Figure 4j,k: The authors results indicate that MYG1-deltaM promotes tumorigenesis in vitro even in the presence of 2-DG, which goes against the authors' conclusions that glycolysis is required for MYG1's effects. 2-DG certainly impairs tumorigenesis in both control and MYG1-deltaM lines, but this result does not imply that glycolysis is specifically required for MYG1-

deltaM's effects, only that glycolysis is important for tumorigenesis.

Thank you for your comments. The results showed that although 2-DG treatment showed a significant decrease in both EV and MYG1^N groups, the promoting effect of MYG1^N on colony formation and invasion was weakened (Fig. 5l-m in the manuscript). The effect of 2-DG can be reflected by the ability changes of MYG1^N in colony formation rate and cell invasion.

However, under the presence of 2-DG, MYG1^N still exhibits a certain degree of tumor-promoting activity, suggesting that while glycolysis plays a crucial role in this process, MYG1^N may also function through other pathways. MYG1^N could potentially impact tumor biology through other metabolic pathways, signaling pathways, or cellular physiological processes. Additionally, tumor cells have the ability to adapt to their environment and develop resistance. Under conditions of glycolysis inhibition, tumor cells may adapt to the changes by adjusting their metabolic pathways, enhancing alternative metabolic strategies, or activating other cellular survival pathways. In such cases, the overexpression of MYG1^N may also play a role by participating in these adaptive responses. However, the other functions of MYG1^N in CRC are yet to be further investigated in our future studies.

14) With respect of Figure 5: As presented, figure 5a implies that there is limited to no-colocalization between MYG1 and PKM2 ... they appear to be in separate compartments. It is not clear how they can conclude that there is co-localization. Similarly, the resolution/magnification presented in Figure 5c, precludes assessment of subcellular colocalization.

Thanks for your comments. To address these concerns, we have improved our study design and imaging techniques. As our results show, PKM2 is primarily distributed in the cytoplasm and to a lesser extent in the nucleus. Studies have reported that EGF can activate RAS/RAF signals to promote PKM2 phosphorylation and enter the nucleus, and this is required for tumor development¹⁻². In the revision, we proved that MYG1 is upregulated in patients with KRAS mutations. Sustained activation of KRAS promotes nuclear translocation of PKM2 and enhanced the downstream c-Myc mediated glycolysis. In this scenario, increased interaction between MYG1 and PKM2 plays an important role in promoting glycolysis in CRC.

Based on your considerations, we therefor tested the relationship between MYG1 and PKM2 under EGF stimulation and found that the interaction of MYG1 and PKM2 was increased. However, the effect of EGF was limited in LoVo cells with sustained activation of KRAS (Fig. 5a-f and Supplementary Fig. 6b in the manuscript). In addition, we selected CRC tissues from patients with KRAS mutations for immunofluorescence co-localization analysis. The results

showed that MYG1 and PKM2 are co-located in the nucleus of CRC patients' tissues (Supplementary Fig. 6c in the manuscript).

Reference

- 1, Yang W et al. PKM2 phosphorylates histone H3 and promotes gene transcription and tumorigenesis. *Cell*. 2012 Aug 17;150(4):685-96.
- 2, Yang W et al. Nuclear PKM2 regulates β -catenin transactivation upon EGFR activation. *Nature*. 2011 Dec 1;480(7375):118-22.

15) With respect to Figure 6K,L – several primer pairs are used to validate c-myc binding in SW620 cell lines. No negative control primer set is used. The authors should discuss why they observe such broad binding over such a large region (~2kb) ... are there 100's of c-myc binding sequences in the MYG1 promoter region? Other cell lines should be similar assessed.

Thank you for your valuable feedback. We understand the importance of including negative controls to ensure the specificity of the observed binding. In our experimental design, we used primers from the MYG1 CDS region as negative controls to confirm the observed specificity of c-Myc binding as transcription factors generally binds to the promoter or 5'UTR. As our results demonstrated, the negative control yielded no binding of c-Myc to the CDS region of MYG1, while showing binding affinity to the promoter region.

To examine the possible binding site of c-Myc, we analyzed the 2000 bases before TTS of MYG1 on JASPAR website. The results showed multiple loci exhibited the potential of binding to c-Myc (**Rebuttal Fig. 3k**; Fig. 6l in the manuscript). We then performed ChIP-qPCR using SW620 and LoVo cells. As the result shown, c-Myc can bind to the MYG1 promoter at multiple sites, mainly at P1 and P6-P11 sequence (**Rebuttal Fig. 3i-j**; Fig. 6m-n and Supplementary Fig. 8i in the manuscript). However, whether c-Myc transcriptionally regulate MYG1 expression was unclear. We next performed double luciferase reporter gene experiment to examine the regulatory sequences. Fragment 1 (F1, from -1926 to -1735 bases) and fragment 2 (F2, from -713 to 0 bases) from MYG1 promoter were used to validate the transcriptional regulation activity and F2 was regulated by c-Myc and PKM2 significantly (**Rebuttal Fig. 3k**; Fig. 6o in the manuscript).

16) With respect to Figure 7A: resolution is not sufficient to observe mitochondrial co-localization ... the protein appears to be mainly nuclear. Mitochondrial localization should be validated with proteinase K shaving assays.

Thank you for your feedback. To address this concern and validate the mitochondrial localization of the protein, we have performed proteinase K shaving assays as suggested. This experiment has provided a more robust and

definitive assessment of the subcellular localization of MYG1 in mitochondria (Fig. 4b and Supplementary Fig. 5a in the manuscript).

Besides, we also conducted co-localization observations on Cyt c and MYG1. We labeled mitochondria with Mitotracker and analyzed the co-localization of MYG1 and Cyt c in mitochondria. (Fig. 7a in the manuscript).

17) With respect to Figure 7C: the magnification presented to not appropriate to observe Cyt c co-localization with MYG1. Similarly, the “obvious” nuclear localization in Figure 8D is not appropriately presented at the provided magnification.

Thank you for your comments. We apologized for the inappropriate magnification presented in Figure 7c and Figure 8. We have revised these figures with suitable magnification (Fig. 7d and Fig. 8d in the manuscript).

18) With respect to Figure 7F,G,H – the authors show that MYG1-deltaN promotes cytochrome C release from mitochondria in panel F, which is counter to the inhibition of apoptosis shown in panels G,H, and the discussion in the text (lines 354-6).

Thank you for your comment. Upon re-evaluation, we have identified an error in the figure presentation, and we apologize for any confusion caused. The panels in question were mislabeled, leading to an inconsistency between the results and the conclusions.

To address this issue, we have repeated the experiments and confirmed our results. MYG1^M inhibited Cyt c release from mitochondria (Fig. 7g in this manuscript). We have revised Figure 7 by correctly labeling the panels. Additionally, we have corrected the associated text and any references to these panels to reflect the revised figure.

Reviewer #2 (Remarks to the Author): expertise in MYG1 biology

In this work the authors address the molecular mechanistic basis of colorectal cancer (CRC) progression. Based on poor prognosis of CRC patients with elevated Myg1 levels, this nucleo-mitochondrial orchestrator emerges as a key orchestrator of CRC progression.

The authors demonstrate that this effect is mediated via the MYC transcription factor and involves alterations in OXPHOS and inhibition of apoptosis. The work is meticulously designed and executed well with appropriate controls for comparison.

Overall I am happy with the work. However, some of the points need to be clarified and adequately addressed to warrant publication.

We thank the reviewer for reviewing our manuscript and providing valuable comments. We greatly appreciate your comments on our work. We have taken the concerns seriously and revised our manuscript in the revision. With your comments, our study will be further improved. The newly added contents are highlighted in the red text and we hope our response sufficiently addresses your concerns. Please note that the figure citations in our response blow refer to the rebuttal figures and the corresponding figures in the revised manuscript.

1. Since Myg1 is a mitochondrial gene it is likely that silencing affects mitochondrial functions and thereby reduces migration, invasion and cell proliferation. In this light the over expression data is important and has been demonstrated. However, if Myg1 accelerates cell cycle, how are the authors sure that observed effects on migration and invasion are not a consequence of altered proliferation?

Thank you for your concern. In the migration and invasion experiments, cells were cultured in serum-free media. Cells were maintained low proliferation activity in this condition. Additionally, the experiments were performed within 24 hours to avoid potential impact on cell proliferation due to prolonged culture. Further details of the experiments have been provided in the Methods section (Page 29, Line 624 in the manuscript).

2. In Fig 2d silencing of MYG1 in LoVo cells is not evident and the same must be quantified.

Thank you for your suggestions. We have revised the results and conducted quantitative analysis (Supplementary Fig. 3 in the manuscript).

3. In Fig 3 the authors compare the MYG1 high and Myg1 low CRCs in TCGA

data. While some of them could be attributed to Myg1 others can be confounding changes not attributable to Myg1 directly. Have the authors performed transcriptomic analysis of the Myg1 OE or silenced CRC cells to ascertain the causality? Also a previous study in Myg1 identified several transcripts altered by Myg1 in melanoma cells (Grover R et al NAR 2019). Have the authors compared these datasets?

Thank you for your suggestion. we performed a CRISPR-Cas9-induced knockout of MYG1 in LoVo cells followed by RNA-seq, and the results indicated its association with metabolic pathways, such as glycolysis (Fig. 3a in the manuscript). The result provided a causality of MYG1 and glycolysis.

Based on your suggestion, we also compared our results with previously published transcriptome data (Grover R et al. NAR 2019). The results showed that genes with abnormal expression in the Myg1 siRNA group in B16 cells did not exhibit a similar expression pattern after knocking out MYG1 in LoVo cells (**Rebuttal Fig. 6**). We speculate that these differences may be due to variations in samples and diverse types of tumors.

Rebuttal Fig. 6. Expression (Log₁₀ count) of 10 genes in sgCtrl and sgMYG1 group in LoVo cells.

To further confirm the consistency of our transcriptome profiling with the previous results, we performed functional enrichment analysis on the differentially expressed genes. We found that apart from being associated with metabolic processes, MYG1 is also involved in innate immune response, developmental processes, and antiviral response. This is consistent with previous transcriptomic findings in HeLa cells ¹, and recent research on the antiviral immune response system also suggests a crucial role of MYG1 in antiviral immunity ².

Reference:

1, Philips MA et al. Characterization of MYG1 gene and protein: subcellular distribution and function. *Biol Cell*. 2009 Jun;101(6):361-73.

2, Chau S et al. Diverse yeast antiviral systems prevent lethal pathogenesis caused by the L-A mycovirus. *Proc Natl Acad Sci U S A*. 2023 Mar 14;120(11):e2208695120.

4. Since silencing of MYC as well as PKM2 both downregulate MYG1, suggestion of the positive feedback loop needs substantiation as it is likely that MYC and PKM2 may be upstream of MYG1. Could the augmented levels of PKM2 upon over expression MYG1 a mere stabilization effect? Or there are alterate regulatory mechanisms present? This needs clarity.

Thank you for your suggestion. We have carried out additional experiments to comprehensively validate our model.

1) To begin with, we studied the regulation mechanism of PKM2 by MYG1. MYG1 can regulate the mRNA and protein levels of PKM2, but it remains unclear whether the regulation of PKM2 by MYG1 depends on this interaction. We have identified the binding sequences between MYG1 and PKM2 and proved the significance of their interaction in glycolysis and tumorigenesis. We also confirmed that the upregulation of PKM2 depends on its interaction with MYG1 (**Rebuttal Fig. 1**). Subsequently, we explored the specific mechanism.

Rebuttal Fig. 1. The interaction of MYG1 and PKM2 is important for glycolysis and tumorigenesis induced by MYG1 in CRC.

By knocking out MYG1 with protein translation inhibited by CHX in LoVo cells, we demonstrated that MYG1 can maintain the protein stability of PKM2 (**Rebuttal Fig. 2a**; Fig. 5i in the manuscript). It is reported that post-translational modifications such as phosphorylation, acetylation, or ubiquitination of PKM2 may affect its stability or degradation ¹. Hence, we propose that MYG1 might influence the post-translational modifications of PKM2. Results from our RNA-seq indicated a potential association between MYG1 and heat shock proteins (**Rebuttal Fig. 2b**; Fig. 3a in the manuscript). Heat shock proteins play crucial roles in cellular process, interacting with newly synthesized proteins and participating in post-translational modifications such as phosphorylation, acetylation, methylation, etc. Through these modifications, heat shock proteins can influence the stability, activity, and interaction of proteins, thereby regulating cellular physiological processes or tumorigenesis ²⁻³. Previous two independent studies have reported the physical interaction between MYG1 and Hsp90 ⁴⁻⁵. These clues led us to speculate that MYG1 might mediate the post-translational modifications of PKM2 by interacting with HSP90, thereby enhancing its stability. Subsequently, we confirmed the interaction between MYG1 and HSP90 through Co-IP, and HSP90 was found to interact with PKM2 as well (**Rebuttal Fig. 2c-d**; Fig. 5j-k in the manuscript).

Xu Q. et al. has reported that HSP90 can recruit GSK3 β to promote PKM2 Thr-328 phosphorylation, increasing its stability and facilitating glycolysis and proliferation in liver cancer cells ⁶. Therefore, we hypothesize that a similar mechanism may exist in CRC, where MYG1 serves as a client protein of HSP90, recruiting it to modify PKM2 and enhance its stability. To investigate this, we validated that GSK3 β can interact with MYG1 and PKM2 (**Rebuttal Fig. 2e-f**; Fig. 5l-m in the manuscript). Finally, we also confirmed that overexpression of MYG1 in HCT116 increased the phosphorylation of PKM2 (**Rebuttal Fig. 2g**; Fig. 5n in the manuscript). Knocking down the level of HSP90 in LoVo cells led to a decrease in PKM2 phosphorylation (**Rebuttal Fig. 2h**; Fig. 5o in the manuscript). Treatment with GSK3 inhibitor IX, a GSK3 β inhibitor, also reduced the phosphorylation of PKM2 (**Rebuttal Fig. 2g**; Fig. 5n in the manuscript).

These findings suggest that MYG1 facilitates the phosphorylation of PKM2 through recruits HSP90/GSK3 β complex, enhancing its stability and thereby regulating its protein level.

Rebuttal Fig. 2. MYG1 recruits HSP90/GSK3 β complex and enhances the stability of PKM2.

2) Nuclear PKM2 transcriptionally regulate MYC expression. Nuclear PKM2 can regulate glycolysis by acting as a kinase to regulate downstream key transcription factors, or directly interacting with transcription factors to regulate transcription in cancer ¹. GSEA analysis of TCGA CRC datasets revealed the association of MYG1 with the MYC targets pathway (**Rebuttal Fig. 3a**; Fig. 6g in the manuscript). Studies have reported that PKM2 can interact with β -catenin to promote MYC transcription and thereby regulate key enzymes in the glycolysis process ⁷. Therefore, we speculate that MYG1 may activate MYC through PKM2 in CRC. Firstly, we established the correlation between MYG1, PKM2, and MYC expression in cell lines (**Rebuttal Fig. 3b**; Supplementary Fig. 8g in the manuscript). Additionally, we found that nuclear MYG1 can enhance MYC protein expression (**Rebuttal Fig. 3c**; Fig. 6h in the manuscript).

3) c-Myc binds to MYG1 promoter and promotes MYG1 transcription. Interestingly, when we knocked down PKM2, we observed a downregulation not only in c-Myc expression but also in MYG1 protein levels (**Rebuttal Fig. 3d**; Fig. 6i in the manuscript). This led us to consider the existence of a feedback regulatory mechanism, where Myc may regulate MYG1 transcription. Initially, we knocked down and overexpressed Myc and found that Myc can regulate MYG1 mRNA levels (**Rebuttal Fig. 3e-f**; Fig. 6j-k in the manuscript). We next analyzed whether c-Myc could bind to MYG1 promoter using online ChIP-seq

dataset. The results showed an enrichment of a peak before the transcriptional start site (TTS) of MYG1 in several CRC cell lines (**Rebuttal Fig. 3g**; Supplementary Fig. 8h in the manuscript). To examine the possible binding site of c-Myc, we analyzed the 2000 bases before TSS of MYG1 on JASPER website. The results showed multiple loci exhibited the potential of binding to c-Myc (**Rebuttal Fig. 3h**; Fig. 6l in the manuscript). We then performed ChIP-qPCR using SW620 and LoVo cells. As the result shown, c-Myc can bind to the MYG1 promoter at multiple sites, mainly at P1 and P6-P11 sequence (**Rebuttal Fig. 3i-j**; Fig. 6m-n and Supplementary Fig. 8i in the manuscript). However, whether c-Myc transcriptionally regulate MYG1 expression was unclear. We next performed double luciferase reporter gene experiment to examine the regulatory sequences. Fragment 1 (F1, from -1926 to -1735 bases) and fragment 2 (F2, from -713 to 0 bases) from MYG1 promoter were used to validate the transcriptional regulation activity and F2 was regulated by c-Myc and PKM2 significantly (**Rebuttal Fig. 3k**; Fig. 6o in the manuscript).

In summary, these experiments successfully confirmed the presence of a positive feedback regulatory loop, where MYG1 regulates PKM2 and MYC expression, while Myc, conversely, promotes MYG1 transcription.

References:

- 1, Zhang Z et al. PKM2, function and expression and regulation. *Cell Biosci.* 2019 Jun 26;9:52.
- 2, Wu J et al. Heat Shock Proteins and Cancer. *Trends Pharmacol Sci.* 2017 Mar;38(3):226-256.
- 3, Hu C et al. Heat shock proteins: Biological functions, pathological roles, and therapeutic opportunities. *MedComm (2020).* 2022 Aug 2;3(3):e161.
- 4, Falsone S.F. et al. A proteomic snapshot of the human heat shock protein 90 interactome. *FEBS Lett.* 2005 Nov 21;579(28):6350-4.
- 5, Millson SH, et al. A two-hybrid screen of the yeast proteome for Hsp90 interactors uncovers a novel Hsp90 chaperone requirement in the activity of a stress-activated mitogen-activated protein kinase, Slt2p (Mpk1p). *Eukaryot Cell.* 2005 May;4(5):849-60.
- 6, Xu Q, et al. HSP90 promotes cell glycolysis, proliferation and inhibits apoptosis by regulating PKM2 abundance via Thr-328 phosphorylation in hepatocellular carcinoma. *Mol Cancer.* 2017 Dec 20;16(1):178.
- 7, Yang W et al. Nuclear PKM2 regulates β -catenin transactivation upon EGFR activation. *Nature.* 2011 Dec 1;480(7375):118-22.

Rebuttal Fig. 3. c-Myc binds to MYG1 promoter and promotes MYG1 transcription.

5. While the nuclear form of MYG1 seems to be important and mechanistic support of its involvement is clear, the involvement of the mitochondrial form is not very clear and needs to be elaborated.

Thank you for your feedback, and based on your suggestions as well as those of other reviewers, we have further investigated the mechanism of nuclear MYG1. Considering that mitochondria play a relatively minor role, we have included this aspect in our research paper to ensure the integrity of the study and only studied the function of inhibiting apoptosis and OXPHOS. However, due to limited understanding of its specific functions, we did not conduct in-depth research on the specific mechanism of its action, which requires further exploration in the future.

6. From the study the role of MYG1 as an exonuclease is not explored. Do the authors think that DHH mutant would be unable to recapitulate the effects of the nuclear MYG1? Also previous work seems to suggest that the nuclear role of Myg1 is two-fold, it regulates NE mito genes and alters ribosomal RNA processing and eventually controls ribosomal assembly and the overall translation in the cell.

How do the authors interpret their results in this context?

Thank you very much for your comments. Inspired by your feedback, we investigated the exonuclease function of MYG1 in CRC. Building on previous studies, we mutated the DHH amino acid sequence of MYG1 to ALL (MYG1^{ALL}), resulting in a loss of RNA 3' to 5' exonuclease activity. Interestingly, functional studies demonstrated that the loss of enzymatic activity of MYG1 also exhibited the impact on CRC biology (Fig. 2e-h and Supplementary Fig. 2e-h in the manuscript) and glycolysis of CRC cells (Fig. 3g-i in the manuscript). These findings suggested that MYG1's function does not primarily depend on its exonuclease activity in CRC, which means that it mainly does not rely on its role in processing RNA.

Previous studies in yeast have indicated that Myg1 maintains oxidative phosphorylation and controls mitochondrial function by modifying RNA or regulating RNA transcription. However, its function in CRC does not mainly rely on its enzymatic activity. Based on this, we propose that MYG1 may exert its role in metabolic reprogramming of tumor cells through other methods such as protein-protein interaction. In this study, we elucidated the mechanism by which MYG1 regulates CRC glycolysis by modulating PKM2 through a protein-interaction network. This interaction network is likely to be an important mechanism through which MYG1 functions in CRC.

In fact, enzyme activity independent functions of various nucleases have been reported in previous studies. For example, XRN1, an essential 3'-5' exonuclease involved in mRNA degradation, has recently been found to act as a transcriptional activator. Xrn1, together with other components of the deadenylation-dependent mRNA decay pathway, shuttles between the cytoplasm and the nucleus, where they bind to transcription start sites and directly stimulate transcription initiation and elongation of many yeast genes ¹. Hence, we believe that the main role of MYG1 in tumors is not exerted through the regulation of RNA processing and transcription.

Reference:

1, Blasco-Moreno B, et al. The exonuclease Xrn1 activates transcription and translation of mRNAs encoding membrane proteins. *Nat Commun.* 2019 Mar 21;10(1):1298.

Reviewer #3 (Remarks to the Author): expertise in colorectal cancer metabolism

In the current manuscript, the authors report that melanocyte proliferating gene 1 (MYG1) can regulate glycolysis and promote CRC progression by coordinating its nuclear-mitochondrial functions. They proposed that nuclear MYG1 forms a complex with PKM2 dimer and induces the accumulation of PKM2 in the nucleus. Nuclear PKM2 transcriptionally activates MYC and promotes MYC-mediated glycolysis. c-Myc also transcriptionally upregulates MYG1, forming a positive feedback loop. They also proposed that mitochondrial MYG1 inhibits apoptosis and interferes with OXPHOS.

We thank the reviewer for reviewing our manuscript and providing valuable comments. We greatly appreciate your comments on our work. We have taken the concerns seriously and revised our manuscript in the revision. With your comments, our study will be further improved. The newly added contents are highlighted in the red text and we hope our response sufficiently addresses your concerns. Please note that the figure citations in our response now refer to the rebuttal figures and the corresponding figures in the revised manuscript.

There are a few critical comments:

1. In the introduction, the authors talked about “approximately 900 000 individuals die from this malignancy”. But it is not clear how long and where this number is related to.

Thank you for your feedback on our introduction. The data mentioned in the introduction is derived from the Global Cancer Burden Assessment of the International Agency for Research on Cancer (IARC) in 2020. According to the data, the number of deaths due to colorectal cancer in 2020 was 935,173, accounting for 9.4% of all cancer cases. We have revised the manuscript and provided a clearer explanation of the mentioned statistic (Page 3, Line 50 in the manuscript).

2. Statistical analysis is missing for Figure 1C.

Thank you for your reminding. We performed paired-sample t-test on the data for relative quantification of western blot. The results indicate that MYG1 is highly expressed in CRC tissues compared to adjacent normal intestinal mucosa. The quantification results were added in the revision (Fig. 1b in the manuscript).

3. The statement that “MYG1 is upregulated in the genetic background of APC loss and KRAS mutation in CRC” is not convincing. Figure 1G showed that SW480 cells have mutations in APC and KRAS, but MYG1 expression was very

low. Moreover, RKO cells do not have mutations in APC and KRAS, but MYG1 expression was very high.

Thank you for bringing up this concern regarding the statement. Having carefully considered the reviewer's feedback, we agree that our previous statement may not be fully supported by the data provided.

Indeed, we aimed to investigate the reasons behind the upregulation of MYG1 in cancer. Copy number variants analysis of TCGA COADREAD data revealed that the MYG1 gene copy number remained unchanged in the majority of CRC patients. This led us to consider the possibility of key gene regulation driving CRC progression. Statistical analysis of our cell line results showed that MYG1 protein expression was upregulated in cells with KRAS mutations compared to wild-type KRAS cells (Fig. 1f and Supplementary Fig. 1g in the manuscript). Furthermore, our data from mice also indicated a significant transcriptional upregulation of MYG1 following APC and KRAS mutations (Supplementary Fig. 1f in the manuscript). To further validate these findings in human CRC samples, we first analyzed the relationship between MYG1 and APC/KRAS mutations in CRC patients from the TCGA database. The results revealed a significant upregulation of MYG1 levels in patients with KRAS mutations (Fig. 1g in the manuscript). Additionally, we confirmed this conclusion through immunohistochemical analysis in a NF-KRAS cohort. The results showed that MYG1 expression was higher in patients with KRAS mutation compared with wild type patients (Fig. 1h in the manuscript).

Combining our findings on the mechanism of MYG1's function, we discovered that MYC plays a crucial role in regulating MYG1 expression. MYC is also considered one of the downstream effectors of the KRAS signaling pathway. KRAS mutations can activate the PI3K/AKT and MAPK pathways, leading to MYC transcription activation and promoting cell proliferation and growth. Furthermore, KRAS mutations play a crucial role in glycolysis regulation and metabolic reprogramming. Thus, KRAS mutations might be a potential driver for MYG1 promoting glycolysis.

4. It is very confusing that "high expression of MYG1 harms the progression and survival of CRC patients".

Thank you for pointing out the confusion in our statement. What we would like to express is that high expression of MYG1 is associated with poor outcome in CRC patients. We have revised the manuscript (Page 7, Line 136 in the manuscript) and sincerely apologize for any confusion caused.

5. It is not clear what is "MYG1 accelerated G1/S cells conversion in CRC cells".

Thank you for highlighting the lack of clarity in our statement. To address this concern, we have revised the manuscript to provide a clearer statement. We have changed the sentences into “MYG1 accelerates the G1-S cell cycle transition” (Page 9, Line 193 in the manuscript).

6. The quantification results for figure 2D and Figure S3 are lacking. The quality for Figure S3 needs to be improved as well.

Thank you for your feedback regarding the quantification and the quality of the results. To address this concern, we have included the necessary quantification data for Figure 2D (previous version). We have also provided accurate measurements and statistical analysis to support the conclusions drawn from the experiment (Supplementary Fig. 3 in the manuscript).

Regarding Figure S3, we acknowledge that the quality may have been insufficient in the previous version of the manuscript. In the revised version, we have repeated the IF assays to improve the visual presentation and ensure that the figure is clear and easily interpretable. Besides, we also performed the quantification for IF (Fig. 2p-q in the manuscript).

7. “Next, we confirmed that MYG1 promoted the expression of GLUT1, LDHA, and PKM2 using qPCR (Supplementary Fig. S4B).” Why only these three genes were selected? Why not GAPDH?

Thank you for comment. Based on the potential functions of MYG1 on glycolysis, we firstly analyzed the correlation of MYG1 and key genes involved in glycolysis in TCGA COADREAD cohort (Supplementary Fig. 3b in the manuscript). Most of genes correlated with MYG1 was regulated by c-Myc and HIF. We next evaluated the effect of MYG1 on the glycolysis-related genes that are highly correlated with MYG1 in HCT116 cells. The results showed that MYG1 can regulate the expression of LDHA, GLUT1, and PKM2 (Fig. 3b in the manuscript). GLUT1, LDHA, and PKM2 are well-established metabolic genes that have been implicated in cancer progression, including CRC. We have selected these genes for further study. In the analysis, GAPDH was also analyzed and the results showed that MYG1 did not regulate the expression of GAPDH (Fig. 3b in the manuscript).

8. Quantification and statistical analysis are preferred for Fig. 3D-E.

Thanks for your comment. We have supplied the quantification and statistical analysis for Fig. 3D-E (Fig. 3c-f and Supplementary Fig. 3d-g in the manuscript).

9. Figure 4A, “Interestingly, MYG1 was also detected in the cytoplasm with the

mitochondria removed (C-M) with a smaller molecular weight.” Which is the right band for MYG1? It seems that there are no gel shifts in RKO cells.

Thank you for your feedback. We have retested the expression of MYG1 in different cellular fractions and have replaced the blurry bands. Our previous bands with different size were detected by Western blot using anti-MYG1 antibodies from MilliporeSigma (HPA038627). In the revision, we used anti-MYG1 antibody from Abcam (ab204420) and no gel shift was observed. We have replaced the bands of MYG1 with the new one.

Additionally, western blot was performed using the anti-MYG1 antibody from Abcam (ab204420) in our revision, and the bands of MYG1 was validated by KO and KD experiments.

10. Where is the construct with both signal peptide deleted (MYG1 Δ L) in figure 4C?

Thanks for your comment. We have supplied the western blot of MYG1 Δ L and validated its expression using the anti-Flag antibody (Supplementary Fig. 5c in the manuscript).

11. It is also not convincing that the construct for nuclear localization peptide deleted (MYG1 Δ N) is working based on the results of Figure S5A.

Thanks for your concern. We have designed C-terminally Flag-tagged MYG1 (wild-type) and MYG1 variants in the revision. Then we validated their expression and localization using western blot and immunofluorescence (IF) experiments (**Rebuttal Fig. 5a-c**; Fig. 4d-e and Supplementary Fig. 5c in the manuscript). All variants were successfully expressed and localized as intended.

Rebuttal Fig. 5. Schematic of MYG1 domain and variant constructs and variants were validated by western blot and IF.

12. Again, the results in figure 4d is not convincing.

Thanks for your comments. Based on your and other reviewers' suggestions, we have constructed the MYG1 variants and validated by western blot and IF (**Rebuttal Fig. 5a-c**; Fig. 4d-e and Supplementary Fig. 5c in the manuscript).

13. "Although the MYG1^{ΔN} tumor showed no statistical difference in tumor size, the average volume and luciferin signal were also higher than those in the control group." This sentence is confusing.

Thank you for your feedback. We have revised the description and interpretation of the experimental results (Page 13 in the manuscript). Unlike the in vitro experiments, the in vivo experiments showed no significant difference in tumor volume between the MYG1^M group and the control group. This may be due to the weaker effect of MYG1^M on tumor, the small sample size, and the influence of various complex factors in vivo.

14. "In addition, MYG1^{ΔM} mice also had advanced tumor cell invasion and intraperitoneal dissemination of tumors compared with the MYG1^{ΔN} group (Supplementary Fig.S5C-D)." It is hard for the reviewer to appreciate these results.

Thank you for your feedback. We have re-evaluated the data and described our results with additional statistical analysis (Page 13, and Supplementary Fig. 5d-e in the manuscript).

15. "The transwell invasion assay showed that cells in MYG1^{ΔM} group were more aggressive (Supplementary Fig. S5G)." It seems that the WT MYG1 had the greatest effect. It may be worth to include data from MYG1^{ΔL}.

Thank you for your suggestion. It is reasonable that the WT MYG1 group exhibited the highest invasive ability due to the synergistic effect of MYG1^M and MYG1^N. Taking your advice into consideration, we have included the MYG1^{ΔL} group in our study. The results showed that the invasive ability of the MYG1^{ΔL} cells did not show significant changes compared to the control group (Supplementary Fig. 5g-h in the manuscript).

16. "Among 180 candidate proteins in the nucleus (Supplementary table S3), we focused on PKM2, a key enzyme that catalyze the conversion of phosphoenolpyruvate to pyruvate." Only PKM1 was found in the table. Any explanation?

Thank you for your question. Among the 180 candidate molecules, PKM (Pyruvate Kinase M), not PKM1, was identified. PKM1 and PKM2 are two isoforms encoded by the same gene PKM. The choice of studying PKM1 or PKM2 depends on the specific questions and research objectives we address. PKM1 and PKM2 have distinct functions and regulatory mechanisms in physiological and pathological processes. It is generally believed that PKM1 is primarily present in mature tissues such as heart muscle and skeletal muscle, where it participates in the process of glycolysis and plays an important role in maintaining normal cellular metabolism. In contrast, PKM2 is widely expressed in embryonic development, tumor cells, and other abnormal cell growth, and is associated with tumor cell glycolysis, growth, and metabolic reprogramming. PKM2 exhibits different activity and regulatory mechanisms compared to PKM1 and is considered a key regulator of tumor cell growth and metabolic pathways.

Therefore, our research goal is to explore the role of PKM in tumor development, particularly in the context of tumor metabolism, studying PKM2 may be more suitable. The choice of which isoform to study is based on the specific research questions and objectives.

17. Figure 5, the majority of PKM2 locates in the cytosol. The reason for the specificity of PKM2 and MYG1 interaction in the nucleus needs to be discussed.

Thanks for your comments. To address these concerns, we have improved our study design and imaging techniques. As our results show, PKM2 is primarily distributed in the cytoplasm and to a lesser extent in the nucleus. Studies have reported that EGF can activate RAS/RAF signals to promote PKM2 phosphorylation and enter the nucleus, and this is required for tumor development¹⁻². In the revision, we proved that MYG1 is upregulated in patients with KRAS mutations. Sustained activation of KRAS promotes nuclear translocation of PKM2 and enhanced the downstream c-Myc mediated glycolysis. In this scenario, increased interaction between MYG1 and PKM2 plays an important role in promoting glycolysis in CRC.

Based on your considerations, we therefore tested the relationship between MYG1 and PKM2 under EGF stimulation and found that the interaction of MYG1 and PKM2 was increased. However, the effect of EGF was limited in LoVo cells with sustained activation of KRAS (Fig. 5a-f and Supplementary Fig. 6b in the manuscript). In addition, we selected CRC tissues from patients with KRAS mutations for immunofluorescence co-localization analysis. The results showed that MYG1 and PKM2 are co-located in the nucleus of CRC patients' tissues (Supplementary Fig. 6c in the manuscript).

Reference

1, Yang W et al. PKM2 phosphorylates histone H3 and promotes gene transcription and tumorigenesis. Cell. 2012 Aug 17;150(4):685-96.

2, Yang W et al. Nuclear PKM2 regulates β -catenin transactivation upon EGFR activation. Nature. 2011 Dec 1;480(7375):118-22.

18. Grammar and spelling errors need to be carefully corrected. Eg. Line 258, 266, 295, 349, 382, 573, 730.

Thank you for your suggestion. We have made the correction in the respective sections accordingly.

19. How many times the authors have repeated for figure 5D? Statistical analysis is needed.

Thank you for your reminder. We have repeated three times for figure 5D and included quantification (Fig. 5h in the manuscript).

20. The labeling for constructs for figure 5e should be consistent with others in the manuscript.

Thank you for your reminder. We have made the change.

21. "The results showed that PKM2 and c-Myc were highly expressed in CRC cell lines compared with FHC, and the expression of PKM2, c-Myc, and MYG1 was highly correlated (Supplementary Fig. S7A)." This statement needs statistical analysis support.

Thank you for your reminder. We have included quantification and statistical analysis (Supplementary Fig. 8g in the manuscript).

22. Figure 6C, which is the right band for c-Myc?

Thank you for your suggestion. We carefully compared the molecular weights of c-Myc and cropped the appropriate bands for display. As shown right, there are two bands of c-Myc around 50 kDa. Both two bands are c-Myc as it was knocked down by siRNA.

23. Figure 6D, why suddenly switched to Caco-2 cells? No quantification?

We apologize for any confusion caused. Due to the low expression levels of PKM2 and MYG1 in SW480 and CACO2 cells, we overexpressed PKM2 in these two cell lines to study its regulatory effects on other proteins. Since the great changes of our study, we have removed these results in the revision.

24. Figure S7E, were the changes statistically significant?

Thank you for your query. Due to the inappropriate concentration of C3K chosen during our experiment, we have re-designed and conducted relevant experiments. In order to clarify this issue, we have selected different concentrations of C3K gradients for experiments. The specific purpose of the experiment and the interpretation of the results are explained in detail in the subsequent question 29.

25. Figure 6K-I, where are the exact binding sites for c-Myc in MYG1 promoter? Any Myc response element? Figure 6L is not shown.

Thanks for your concern. To examine the possible binding site of c-Myc, we analyzed the 2000 bases before transcriptional start site of MYG1 on JASPAR website. The results showed multiple loci exhibited the potential of binding to c-Myc (Fig. 6I in the manuscript). We then performed ChIP-qPCR using SW620 and LoVo cells. As the result shown, c-Myc can bind to the MYG1 promoter at multiple sites, mainly at P1 and P6-P11 sequence (Fig. 6m-n and Supplementary Fig. 8i in the manuscript).

26. This statement "The result showed that cyt c was obvious downregulated in cells expressing MYG1 Δ WT and MYG1 Δ N (Fig. 7D)." lacks statistical analysis support.

Thank you very much for your suggestion. We have performed the quantification of the western blot (Fig. 7e in the manuscript).

27. This statement "MYG1 Δ N can downregulate cyt c and inhibit the OXPHOS of cells, while it can also inhibit the release of cyt c from the mitochondria and inhibit the apoptosis of CRC cells" conflicts with the results in Figure 7F at least.

Thank you for pointing out our mistake. The previous results were mislabeled. We have repeated the experiment and reorganized the results. Our results indicated that MYG1 inhibits the release of Cyt c from mitochondria to the cytoplasm (Fig. 7g in the manuscript).

28. Figure 8d, apoptosis should be examined to verify the function of mitochondrial MYG1.

Thank you very much for your suggestion. It is crucial to validate the function of MYG1 in animal models. We performed immunofluorescence staining on paraffin-embedded mouse tumors and quantitatively analyzed the percentage of apoptotic cells using the TUNEL assay. The results indicated that MYG1

plays a crucial role in inhibiting tumor cell apoptosis in the tumors (Fig. 8d-e in the manuscript).

29. In figure 6E, why C3K inhibits PKM2 but increased the expression of c-Myc and MYG1? This conflicts with Figure 6B. The discussion should move to the results section.

Thanks for your comments. In previous experiments, the use of inappropriate concentrations of compound 3K (C3K) resulted in confused experimental results. In the revised manuscript, we conducted a series of experiments using C3K to confirm that MYG1 regulates glycolysis independently of PKM2 enzyme activity. The corresponding conclusions have been presented in the paper.

Compound 3K is a potent PKM2 enzyme activity inhibitor identified by Xianling Ning et al., which inhibits PKM2 activation by blocking its binding with fructose-1,6-bisphosphate (FBP) ¹. The IC₅₀ range of compound 3K varies from 0.18-1.56 μM in different cell lines. To investigate whether inhibiting PKM2 enzyme activity affects the function of MYG1, we treated LoVo and SW620 CRC cell lines with different concentrations of C3K (0 μM, 1 μM, 2 μM, and 4 μM) and confirmed the inhibition of PKM2 enzyme activity through pyruvate kinase activity detection (Supplementary Fig. 8a in the manuscript). As literature suggests that the use of C3K may alter the nuclear-cytoplasmic distribution of PKM2 ², we also examined the distribution of PKM2 using immunofluorescence (IF) and quantitatively found no significant changes in the overall expression levels and nuclear-cytoplasmic distribution of PKM2 (Supplementary Fig. 8b-c in the manuscript). Furthermore, we treated cells overexpressing MYG1 with C3K and performed western blot analysis. The results showed that C3K did not affect the expression levels of PKM2 and MYG1 (Supplementary Fig. 8d in the manuscript). We also detected the invasion and proliferation ability of HCT116 cells with C3K treated. The results showed that C3K treatment did not influence the function of MYG1^N compared with EV (Supplementary Fig. 8e in the manuscript). These results suggest that the function of MYG1 in promoting glycolysis is not mediated through the regulation of PKM2 enzyme activity, but through the transcriptional regulation of PKM2.

We have described these assays in the Results section and the results have been thoroughly explained.

Reference:

1, Ning X, et al. Discovery of novel naphthoquinone derivatives as inhibitors of the tumor cell specific M2 isoform of pyruvate kinase. *Eur J Med Chem.* 2017 Sep 29;138:343-352.

2, Guo J, et al. PKM2 suppresses osteogenesis and facilitates adipogenesis by regulating β -catenin signaling and mitochondrial fusion and fission. *Aging* (Albany NY). 2020 Feb 25;12(4):3976-3992.

30. Line 479-480, Cells (1x 10⁶) cells were subcutaneously injected into the splenic capsules of mice?

Thank you for pointing out the error in our paper. We have revised the sentences to “Cells (1 × 10⁶) were injected into the splenic capsules of mice” (Page 30, Line 655 in the manuscript).

31. Supplementary information is available at Cell death & Differentiation’s website?

Thanks for pointing our mistake. We have separately uploaded the supplementary materials. The description of supplementary information was revised in the manuscript.

Reviewer #4 (Remarks to the Author): expertise in colorectal cancer cell biology

In this manuscript Chen and colleagues describe how MYG1 regulates tumor aggressiveness in CRC. They show that MYG1 is higher expressed in CRC, associated with prognosis and stage and that loss of MYG1 dampens and gain of MYG1 increases migration proliferation, invasion and in vivo metastasis. They continue to show that MYG1 requires its nuclear and not its mitochondrial localization for this effect and show that MYG1 is regulated by PKM2 and MYC. Importantly, MYG1 dampens respiration and enhances glycolysis.

Although the data are of interest they require a lot more validation in relation to earlier data with MYG1. The findings of the authors maybe true and tissue specific but to choose to ignore the 180 degree different observations in LUAD, skin and yeast cells is not the way forward.

Thank you for your interest in our manuscript and for providing valuable feedback. We are grateful for your constructive comments and suggestions. Indeed, we acknowledge the importance of comparing our results with earlier data as you concern. To address this, we have conducted further experiments to validate our findings and compared our results with previous studies. We value the importance of this step and will ensure that our revised manuscript includes a comprehensive comparison with the relevant existing literatures.

Once again, we would like to express our gratitude for your feedback. We believe that incorporating your suggestions will significantly improve the quality and impact of our research. The newly added contents are highlighted in the red text and we hope our response sufficiently addresses your concerns. Thank you for your time and consideration. Please note that the figure citations in our response blow refer to the rebuttal figures and the corresponding figures in the revised manuscript.

1. Others have provided compelling evidence that loss of MYG1 decreases respiration, while gain of expression as seen in LUAD is associated with enhanced oxidative phosphorylation. The authors will need to explain these differences and study their effects in relation to the observed exonuclease activity of MYG1. Maybe the difference is in nuclear versus mitochondrial but they need to address these differences experimentally. If the want to claim this role in CRC they need to show that under the exact same expression levels the role is different in LUAD. If proven then it would also make sense to look for the reason as Myc is equally important in LUAD.

We thank the review for this comment. We have now performed a serious of experiments and added data to explain the differences you are concerned about.

1) MYG1 promotes CRC glycolysis and progression independent of its endonuclease activity. Ritika Grover et al. identified MYG1 as a new RNA endonuclease with 3' to 5' activity. MYG1 plays a crucial role in regulating mitochondrial function. Their study revealed three functions of MYG1: 1) maturing 18S rRNA by cleaving at the ITS-1 region, 2) cleaving nuclear-encoded NeMito mRNA in the cell nucleus, and 3) trimming mitochondrial-encoded RNA. All of the functions of MYG1 in yeast are depend on its RNA endonuclease activity. And they concluded that MYG1's dual localization is essential for the function of mitochondria. Considering the reviewer's suggestion, we investigated whether the function of MYG1 was depend on its RNA endonuclease activity. We generated an RNA endonuclease activity inactive MYG1 mutant by replacing the DHH residue with ALL (MYG1^{ALL}) as previously described ¹. Upon wild type MYG1 and MYG1^{ALL} expression at comparable levels in CRC cells, MYG1^{ALL} promote cell growth, migration and invasion , although to a slightly lesser extent then that of MYG1, implying that MYG1 has endonuclease-independent cellular function (Fig. 2e-h and Supplementary Fig. 2e-h in the manuscript).

Based on this interesting discovery, we have revealed a molecular regulatory network through studying the protein interaction with MYG1, which promotes metabolic remodeling and CRC progression in coordination with the nucleus and mitochondria. In our research, we found that MYG1 can enhance glycolysis in CRC cells, and MYG1^{ALL} exhibits consistent phenotypes with the wild type (Fig. 3g-i in the manuscript). Our finding uncovers a novel role of MYG1 in CRC metabolic reprogramming: under physiological conditions, MYG1 relies on its endonuclease activity to maintain mitochondrial function and promotes cellular respiration. However, in CRC, MYG1 regulates aerobic glycolysis independent of its endonuclease activity, thereby meeting the increased energy and material demands for rapid proliferation and metastasis of CRC cells.

2) MYG1 plays different roles in CRC and LUAD owing to different regulation mechanisms. In addition to our research, Xiaodan Han et al. have preliminarily revealed that MYG1 can promote proliferation of lung cancer cells and inhibits autophagy through the AMPK/mTOR complex 1 signaling. They found that MYG1 may correlate with oxidative phosphorylation (OXPHOS) and can enhance the production of ATP in lung cancer cells ². Interestingly, their finding differs from our research in CRC. To better explain this discrepancy, we examined the impact of MYG1 on glucose uptake and lactate secretion in lung adenocarcinoma cell line PC9 in the similar conditions. The results demonstrated that MYG1^{ALL} inhibited lactate secretion and glucose uptake in PC9 cells, exhibiting similar behavior to the wild-type MYG1 (Supplementary Fig. 4h-i in the manuscript). These results suggest that MYG1 suppresses glycolysis in lung cancer cells. As KRAS also exhibits a high mutation rate in

lung adenocarcinoma. Subsequently, we explored the relationship between KRAS mutations and MYG1 expression. The results revealed that lung adenocarcinoma patients with KRAS mutations showed comparable levels of MYG1 expression compared to patients with wild-type KRAS (Supplementary Fig. 4j in the manuscript). While distinct relationship between KRAS and MYG1 can partially explain the differential impact of MYG1 on glycolysis in colorectal and lung cancers, we are indeed intrigued by how MYG1 functions differently in lung cancer.

Continuing our research, we investigated the roles of nuclear and mitochondrial MYG1 in glycolysis in PC9 cells. We detected the effects of MYG1 variants on glucose uptake and lactate secretion. These results indicated that MYG1^M predominantly inhibits glycolysis in PC9 cells (Supplementary Fig. 5k in the manuscript). Based on this, we speculate that the difference of subcellular compartment dependence may contribute to the contrasting phenotypes observed in CRC and lung adenocarcinoma. Since MYG1 may have a completely different functional pattern in LUAD, the mechanism of MYG1 function is likely to be distinct as well. Therefore, exploring the mechanisms of MYG1 in LUAD, as well as unraveling the regulatory mechanisms of MYG1 in different tumors, will be a future challenge that requires further work and time to address.

Thank you once again for your constructive comments. We believe that we have addressed your concerns through experiments and analysis, and we have incorporated the corresponding content and discussions in the revision. The references cited in this response are listed below.

References

- 1, Grover R, et al. Myg1 exonuclease couples the nuclear and mitochondrial translational programs through RNA processing. *Nucleic Acids Res.* 2019 Jun 20;47(11):5852-5866.
- 2, Han X, et al. MYG1 promotes proliferation and inhibits autophagy in lung adenocarcinoma cells via the AMPK/mTOR complex 1 signaling pathway. *Oncol Lett.* 2021 Apr;21(4):334.

2. The authors start with the overexpression data of MYG1, but the data shown do not show a highly convincing impact of transformation. A lot of tumors by RNA or western do not show overexpression at all. What is the underlying reason? The authors suggest that this would be APC Kras but this is also not very solid data. The mice data are marginal higher expressed at best and the human data appear overinterpreted (RKO is a very strong expressor while SW480 has no overexpression). The authors should analyse the databases which often have mutation data associated to it (TCGA) or use subtype

stratification.

Thank you for your valuable feedback and suggestions. To demonstrate the impact of MYG1 during the transformation of CRC, we collected a NF-CRC2 cohort to investigate the expression of MYG1 at different stages of colorectal carcinogenesis. The results showed a gradual increase in MYG1 expression from normal mucosa to adenomas and further progression to carcinoma. This finding, combined with our other data, suggests that MYG1 plays a significant role in the evolution of colorectal cancer (Supplementary Fig. 1d-e in the manuscript).

Some tumors do not show overexpression of MYG1 as detected by qPCR, because cancer are heterogeneous diseases, exhibiting extensive heterogeneity not only between different patients with the same cancer type but also within different cells of an individual patient. Our study provides an overall perspective, demonstrating that in a large cohort of CRC patients, MYG1 is upregulated in tumor tissues when considering the tumor as a whole entity. However, we acknowledge that during experimental procedures, the tumor and its microenvironment were treated as a unified whole, which may have included other cell types. To highlight the differences in MYG1 expression between tumor cells and stromal cells, we have validated our findings through immunohistochemistry in tissue samples and cell line experiments. Indeed, it is necessary to investigate the expression differences of MYG1 in different CRC patients, as it provides implications for personalized treatment. With this in mind, we have explored the factors leading to the upregulation of MYG1.

Firstly, copy number variants analysis of TCGA COADREAD data revealed that the MYG1 gene copy number remained unchanged in the majority of CRC patients. This led us to consider the possibility of key gene regulation driving CRC progression. Statistical analysis of our cell line results showed that MYG1 protein expression was upregulated in cells with KRAS mutations compared to wild-type KRAS cells (Fig. 1f and Supplementary Fig. 1g in the manuscript). Furthermore, our data from mice also indicated a significant transcriptional upregulation of MYG1 after APC loss and KRAS mutations (Supplementary Fig. 1f in the manuscript). To further validate these findings in human CRC samples, we analyzed the relationship between MYG1 and APC/KRAS mutations in CRC patients from the TCGA database. The results revealed a significant upregulation of MYG1 levels in patients with KRAS mutations (Fig. 1g in the manuscript). Additionally, we confirmed this conclusion through immunohistochemical analysis in a NF-KRAS cohort. The results showed that MYG1 expression was higher in patients with KRAS mutation compared with KRAS wild type patients (Fig. 1h in the manuscript).

3. The authors should also increase the survival analysis as the association

with poor prognosis is not detected in publicly available sets.

Thanks for your suggestion. We conducted survival analysis based on the TCGA COADREAD cohort to investigate the relationship between MYG1 and survival (overall survival rate and disease-free survival rate within five years). We found that high expression of MYG1 was associated with poor prognosis in CRC patients (Supplementary Fig. 1c in the manuscript).

4. The authors show that the glucose uptake is dependent on nuclear localization of MYG1 but surprisingly the lactate production seems less or not affected if nuclear localization is blocked. This needs further substantiation. Same for the in vivo PET analysis which to this reviewer shows that the PET activity measured is identical in all 4 conditions. If the authors would have normalized the signal to the tumor volume there is no difference left. This implies that all tumor cells are as active in glucose uptake which is not consistent with the in vitro data. They clearly grow differently but do not take up glucose at a different rate.

Thanks for your comments. Based on your concerns, we re-evaluated the impact of MYG1 with different subcellular localizations on glucose and lactate secretion in CRC cells. The results indicated that nuclear-localized MYG1 plays a more critical role in promoting the glycolytic phenotype, while mitochondrial-localized MYG1 also has a negligible effect (Fig. 4f in the manuscript). Overall, nuclear-localized MYG1 dominates in the regulation of glycolysis.

Besides, we thank the review for the comment of PET analysis. We agree that it is necessary to standardize tumor volume and individual differences when assessing the glucose uptake in vivo.

1) In fact, we have standardized the PET results using SUV in evaluating glucose uptake. During PET-CT detection of tumor glucose uptake, standard uptake rate (SUL) or standard uptake value (SUV) are commonly used as metrics. Due to the convenience and clinical relevance, SUV is a more commonly used and accepted method for assessing tumor glucose uptake in clinical practice. This metric allows for quantitative analysis of the uptake level between different patients and different lesions. It provides information about tumor biology, metabolic status, and the entire lesion or the overall tumor burden. By calculating the maximum SUV (SUV_{max}), information about the voxel with the highest uptake in the lesion can be obtained, thus reflecting the maximum uptake level of the lesion. This index makes it relatively easy to compare the metabolic activity between different lesions. According to the FDG PET/CT procedure guidelines for tumor imaging ¹, we referenced the operational procedures and result analysis methods, and calculated SUV using the following formula: $SUV = \frac{\text{activity in tumor (kBq/mL)}}{\text{injected activity (MBq)/mouse weight (kg)}}$. According to the guideline, SUV_{max} is

recommended as a tumor uptake metrics. Therefore, we employed a similar approach to previous researchers in evaluating tumor glucose uptake ²⁻⁵. In the revised version, we have detailed experimental operations and analysis methods (Page 30-31).

2) However, when studying the glucose uptake function of gene in vivo using PET-CT, it is difficult to exclude the confounding effects of gene on glucose uptake. Our aim in the experiment is to demonstrate the regulatory role of MYG1 in the glycolytic process of colorectal cancer in vivo. Through PET-CT imaging, we can assess the uptake of glucose in tumors with different group of tumors. In order to simulate CRC in mice as closely as possible, we constructed an orthotopic model. However, due to the functional complexity of MYG1 in colorectal cancer, it may exert regulatory effects on various biological characteristics of tumors, such as tumor volume, the number of blood vessels in the tumor, tumor microenvironment, and so on. These factors may interfere with the evaluation of SUVmax. However, it is challenging to completely eliminate confounding factors, especially tumor volume, in in-vivo studies. Interestingly, in a recent study investigating the relationship between cold exposure and tumors ⁶, researchers conducted an intriguing experiment by controlling the tumor growth to the same volume in different groups of mice before performing PET-CT scans. This approach can eliminate the confounding effect of tumor volume differences, allowing for a more accurate assessment of the impact of treatment factors on glucose uptake and avoiding bias caused by varying tumor sizes. However, there are some limitations to consider. Although standardizing tumor volume to the same size can mitigate the impact of volume differences, other extrinsic factors may still confound the results. Variations in tumor biology, cellular composition, and other factors may still exert influence on glucose uptake. Besides, prolonging the growth time of tumors may alter their pathological characteristics and malignancy level.

3) Therefore, we attempted sensitivity analysis to assess the robustness of the results to these confounding factors. By adjusting the impact of confounding factors, such as excluding volumes within a specific range, we can verify whether the results are sensitive to these factors and evaluate the consistency of the results. Due to the fact that the bioluminescent signal can better reflect the cell population within the tumor than volume, we excluded tumors with signals lower than 5×10^5 P/s/mm² and higher than 30×10^5 P/s/mm², and re-evaluated the levels of glucose uptake regulated by MYG1 in different groups. The results indicated that MYG1 regulates glucose uptake consistently, and its variation is not sensitive to tumor volume (**Rebuttal Fig. 4a**).

4) Importantly, we also examined the GLUT1 expression of different tumors to evaluate the glucose uptake. GLUT1 is a key transporter responsible for transporting glucose into epithelial cells and playing a crucial

role in cellular glycolysis. Immunohistochemical staining results indicated that MYG1 upregulated the expression levels of GLUT1 in tumor cells, particularly in MYG1^N group, which is consistent with the in vitro results of western blot (**Rebuttal Fig. 4b**; Supplementary Fig. 5d and f in the manuscript). This offers additional evidence to support our conclusions regarding tumor glucose uptake.

We sincerely appreciate your insightful comments. Necessary adjustments and additional experiments have been added in the revised version to prove our conclusion.

Rebuttal Fig. 4. MYG1^N promotes glucose uptake and GLUT1 expression in vivo.

Reference:

- 1, Boellaard R et al. FDG PET/CT: EANM procedure guidelines for tumour imaging: version 2.0. *Eur J Nucl Med Mol Imaging*. 2015 Feb;42(2):328-54.
- 2, Li Q et al. Circular RNA MAT2B Promotes Glycolysis and Malignancy of Hepatocellular Carcinoma Through the miR-338-3p/PKM2 Axis Under Hypoxic Stress. *Hepatology*. 2019 Oct;70(4):1298-1316.
- 3, Weng ML et al. Fasting inhibits aerobic glycolysis and proliferation in colorectal cancer via the Fdft1-mediated AKT/mTOR/HIF1 α pathway suppression. *Nat Commun*. 2020 Apr 20;11(1):1869.
- 4, Demircioglu F et al. Cancer associated fibroblast FAK regulates malignant cell metabolism. *Nat Commun*. 2020 Mar 10;11(1):1290.
- 5, Zheng F et al. The HIF-1 α antisense long non-coding RNA drives a positive feedback loop of HIF-1 α mediated transactivation and glycolysis. *Nat Commun*. 2021 Feb 26;12(1):1341.
- 6, Seki T. Brown-fat-mediated tumour suppression by cold-altered global metabolism. *Nature*. 2022 Aug;608(7922):421-428.

REVIEWER COMMENTS

Reviewer #1 (Remarks to the Author):

In the revised version, Chen et al. provide several additional experiments and analyses to assess the role of Myg1 in CRC, including an additional shMyG1 species tested in vitro (but not in vivo), and new data implicating HSP90 in the regulation of PKM2 stability. Unfortunately, these data don't specifically address the previous concerns, and I find the support of their overall model to still be tentative, thereby precluding publication at Nature Communications. In addition, a number of statistical and technical issues are still outstanding. Specific issues are outlined below:

1) With respect to the previous major point 1, the authors identify a deletion mutant which impacts PKM2 binding, and impairs the effect of MYG1 on glycolysis and cell proliferation. Given that this is a deletion mutant, it is important to evaluate if the deletion impacts other functions of Myg1 (e.g., exonuclease activity, and the proposed roles in the mitochondria). In addition, to conclude that PKM2 binding is important to tumorigenesis and glycolysis, the authors should evaluate xenografted tumors of the deletion mutant (similar to experiments in Figure 2M-Q, Figure 4g-i) as previously mentioned in major point 1.

2) With respect to the previous major point 2, the authors better explain their idea for a positive feedback loop involving Myg1-PKM2-Myc. The two key pieces of data appear to be that nuclear Myg1 activates Myc expression (Fig. 6G), and that Myc activates Myg1 expression (Fig. 6K). These events set up a theoretical possibility of a positive feedback loop. However, the authors do not definitively test that this feedback loop is in play. Simply knocking down PKM2 or Myc is not sufficient, since these proteins have well-known direct roles in glycolysis and tumorigenesis. The key prediction/definition of a feedback loop is that Myg1 levels should adopt a switch-like behavior, which would result in a bimodal distribution of Myg1 expression in cells. I do not see that observation of a feedback loop is directly addressed in the manuscript, although perhaps I missed it.

Moreover, this idea now draws in a bit of confusion with respect to the other experiments. From my understanding, this feedback loop should not be present in the Myg1 overexpression experiments, since these are based off a CMV promoter? Thus, I find it difficult now to ascertain that the effects of Myg1 overexpression can be consistent with the requirement for a feedback loop.

3) With respect to the previous major point 3: The authors provide a proteinase K shaving assay; however, the version they performed does not distinguish between matrix and intermembrane space localization. To verify matrix localization, the authors need to modify their proteinase K assay to assess mitoplasts, using appropriate controls in the intermembrane space. Otherwise, the text should be modified to indicate that the authors cannot definitively prove matrix localization of Myg1. Related to this, it is not clear from the single image presented in Fig. 4e that Myg1_M is exclusively mitochondrial (it looks to be more diffuse), due to the resolution and magnification – additional analyses would have to be performed to make this conclusion.

In the title and discussion, the authors conclude that nuclear and mitochondrial effects of Myg1 are "coordinated". I am not sure what "coordinated" means in this context, but as far as I can tell, the authors analyze these functions separately, and have not performed experiments to address if there are synergistic or antagonizing effects due to nuclear and mitochondrial Myg1.

Other technical issues:

1) With respect to figure 1A (previous technical issue 1), my impression now is that a GSEA is performed using Kegg pathways on genes somehow identified from Myg1 high vs. Myg1 low TCGA COADREAD tumors. The authors indicate in their methods:

"For GSEA, we utilized TCGA RNA-Seq data to classify samples into high and low expression groups based on MYG1 expression. GSEA was then performed to assess the enrichment of pathways or gene sets between these two groups, providing insights into the functional significance of MYG1 expression in 41 CRC patients."

I may be incorrect, but from my understanding GSEA is typically performed after identification of a set of genes. It is not clear what genes were used. Their supplementary data 1 only provides the enrichment results, but not which genes were used to assess enrichment. Some clarification is still needed here.

2) With respect to previous technical issue 2, the authors now use a second Myg1 shRNA for some of their in vitro analyses. The second myg1 shRNA should be used for experiments performed in Fig 2m-q, to verify that the effects are on-target.

The authors also now use an sgMyg1 approach in a separate cell line (LoVo) which does not directly address if the previous effects in SW480 and HCT116 were on target. A second independent sgMyg1 in LoVo cells would be needed to validate that off-target effects are not responsible for the phenotype in LoVo cells.

3) With respect to previous technical point 4 (now figure 3g,h, etc.): Based on the methods, the authors appear to have normalized these data from the seeded cell count (5×10^5 cells) on the day prior to the assay, as opposed to the cell count at the time of the assay. This is likely to confound the results, given the proliferation changes noted in these cell lines. Normalization should be performed based on cell counts / protein amounts on the day of the assay.

Similarly, extracellular flux measurements (previous technical point 9) should be performed based on cell count – the y-axis does not indicate the normalization measure.

4) In the revised version, the authors now report basal/max/capacity data for extracellular flux measurements (OCR/ECAR). These data appear to be incorrect, as it does not appear that the non-mitochondrial OCR was subtracted from the data. No methods are provided for how these calculations are made. The normalization is also not presented within the figure (i.e., the y-axis is not normalized as depicted). To my eye, Myg1_M does not inhibit basal or max OCR in HCT116 cells, as the non-mito OCR is also decreased. Thus, the conclusion that mitochondrial form of Myg1 inhibits oxidative phosphorylation is not substantiated by the data, and it is unclear how Myg1_M would affect the non-mitochondrial OCR.

5) I find the co-localization analyses presented in figures 7a,b,e, etc. to be misleading. By using a line that extends beyond the cell boundaries, or into the nucleus, the authors mistakenly given the impression of co-localization, when in fact the fluorescence correlation is only due to whether the pixel is inside or outside of the cell. Moreover, only a single line from a single image is analyzed, without any quantitation or statistical analyses. In these images, the Myg1 and/or Cy5 staining is often quite diffuse, which precludes a meaningful co-localization analysis.

6) Statistical issues:

Fig. 1b – should be a paired samples t-test?

In a number of panels (Fig. 3G,H, Supplemental Figure 4E, Figure 6A-C, etc.) – From my understanding, these data should be analyzed with a 1-way ANOVA with correction for multiple comparisons (not a student's t-test).

Figure 6D: The legend indicates a two-way ANOVA was performed, but only a single p-value is presented. What does this p-value represent?

7) Minor points:

There are several grammar/spelling errors in the text and figures that should be corrected.

For glucose/lactate uptake measurements, an additional minor point is that the same units should be used (either μmol or mmol) to make it easier for the reader to compare glucose uptake and lactate excretion.

Reviewer #2 (Remarks to the Author):

I have gone through the detailed point by point response as well as the revised manuscript document.

I feel that the points raised by me have been sufficiently addressed to warrant my affirmation for publishing this work in Nat Comm.

Reviewer #3 (Remarks to the Author):

The authors addressed all of my comments. Thank you!

Reviewer #4 (Remarks to the Author):

The authors have performed a serious amount of extra work to support their data and to accommodate previous findings on MYG1. Questions have been addressed properly and model is much more substantiated in the current manuscript.

RESPONSE TO REVIEWERS' COMMENTS

Note to reviewers: We thank reviewers for taking efforts to review our manuscript and for providing insightful comments to further improve our paper. Below we provide the detailed point-by-point response to address all the comments raised by reviewers. To facilitate the review of our manuscript and the rebuttal letter by reviewers, we present the key data in this letter, with referrals to corresponding revised figures and text in our revised manuscript. We have also marked all the changes in our revised manuscript by red highlighting.

REVIEWER COMMENTS

Reviewer #1 (Remarks to the Author):

In the revised version, Chen et al. provide several additional experiments and analyses to assess the role of Myg1 in CRC, including an additional shMyG1 species tested in vitro (but not in vivo), and new data implicating HSP90 in the regulation of PKM2 stability. Unfortunately, these data don't specifically address the previous concerns, and I find the support of their overall model to still be tentative, thereby precluding publication at Nature Communications. In addition, a number of statistical and technical issues are still outstanding. Specific issues are outlined below:

We thank the reviewer for reviewing our manuscript and providing valuable comments. We have taken these concerns seriously and revised our manuscript. The newly added contents are highlighted in the red text and we hope our response sufficiently addresses your concerns.

1) With respect to the previous major point 1, the authors identify a deletion mutant which impacts PKM2 binding, and impairs the effect of MYG1 on glycolysis and cell proliferation. Given that this is a deletion mutant, it is important to evaluate if the deletion impacts other functions of Myg1 (e.g., exonuclease activity, and the proposed roles in the mitochondria). In addition, to conclude that PKM2 binding is important to tumorigenesis and glycolysis, the authors should evaluate xenografted tumors of the deletion mutant (similar to experiments in Figure 2M-Q, Figure 4g-i) as previously mentioned in major point 1.

We agree with the reviewer's comment. Our results have demonstrated that MYG1 promotes CRC glycolysis and progression mainly independent of its exonuclease activity. In this study, we specifically focused on investigating the role of MYG1 in interacting with PKM2. The deletion mutant of MYG1 preserves the DHH domain since our findings indicate that MYG1 interacts with PKM2 at the 149-199 site. Considering the importance of PKM2 binding, the preservation

of the DHH domain implies that detecting exonuclease activity is not essential for studying the effect of PKM2 binding.

Furthermore, it is important to note that the functions of MYG1 in cancer cells and yeast may differ, and the exonuclease activity-dependent roles in mitochondrial function in CRC cells remain unclear. Therefore, it would be premature to exclude other potential influences of the deletion mutant on cells. These aspects highlight the need for further extensive research on the functions of MYG1. We have discussed this limitation in the manuscript:

“The functional investigation of MYG1 is not well established, and recent studies have gradually revealed its role in mitochondrial function. Particularly, research on the role of MYG1 in tumors is still in its early stages, and its functions necessitate extensive investigations.”

Following the reviewer’s suggestions, we employed a subcutaneous tumor model to evaluate the influence of PKM2 binding on tumorigenesis and glycolysis. Deleting the PKM2 binding site of MYG1 resulted in inhibited tumor growth compared to the wild type (**Revised Fig. 6f**). Furthermore, we performed immunohistochemical (IHC) analysis on the expression of PKM2 and GLUT1 in the tumors to assess the effect of PKM2 binding on glycolysis. The IHC data from mice tumors indicated that MYG1^Δ suppressed the expression of GLUT1 and PKM2, suggesting reduced levels of glycolysis in MYG1^Δ tumors (**Revised Supplementary Fig.7 d-e**). These results strongly indicate that PKM2 binding plays a crucial role in MYG1-induced tumorigenesis and glycolysis.

Fig. 6f

Supplementary Fig. 7d-e

Thank you for raising this important issue, and we have conducted these additional experiments and explanations to provide further evidence and strengthen the conclusion.

2) With respect to the previous major point 2, the authors better explain their idea for a positive feedback loop involving Myg1-PKM2-Myc. The two key pieces of data appear to be that nuclear Myg1 activates Myc expression (Fig. 6G), and that Myc activates Myg1 expression (Fig. 6K). These events set up a

theoretical possibility of a positive feedback loop. However, the authors do not definitively test that this feedback loop is in play. Simply knocking down PKM2 or Myc is not sufficient, since these proteins have well-known direct roles in glycolysis and tumorigenesis.

The key prediction/definition of a feedback loop is that Myg1 levels should adopt a switch-like behavior, which would result in a bimodal distribution of Myg1 expression in cells. I do not see that observation of a feedback loop is directly addressed in the manuscript, although perhaps I missed it.

Moreover, this idea now draws in a bit of confusion with respect to the other experiments. From my understanding, this feedback loop should not be present in the Myg1 overexpression experiments, since these are based off a CMV promoter? Thus, I find it difficult now to ascertain that the effects of Myg1 overexpression can be consistent with the requirement for a feedback loop.

We appreciate the reviewer's comments on the feedback loop. While it is widely acknowledged that PKM2 and c-Myc have directly effect on glycolysis and tumorigenesis, our aim in knocking down PKM2 and c-Myc was to investigate their regulation on MYG1 expression.

Regarding the key definition of a feedback loop, we agree that MYG1 levels should exhibit a bimodal distribution. Consistent with this notion, our findings indicate the activation of the PKM2-Myc feedback loop in KRAS mutated colorectal cancer (CRC) patients, leading to significantly higher expression levels of MYG1 compared to KRAS wild-type patients. Examination of MYG1 expression in cell lines also revealed increased levels in those with KRAS and APC mutations (**Revised Fig. 1f and Supplementary Fig. 1g**).

In addition to conducting overexpression assays to examine the regulation of PKM2 by MYG1, we have also evaluated the effect of MYG1 knockout (KO) on the PKM2 expression (**Revised Supplementary Fig. 6d**). Furthermore, in response to your comments, we have investigated c-Myc expression in MYG1 KD cells (**Revised Supplementary Fig. 8f**). The results consistently demonstrate that silencing MYG1 inhibits the expression of both PKM2 and c-Myc.

Supplementary Fig. 6d

Supplementary Fig. 8f

To demonstrate the transcriptional regulation of PKM2 and c-Myc on MYG1, we have performed ChIP-qPCR and double luciferase reporter experiments. The results showed that both PKM2 and c-Myc regulate MYG1 transcription

(Revised Fig. 6I-o). Additionally, we observed a high correlation among the expression levels of these proteins (Revised Supplementary Fig. 8g and Revised Fig. 8b).

In summary, the consistency of our data and the biological rationale underlying the relevant findings strongly support the conclusion that MYG1-PKM2-c-Myc forms a positive feedback regulatory loop in CRC.

3) With respect to the previous major point 3: The authors provide a proteinase K shaving assay; however, the version they performed does not distinguish between matrix and intermembrane space localization. To verify matrix localization, the authors need to modify their proteinase K assay to assess mitoplasts, using appropriate controls in the intermembrane space. Otherwise, the text should be modified to indicate that the authors cannot definitively prove matrix localization of Myg1. Related to this, it is not clear from the single image presented in Fig. 4e that Myg1_M is exclusively mitochondrial (it looks to be more diffuse), due to the resolution and magnification – additional analyses would have to be performed to make this conclusion.

In the title and discussion, the authors conclude that nuclear and mitochondrial effects of Myg1 are “coordinated”. I am not sure what “coordinated” means in this context, but as far as I can tell, the authors analyze these functions separately, and have not performed experiments to address if there are synergistic or antagonizing effects due to nuclear and mitochondrial Myg1.

We appreciate the reviewer's suggestion to differentiate the location of MYG1 in the matrix and intermembrane space of mitochondria. To address this, we purified mitochondria from 293T cells and treated them with 0.2% digitonin to disrupt the mitochondrial outer membrane. Subsequently, the matrix and intermembrane space fractions of mitochondria were subjected to proteinase K treatment. For control purposes, we selected MRPS27 as a marker for the matrix, VDAC1 as a marker for the outer membrane, and Cyt c as a marker for the intermembrane space. However, our results indicated that digitonin also disrupted the inner membrane of mitochondria and resulted in digestion of the matrix proteins (Revised Supplementary Fig. 5b). We acknowledge that our methods may not effectively distinguish between mitoplasts and the intermembrane space.

Supplementary Fig. 5b

Based on our findings, along with previous proteinase K assays and immunofluorescence (IF) assays, we can conclude that MYG1 is localized in the mitochondria. However, we acknowledge that the exclusive localization of MYG1^M in Fig. 4e may not be entirely accurate due to the potential presence of MYG1 in the cytoplasm and non-specific signaling of the flag antibody. Moreover, the mitochondrial localization of MYG1 has been confirmed through immunoelectron microscopy observation. Based on these results, we can conclude that MYG1 is located in the mitochondria. We have updated the manuscript to reflect this revised description of the results:

“We further performed the proteinase K shaving assay to validated the mitochondrial location of endogenous and exogenous MYG1 in 293T and LoVo cells. The results showed that MYG1 was located in the mitochondria instead of the mitochondrial outer membrane (Fig. 4b and Supplementary Fig. 5a-b). Immunoelectron microscopy was also employed to observe the location of MYG1 and the results showed that MYG1 was located in mitochondria (Fig. 4c). These results concluded that MYG1 has dual localization of nucleus and mitochondria.”

Regarding the synergistic effect of MYG1^N and MYG1^M on CRC progression, we have confirmed it through in vitro experiments. Proliferation and invasion assays (**Revised Supplementary Fig. 5g-h**) demonstrated that both MYG1^N and MYG1^M can promote the proliferation and invasion of CRC cells. Notably, wild-type MYG1 exhibited the most pronounced effect, indicating a synergistic promotion of CRC progression by MYG1^N and MYG1^M. Furthermore, in vitro glycolysis assays revealed that MYG1^N enhanced CRC glucose uptake and lactate secretion, while MYG1^M disrupted oxygen consumption rate (OCR) and induced apoptosis in CRC cells. These findings suggest that MYG1, with its different subcellular localizations, drives the metabolic shift towards enhancing glycolysis in CRC cells and promotes CRC progression. Based on these results, we conclude that both MYG1^N and MYG1^M collectively contribute to CRC progression and glycolysis.

Other technical issues:

1) With respect to figure 1A (previous technical issue 1), my impression now is that a GSEA is performed using Kegg pathways on genes somehow identified from Myg1 high vs. Myg1 low TCGA COADREAD tumors. The authors indicate in their methods:

“For GSEA, we utilized TCGA RNA-Seq data to classify samples into high and low expression groups based on MYG1 expression. GSEA was then performed

to assess the enrichment of pathways or gene sets between these two groups, providing insights into the functional significance of MYG1 expression in CRC patients.”

I may be incorrect, but from my understanding GSEA is typically performed after identification of a set of genes. It is not clear what genes were used. Their supplementary data 1 only provides the enrichment results, but not which genes were used to assess enrichment. Some clarification is still needed here.

Thank you for your valuable feedback. We appreciate the opportunity to provide further clarification on your concerns.

You have correctly understood the purpose of GSEA, which is a knowledge-based approach used to interpret genome-wide expression profiles. In our study, we utilized mRNA expression data obtained from the TCGA COADREAD cohort, as depicted in Figure 1A of our manuscript. It is important to note that we conducted GSEA using this comprehensive dataset, which means that all genes included in the expression data were analyzed.

To perform the analysis, we classified the samples into two groups based on the expression levels of our gene of interest, MYG1. Specifically, the samples were divided into a high-expression group and a low-expression group. Subsequently, we employed GSEA software with the default metric Signal2Noise to rank all genes listed in the expression matrix. This ranking allowed us to determine the differential expression between the high-expression and low-expression groups. Genes at the top of the ranked list represent upregulated genes, while genes at the bottom indicate downregulated genes. For a more detailed understanding of the specific sorting algorithm, please refer to the official GSEA website: <https://www.gsea-msigdb.org/gsea/doc/GSEAUserGuideFrame.html>.

By employing this approach, we were able to identify pathways that exhibit associations with MYG1 expression. We have made revisions to our manuscript to provide a clearer explanation of the GSEA analysis methodology:

“RNA-Seq (level-3) data of COADREAD (n = 433) were downloaded from TCGA. Initially, we ranked the tumor samples according to the expression levels of MYG1 and divided them into two groups: high expression (top 50% samples) and low expression (bottom 50% samples). Using the GSEA default preranked method, Signal2Noise, we ranked all genes and conducted enrichments using Hallmarkers or KEGG pathways as reference sets.”

2) With respect to previous technical issue 2, the authors now use a second Myg1 shRNA for some of their in vitro analyses. The second myg1 shRNA

should be used for experiments performed in Fig 2m-q, to verify that the effects are on-target.

The authors also now use an sgMyg1 approach in a separate cell line (LoVo) which does not directly address if the previous effects in SW480 and HCT116 were on target. A second independent sgMyg1 in LoVo cells would be needed to validate that off-target effects are not responsible for the phenotype in LoVo cells.

We appreciate the reviewer's concerns regarding the potential off-target effects of shRNAs and sgRNA targeting MYG1. Our in vitro data showed that another shRNA, shRNA#2, yielded similar results to shRNA#1 in terms of inhibiting proliferation, migration, and invasion. These findings suggest that the observed effects were specifically mediated by MYG1, providing evidence for the on-target specificity of shRNA#1. To further address the reviewer's concerns and evaluate the effects in an in vivo setting, we incorporated a xenograft tumor model. As depicted in **Revised Fig. 2m**, both shRNAs demonstrated significant inhibition of tumor growth in vivo. By including these in vivo findings, we have strengthened the evidence supporting the specific and effective targeting of MYG1 by the shRNAs.

We also evaluated the impact of the two shRNAs on the expression of epithelial-mesenchymal transition (EMT) markers (**Revised Fig. 2q and Supplementary Fig. 3a**). These results further demonstrated that MYG1 inhibition led to suppressed proliferation, migration, invasion, and EMT in CRC cells.

The CRISPR-Cas9 system has been reported to have lower off-target efficiency compared to the RNAi system. In our study, we utilized sgRNA sequences designed by CHOPCHOP (<https://chopchop.cbu.uib.no/>) to ensure the absence of potential off-target sites. The knockout (KO) effect of the selected sgRNAs was validated by western blot analysis, confirming the efficacy of the sgRNA sequences used in the manuscript.

3) With respect to previous technical point 4 (now figure 3g,h, etc.): Based on the methods, the authors appear to have normalized these data from the seeded cell count (5×10^5 cells) on the day prior to the assay, as opposed to the cell count at the time of the assay. This is likely to confound the results, given the proliferation changes noted in these cell lines. Normalization should be performed based on cell counts / protein amounts on the day of the assay. Similarly, extracellular flux measurements (previous technical point 9) should be performed based on cell count – the y-axis does not indicate the normalization measure.

We have provided additional details about the assays. Normalization was conducted before and after experiments, and we have described the specific procedures in the Methods section:

“The Seahorse Wave software was used to analyze the data. The cells in each well were digested and counted to normalize the results after detection. The non-mitochondrial OCR and non-glycolytic acidification were subtracted when performing the quantification.”

4) In the revised version, the authors now report basal/max/capacity data for extracellular flux measurements (OCR/ECAR). These data appear to be incorrect, as it does not appear that the non-mitochondrial OCR was subtracted from the data. No methods are provided for how these calculations are made. The normalization is also not presented within the figure (i.e., the y-axis is not normalized as depicted). To my eye, Myg1_M does not inhibit basal or max OCR in HCT116 cells, as the non-mito OCR is also decreased. Thus, the conclusion that mitochondrial form of Myg1 inhibits oxidative phosphorylation is not substantiated by the data, and it is unclear how Myg1_M would affect the non-mitochondrial OCR.

Thank you for your professional suggestions. We have recalculated the OCR/ECAR data by subtracting the non-mitochondrial OCR and non-glycolytic acidification from the OCR and ECAR measurements separately. The mitochondrial OCR data was then subjected to statistical analysis. We have provided a more detailed description of the methodology in the Methods section. Furthermore, we have revised the labeling of the y-axis for the OCR and ECAR data.

Regarding the confusing data observed in HCT116 cells in **Fig. 7f**, we conducted repeated experiments in both HCT116 and SW480 cells and performed normalization. The results consistently showed that MYG1^M inhibited the OCR of both HCT116 and SW480 cells (**Revised Fig. 7f**).

5) I find the co-localization analyses presented in figures 7a,b,e, etc. to be misleading. By using a line that extends beyond the cell boundaries, or into the nucleus, the authors mistakenly given the impression of co-localization, when in fact the fluorescence correlation is only due to whether the pixel is inside or outside of the cell. Moreover, only a single line from a single image is analyzed, without any quantitation or statistical analyses. In these images, the Myg1 and/or Cycs staining is often quite diffuse, which precludes a meaningful co-localization analysis.

Thank you for your valuable feedback. Generally, there are two commonly used methods for performing colocalization analysis of immunofluorescence (IF) images. In our previous analysis, we employed the Plot Profile method, which is a descriptive approach. However, in response to the reviewer's concern, we have now incorporated the calculation of Pearson's correlation coefficient (PCC), a quantitative method for assessing colocalization.

To analyze the colocalization of MYG1 and CYCS in mitochondria, we could only use the Plot Profile method as it is challenging to accurately distinguish regions of interest (ROI) within cells during analysis. As shown in Fig. 7a-b, MYG1 is localized in mitochondria and exhibits good co-localization with Mitotracker. However, the distribution of Cyt c varies among different cells (Cyt c appears to be more distributed in the cytoplasm of HCT116 cells). These distribution patterns do not negate the fact that Cyt c is also localized in mitochondria. We have updated the corresponding results with quantification as requested.

6) Statistical issues:

Fig. 1b – should be a paired samples t-test?

In a number of panels (Fig. 3G,H, Supplemental Figure 4E, Figure 6A-C, etc.) – From my understanding, these data should be analyzed with a 1-way ANOVA with correction for multiple comparisons (not a student's t-test).

Figure 6D: The legend indicates a two-way ANOVA was performed, but only a single p-value is presented. What does this p-value represent?

Thank you for these comments. We utilized paired CRC tumors and adjacent normal mucosa for performing western blot in **Fig. 1b**. Based on our analysis, we believe that the experimental design meets the criteria for employing Paired-Sample t-test.

For the statistical issues, we agree with your viewpoint and a one-way ANOVA was more suitable used to analyze the difference between the means of more than two groups with one independent variable. We have reperformed statistical

analyse in the corresponding panels and changed the corresponding figure legends.

7) Minor points:

There are several grammar/spelling errors in the text and figures that should be corrected.

For glucose/lactate uptake measurements, an additional minor point is that the same units should be used (either μmol or mmol) to make it easier for the reader to compare glucose uptake and lactate excretion.

We have carefully checked the grammar and spelling errors in the text and figures. In the previous version, the glucose uptake and lactate excretion data was displayed in the same unit, but the unit for lactate secretion was mistakenly labeled. We have verified and corrected.

Reviewer #2 (Remarks to the Author):

I have gone through the detailed point by point response as well as the revised manuscript document.

I feel that the points raised by me have been sufficiently addressed to warrant my affirmation for publishing this work in Nat Comm.

We thank the reviewer for the support.

Reviewer #3 (Remarks to the Author):

The authors addressed all of my comments. Thank you!

We thank the reviewer for reviewing our manuscript and providing supportive comment.

Reviewer #4 (Remarks to the Author):

The authors have performed a serious amount of extra work to support their data and to accommodate previous findings on MYG1. Questions have been addressed properly and model is much more substantiated in the current manuscript.

Thank you for your recognition of our manuscript.

Reviewer #1 (Remarks to the Author):

Previous point 1:

The authors explored the effect of overexpressing the Myg1deletion mutant which abrogates binding to PKM2. Consistent with their model, they observe decreased tumor growth (relative to MYG1 wt), and decreased IHC expression of GLUT1 and PKM2. Again though, these results could be due to effects on exonuclease activity or the proposed mitochondrial role, which were not tested by the authors.

As such, their description of the limitation is inadequate:

“The functional investigation of MYG1 is not well established, and recent studies have gradually revealed its role in mitochondrial function. Particularly, research on the role of MYG1 in tumors is still in its early stages, and its functions necessitate extensive investigations.”

The authors should specifically list the limitation ... that although they identified a mutant with decreased PKM2 binding, they cannot exclude the possibility that the effects are due to impacts on other functions of MYG1, including exonuclease and mitochondrial roles. Thus, the impact of MYG1 binding to PKM2 binding is not clear.

Previous point 2:

This point referred to evidence for a positive feedback regulatory loop. While the authors certainly show that the key biological events are in place for a positive feedback loop (i.e., Myg1 binds to PKM2 and regulates Myc levels), and PKM2/Myc binds to and regulates Myg1 expression, this doesn't mean that a feedback loop is engaged or important to the biology. This would require showing that low levels of Myg1 expression result in amplification of Myg1 signals in a PKM2/Myc-dependent manner, resulting in bimodal distributions. Certainly, this might be a challenging experiment, and I would suggest the authors remove reference to a positive feedback loop in the manuscript. The presence or absence of a feedback loop doesn't impact the experimental findings relating to Myg1's control of Myc expression, or Myc expression of Myg1 levels.

Previous point 3:

The authors have adequately adjusted their manuscript to remove reference to “matrix” localization of Myg1.

The authors do not provide evidence of a synergistic effect for nuclear and mitochondrial Myg1. Synergism implies that the observed effect is greater than the combined effect of the individual parts; and would require a statistical test relative to an independence statistical model (e.g., the Bliss model). To my eye, expression of MyG1_wt is approximately equal to the sum of Myg1_N and Myg1_M (Supplemental Figure 5g), and is clearly less than the sum of the individual effects (Supplemental Figure 5h).

Previous technical point 1:

The authors have adequately explained their ranked gene list input into GSEA; thank you for this clarification!

Previous technical point 2:

The authors have adequately responded to concerns about off-target effects from shMYG1#1, by providing an analysis of shMYG#2 in Figure 2m,q; thank you! I will note though that they have not evaluated their second shRNA in other assays requested (Figures 2n,o) or the effect of a second sgRNA in Lovo cells (Figure 2i,j,k,l). I do not find their argument (without citation or evidence) that eliminating off-target effects using the CHOPCHOP system to be compelling. This remains a limitation of the manuscript due to the potential for off-target effects.

Previous technical point 3:

The authors addressed normalization issues regarding Seahorse (OCR/ECAR) data (e.g., Figure 3 c-f) in their response; thank you for that. However, this comment specifically asked about normalization issues regarding glucose uptake and lactate secretion in Figure 3g,h which has not been addressed.

Previous technical point 4:

What is the “Norm.Unit” used in the Seahorse data?

I am concerned about the authors' replacement of data in Figure 7f, which suggests that the

finding is not robust given the differences between the two datasets. The data for SW480 cells was not replaced with their new data ... I'm unclear as to why they chose to replace one dataset and not the other?

Previous technical point 5:

The analysis of Figure 7A,B is identical to the previous comment, and does not address issues surrounding the use of a line that extends outside cytoplasmic boundaries, the use of only a single line from a single image, and no quantitation or statistical analysis.

With respect to Figure E, the figure legend does not clarify which channels were used to calculate the Pearson's correlation coefficient, or what the data points represent? Is this one image each from 3 patients for the MYG1/CYCS image? It is also not clear how this correlation coefficient was calculated (no description in the methods section) ... Were the fluorescent values centered/normalized? Was a mask applied to only include cell boundaries? How was the mask created? By analyzing a region-of-interest that extends beyond cell boundaries, one can artificially elevate correlation as neither channel will have signal in the blank regions.

These same issues apply to Figure 5a, where lines are drawn outside cell boundaries and no methods description is provided for calculation of the PCC.

Previous technical point 6:

It is not clear if p-values from one-way ANOVA calculations and other multiple-tests have been corrected for multiple comparisons and what method was used. For these values, the authors should take care to specify what comparison the p-value refers to (e.g., Fig. 1f)

Shouldn't Figure 4l,m be a 2-way ANOVA?

Previous technical point 7:

Minor point: Y-axis title for Figure 3h should read " 5×10^5 cells" not " 5×10^5 cells"

Response to Reviewer #1:

Thank you for your valuable comments and suggestions on improving our manuscript. We greatly appreciate your thorough and patient review. We have carefully considered each of your suggestions and addressed them accordingly in the revised version of the manuscript. In addition, we have made necessary revisions to grammar and wording throughout the text. All changes have been highlighted in red.

Reviewer #1 (Remarks to the Author):

Previous point 1:

The authors explored the effect of overexpressing the Myg1deletion mutant which abrogates binding to PKM2. Consistent with their model, they observe decreased tumor growth (relative to MYG1 wt), and decreased IHC expression of GLUT1 and PKM2.

Again though, these results could be due to effects on exonuclease activity or the proposed mitochondrial role, which were not tested by the authors.

As such, their description of the limitation is inadequate:

“The functional investigation of MYG1 is not well established, and recent studies have gradually revealed its role in mitochondrial function. Particularly, research on the role of MYG1 in tumors is still in its early stages, and its functions necessitate extensive investigations.”

The authors should specifically list the limitation ... that although they identified a mutant with decreased PKM2 binding, they cannot exclude the possibility that the effects are due to impacts on other functions of MYG1, including exonuclease and mitochondrial roles. Thus, the impact of MYG1 binding to PKM2 binding is not clear.

We thank the reviewer for the comment and suggestion. We have made changes to the limitations in the discussion and provided a more objective and comprehensive analysis of the limitations of our research.

(Page 25, Line 546-557)

“The functional investigation of MYG1 is not well established, and recent studies have gradually revealed its role in mitochondrial function. Particularly, research on the role of MYG1 in tumors is still in its early stages, and its functions necessitate extensive investigations. Our study found that MYG1 binds to PKM2 and maintains its stability, and we attempted to verify the importance of MYG1-PKM2 binding in glycolysis and CRC tumor progression by deleting the PKM2 binding site of MYG1. While our findings demonstrate that MYG1^Δ, which lacks the PKM2 binding site, partially reverses glycolysis and tumor progression compared to wild-type MYG1, it is important to acknowledge that there may be other functional abnormalities associated with MYG1^Δ,

such as exonuclease activity or mitochondrial function. Hence, the observed effects cannot definitively exclude the potential impact of these alternative functions.”

Previous point 2:

This point referred to evidence for a positive feedback regulatory loop. While the authors certainly show that the key biological events are in place for a positive feedback loop (i.e., Myg1 binds to PKM2 and regulates Myc levels), and PKM2/Myc binds to and regulates Myg1 expression, this doesn't mean that a feedback loop is engaged or important to the biology. This would require showing that low levels of Myg1 expression result in amplification of Myg1 signals in a PKM2/Myc-dependent manner, resulting in bimodal distributions. Certainly, this might be a challenging experiment, and I would suggest the authors remove reference to a positive feedback loop in the manuscript. The presence or absence of a feedback loop doesn't impact the experimental findings relating to Myg1's control of Myc expression, or Myc expression of Myg1 levels.

Thank you for your valuable feedback on the positive feedback regulatory loop. We agree with your suggestion and have made appropriate revisions to the manuscript regarding the existence of the feedback loop (Page 18, Line 380; Page 20, Line 424-429, Page 21, Line 444-446, Page 23, Line 507).

Previous point 3:

The authors have adequately adjusted their manuscript to remove reference to “matrix” localization of Myg1.

The authors do not provide evidence of a synergistic effect for nuclear and mitochondrial Myg1. Synergism implies that the observed effect is greater than the combined effect of the individual parts; and would require a statistical test relative to an independence statistical model (e.g., the Bliss model). To my eye, expression of MyG1_wt is approximately equal to the sum of Myg1_N and Myg1_M (Supplemental Figure 5g), and is clearly less than the sum of the individual effects (Supplemental Figure 5h).

Thank you for your feedback on the synergistic effect of the nuclear and mitochondrial MYG1. We appreciate your clarification on the definition of synergy and agree with your point that the observed effects do not exhibit a synergistic effect.

Given your insightful suggestion, we have re-evaluated our findings and corresponding descriptions in the manuscript and revised the relevant statements to accurately reflect this relationship. We have changed the mention of synergistic effects and used “collaboration” to describe the role of nuclear

and mitochondrial MYG1. The corresponding descriptions were changed.

(Page 5, Line 95)

“Specifically, MYG1 promotes glycolysis through its functional coordination of nucleus and mitochondria, independent of its exonuclease activity.”

(Page 23, Line 497-499)

Taken together, our study identifies the oncogenic protein MYG1 and reveals that both nuclear and mitochondrial MYG1 cooperatively promote metabolic remodeling and tumor progression of CRC (Fig. 8f).

(Page 25, Line 531-536)

“Highly expressed MYG1 induces OXPHOS damage and strengthens glycolysis in CRC, and differently localized MYG1 showed a cooperated function. Our study reveals a mechanism of MYG1, especially in KRAS-mutant CRC patients where MYG1 is upregulated. MYG1, in a manner independent of exonuclease activity, promotes tumor glycolysis and progression through nuclear and mitochondrial functions.”

Previous technical point 1:

The authors have adequately explained their ranked gene list input into GSEA; thank you for this clarification!

Thank you for acknowledging our explanation. We are pleased to clarify this point.

Previous technical point 2:

The authors have adequately responded to concerns about off-target effects from shMYG1#1, by providing an analysis of shMYG#2 in Figure 2m,q; thank you! I will note though that they have not evaluated their second shRNA in other assays requested (Figures 2n,o) or the effect of a second sgRNA in Lovo cells (Figure 2i,j,k,l). I do not find their argument (without citation or evidence) that eliminating off-target effects using the CHOPCHOP system to be compelling. This remains a limitation of the manuscript due to the potential for off-target effects.

Thank you for raising concerns about the potential off-target effects of the shRNA and sgRNA.

We have previously attempted to address this issue by using the shMYG1 #2 in some vitro experiments and the subcutaneous xenograft model to provide additional analysis. We believe that the various methods we employed, including shRNA and sgRNA, have provided a sufficient basis for concluding shRNAs and sgRNA are on-target and the biological functions we aim to

elucidate.

Of course, we also agree with your concerns about off-target effects of shRNAs and sgRNA. The possibility of off-target effects remains a limitation of our research, and we have addressed this limitation in the discussion, emphasizing the necessity of further investigation to fully evaluate and mitigate these risks.

(Page 24, Line 524-526)

“In addition, some functional studies of MYG1 were conducted using a single shRNA or sgRNA, which suggests potential off-target effects. Future studies will evaluate and mitigate these risks.”

Previous technical point 3:

The authors addressed normalization issues regarding Seahorse (OCR/ECAR) data (e.g., Figure 3 c-f) in their response; thank you for that. However, this comment specifically asked about normalization issues regarding glucose uptake and lactate secretion in Figure 3g,h which has not been addressed.

Thank you for pointing out the concern regarding normalization issues with glucose uptake and lactate secretion in Figure 3g and h. We apologize for not addressing this specific comment in our previous response.

To address these standardization issues, we have clarified them in the method. Before measurement, we ensure that the same number of cells are seeded into each well to maintain consistency between wells during the detection. To prevent the impact of cell proliferation, we used serum-free culture medium to inhibit cell proliferation before the experiments. By carefully controlling the cell count before performing the assays, we aimed to minimize any potential variability associated with cell number, which could impact the glucose uptake and lactate secretion readings.

(Page 37, Line 795)

“For glucose uptake and lactate secretion assays, cells (5×10^5) were seeded in a 96-well plate and cultured overnight in serum-free medium.”

Previous technical point 4:

What is the “Norm.Unit” used in the Seahorse data?

I am concerned about the authors’ replacement of data in Figure 7f, which suggests that the finding is not robust given the differences between the two datasets. The data for SW480 cells was not replaced with their new data ... I’m unclear as to why they chose to replace one dataset and not the other?

We apologize for the confusion. The term "Norm.Unit" in the Seahorse data refers to data that has been normalized by cell number. We have clarified the description in the corresponding figure legends.

We appreciate your concern regarding the stability and consistency of the results. Regarding the replacement of data in Figure 7f, We apologize for any inconsistency. We have conducted replicates of the experiments for both SW480 and HCT116 cells in the previous revision. Through these replicates, we consistently observed that overexpression of MYG1^M inhibited OCR in both SW480 and HCT116 cells. Therefore, we chose to show the more representative results by replacing the previous data for SW480. The results obtained from HCT116 cells were consistent across the replicates and the previous data provided representative results for the finding, so we didn't replace it with the new data.

Previous technical point 5:

The analysis of Figure 7A,B is identical to the previous comment, and does not address issues surrounding the use of a line that extends outside cytoplasmic boundaries, the use of only a single line from a single image, and no quantitation or statistical analysis.

With respect to Figure E, the figure legend does not clarify which channels were used to calculate the pearson's correlation coefficient, or what the data points represent? Is this one image each from 3 patients for the MYG1/CYCS image? It is also not clear how this correlation coefficient was calculated (no description in the methods section) ... Were the fluorescent values centered/normalized? Was a mask applied to only include cell boundaries? How was the mask created? By analyzing a region-of-interest that extends beyond cell boundaries, one can artificially elevate correlation as neither channel will have signal in the blank regions.

These same issues apply to Figure 5a, where lines are drawn outside cell boundaries and no methods description is provided for calculation of the PCC.

Thank you for your comments. We appreciate your concerns regarding the analysis of Figure 7A and B. In the revised version, we have addressed these issues by selecting regions of interest (ROIs) for quantitative. As shown in revised Fig. 7a, We analyzed at least 2 ROIs for each image and calculated the average Pearson correlation coefficient (PCC) for multiple images for quantitative presentation. In Fig.7c, each point represents the average PCC value of an independent experiment and we conducted co-localization analysis on three channels pairwise.

Regarding previous Fig. 7E, we used the MYG1 and Cyt c channels for co-localization analysis and calculated the PCC. The PCC of multiple images for each patient was analyzed and averaged. Each data point in PCC analysis represents the average value of each patient and a total of three patients were analyzed (Revised Fig. 7f). Similar considerations were taken into account for Fig. 5a, where we also used the nuclear region as an ROI for analysis. Each data point represents the average value from individual independent experiments.

PCC was calculated using the following formula:

$$PCC = \frac{\sum_i (R_i - \bar{R}) \times (G_i - \bar{G})}{\sqrt{\sum_i (R_i - \bar{R})^2 \times \sum_i (G_i - \bar{G})^2}}$$

The PCC values were quantified using ImageJ software, and a detailed description of the calculation method has been provided in the methods section and figure legends.

(Page 35, Line 751-754)

“For co-localization analysis, at least two regions of interest (ROIs) are selected for each image, and image J is used for analysis. Pearson correlation coefficient (PCC) of two channels was calculated and averaged across all images in each independent experiment (35).”

References:

(35) Dunn KW, Kamocka MM, McDonald JH. A practical guide to evaluating colocalization in biological microscopy. *Am J Physiol Cell Physiol.* 2011 Apr;300(4):C723-42.

Previous technical point 6:

It is not clear if p-values from one-way ANOVA calculations and other multiple-tests have been corrected for multiple comparisons and what method was used. For these values, the authors should take care to specify what comparison the p-value refers to (e.g., Fig. 1f)

Shouldn't Figure 4l,m be a 2-way ANOVA?

Thank you for the comment. We apologize for any confusion caused by the lack of clarity in our initial descriptions. We used Tukey's multiple comparisons test between groups. We have described this in the figure legends.

In the revised figure legends, we have now explicitly mentioned the control group corresponding to the p-value of each multiple comparison.

Regarding Figure 4l and m, we agree with your suggestion that a two-way

ANOVA should be used. Additionally, we have utilized the Tukey test for multiple comparisons.

Previous technical point 7:

Minor point: Y-axis title for Figure 3h should read “ 5×10^5 cells” not “ 5×10^5 cells”

We are sorry for the oversight. We have now corrected the Y-axis title in Figure 3h, changing it to " 5×10^5 cells".